# Extragradient Method for $(L_0, L_1)$-Lipschitz Root-finding Problems

**Sayantan Choudhury**
AMS & MINDS
Johns Hopkins University
schoudh8@jhu.edu

**Nicolas Loizou**
AMS & MINDS
Johns Hopkins University
nloizou1@jhu.edu

## Abstract

Introduced by Korpelevich in 1976, the extragradient method (EG) has become a cornerstone technique for solving min-max optimization, root-finding problems, and variational inequalities (VIs). Despite its longstanding presence and significant attention within the optimization community, most works focusing on understanding its convergence guarantees assume the strong $L$-Lipschitz condition. In this work, building on the proposed assumptions by Zhang et al. [2020b] for minimization and Vankov et al. [2024] for VIs, we focus on the more relaxed $\alpha$-symmetric $(L_0, L_1)$-Lipschitz condition. This condition generalizes the standard Lipschitz assumption by allowing the Lipschitz constant to scale with the operator norm, providing a more refined characterization of problem structures in modern machine learning. Under the $\alpha$-symmetric $(L_0, L_1)$-Lipschitz condition, we propose a novel step size strategy for EG to solve root-finding problems and establish sublinear convergence rates for monotone operators and linear convergence rates for strongly monotone operators. Additionally, we prove local convergence guarantees for weak Minty operators. We supplement our analysis with experiments validating our theory and demonstrating the effectiveness and robustness of the proposed step sizes for EG.

## 1 Introduction

Min-max optimization problems have recently attracted significant interest due to their widespread applications in machine learning, such as reinforcement learning [Brown et al., 2020, Sokota et al., 2023], distributionally robust optimization [Namkoong and Duchi, 2016], and generative adversarial network training [Goodfellow et al., 2020]. These problems are often formulated as variational inequalities (VIs) [Ryu and Yin, 2022, Gidel et al., 2019, Sokota et al., 2023]. In the unconstrained case, the VI problem simplifies to the root-finding problem [Luo and Tran-Dinh, 2022, Tran-Dinh, 2024], defined as follows [Gorbunov et al., 2022b]:

$$\text{Find } x_* \in \mathbb{R}^d \text{ such that } F(x_*) = 0, \tag{1}$$

where $F : \mathbb{R}^d \to \mathbb{R}^d$ is an operator. Root finding problems of the form (1) encompass a variety of problems as special cases, such as: (i) **Unconstrained minimization:** finding a stationary point of $\min_{x \in \mathbb{R}^d} f(x)$, is equivalent to solving (1) with $F(x) = \nabla f(x)$, (ii) **Min-max optimization:** Let $\min_{w_1 \in \mathbb{R}^{d_1}} \max_{w_2 \in \mathbb{R}^{d_2}} \mathcal{L}(w_1, w_2)$ where $\mathcal{L} : \mathbb{R}^{d_1} \times \mathbb{R}^{d_2} \to \mathbb{R}$. In this scenario, if in (1), the operator is selected as follows:

$$F(x) \equiv \left( \nabla_{w_1} \mathcal{L}(w_1, w_2)^\top, \ -\nabla_{w_2} \mathcal{L}(w_1, w_2)^\top \right)^\top, \tag{2}$$

then solving (1) amounts to finding a stationary point $x_* = (w_{1*}^\top, w_{2*}^\top)^\top \in \mathbb{R}^{d_1 + d_2}$ of the min-max problem, which for convex-concave functions $\mathcal{L}$ is a global solution (Nash equilibrium), i.e.,

$\mathcal{L}(w_{1*}, w_2) \le \mathcal{L}(w_{1*}, w_{2*}) \le \mathcal{L}(w_1, w_{2*})$ [Luo and O'Neill, 2025, Choudhury et al., 2024], and (iii) **Multiplayer games:** A Nash equilibrium $x_* = (w_{1*}^\top, \ldots, w_{N*}^\top)^\top$ of an $N$-player game in which each player $i$ minimizes their own convex objective $\mathcal{L}_i(w_i, w_{-i})$ with respect to $w_i$ (here $w_{-i}$ denotes the actions of all players except $i$) is also captured by (1), with an operator $F(x) = \left( \nabla_{w_1} \mathcal{L}_1(w_1, w_{-1})^\top, \ldots, \nabla_{w_N} \mathcal{L}_N(w_N, w_{-N})^\top \right)^\top$ [Yoon et al., 2025].

Problem (1) and algorithms for solving it have been studied extensively in recent years under different conditions on the operator $F$ [Loizou et al., 2021, 2020, Gorbunov et al., 2022a, Diakonikolas et al., 2021, Choudhury et al., 2024]. One of the well-known algorithms for solving VIs and root-finding problems of the form (1) is the extragradient (EG) method [Korpelevich, 1977] due to its superior convergence guarantees [Gorbunov et al., 2022b]. The algorithm is defined as follows

$$
\begin{aligned}
\hat{x}_k &= x_k - \gamma_k F(x_k), \\
x_{k+1} &= x_k - \omega_k F(\hat{x}_k)
\end{aligned}
\tag{3}
$$

where $\gamma_k > 0$ and $\omega_k > 0$ are the extrapolation step size and update step size, respectively. Since its original inception by Korpelevich, the EG method was revisited and extended in various ways, e.g., non-monotone operators [Diakonikolas et al., 2021, Fan et al., 2023] stochastic [Mishchenko et al., 2020, Gorbunov et al., 2022a, Choudhury et al., 2024, Li et al., 2022], distributed [Beznosikov et al., 2022]. Despite a rich literature for analysing EG and its variants, most of the existing convergence guarantees heavily rely on the $L$-Lipschitz assumption of the operator $F$ [Korpelevich, 1977, Diakonikolas et al., 2021], i.e.

$$
\|F(x) - F(y)\| \le L\|x - y\|
\tag{4}
$$

for all $x, y \in \mathbb{R}^d$. However, this assumption can be restrictive; for instance, the operator $F(x) = x^2$ for $x \in \mathbb{R}$ does not satisfy (4) for any finite $L$ [Zhang et al., 2020b]. *The primary goal of this work is to relax this assumption and establish convergence guarantees under a more general framework.*

**Relaxing the $L$-Lipschitz Assumption.** Recently, Zhang et al. [2020b] introduced the $(L_0, L_1)$-smoothness assumption for the minimization problems. Specifically, for $\min_{x \in \mathbb{R}^d} f(x)$, Zhang et al. [2020b] assume $\|\nabla^2 f(x)\| \le L_0 + L_1 \|\nabla f(x)\|$ (when $f$ is twice differentiable) and later [Chen et al., 2023] proved that this is equivalent to:

$$
\|\nabla f(x) - \nabla f(y)\| \le (L_0 + L_1\|\nabla f(x)\|)\|x - y\|.
\tag{5}
$$

Zhang et al. [2020b] demonstrated that modern neural networks, such as LSTMs (Long Short-Term Memorys) [Hochreiter and Schmidhuber, 1997], align with $(L_0, L_1)$-smoothness assumption rather than the traditional $L$-smoothness assumption (i.e. $\|\nabla f(x) - \nabla f(y)\| \le L\|x - y\|$). Moreover, they used this assumption to justify why gradient clipping speeds up neural network training. Later, Ahn et al. [2024] showed that similar trends hold for the transformer [Vaswani et al., 2017] architecture.

In Figure 1, we present an example demonstrating the validity of the $(L_0, L_1)$-smoothness condition (5) for the iterates of the EG. Similar plots have been presented for gradient descent methods [Zhang et al., 2020b, Gorbunov et al., 2025], but to the best of our knowledge, this linear connection between $\|\nabla^2 f(x_k)\|$ and $\|\nabla f(x_k)\|$ for EG iterates was never reported before. Following Gorbunov et al. [2025], we consider the minimization problem $\min_{x \in \mathbb{R}^d} f(x) := \log\left(1 + \exp(-a^\top x)\right)$, and plot the values of $\|\nabla^2 f(x_k)\|$ on the $y$-axis against $\|\nabla f(x_k)\|$ on the $x$-axis. Each point is colored according to the iteration index $k$, as indicated by the accompanying colorbar. The resulting plot reveals an approximately linear relationship between $\|\nabla^2 f(x_k)\|$ and $\|\nabla f(x_k)\|$, thereby supporting the modeling of this function within the $\|\nabla^2 f(x)\| \le L_0 + L_1\|\nabla f(x)\|$ or $(L_0, L_1)$-smoothness framework.

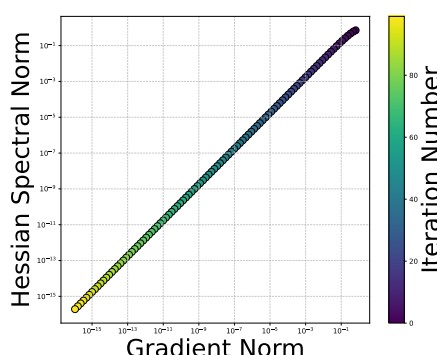

Figure 1: Scatter plot of $\|\nabla^2 f(x_k)\|$ on $y$-axis and $\|\nabla f(x_k)\|$ on $x$-axis.

Now, let us consider the min-max optimization problem $\min_{w_1 \in \mathbb{R}^{d_1}} \max_{w_2 \in \mathbb{R}^{d_2}} \mathcal{L}(w_1, w_2)$, which is captured by (1) with the operator (2). If the operator $F$ is $L$-Lipschitz, then its Jacobian matrix $\mathbf{J}(x)$, defined in (11), satisfies $\|\mathbf{J}(x)\| \le L$ for all $x = (w_1^\top, w_2^\top)^\top$ (follows from Theorem 2.1

with $L_1 = 0$). For example, consider the quadratic min-max objective $\min_{w_1} \max_{w_2} \mathcal{L}(w_1, w_2) = \frac{1}{2}w_1^2 + w_1 w_2 - \frac{1}{2}w_2^2$. In this case, implementing the EG method and plotting the Jacobian norm $\|\mathbf{J}(x_k)\|$ (on the $y$-axis) against the operator norm $\|F(x_k)\|$ (on the $x$-axis) yields a horizontal line parallel to $x$-axis (see Appendix B).

However, this behaviour does not persist for more complex problems. For instance, for the cubic objective

$$\min_{w_1} \max_{w_2} \mathcal{L}(w_1, w_2) = \tfrac{1}{3}w_1^3 + w_1 w_2 - \tfrac{1}{3}w_2^3, \tag{6}$$

$\|\mathbf{J}(x_k)\|$ increases with the $\|F(x_k)\|$. This observation suggests that the standard Lipschitz assumption may be overly restrictive for capturing the structure of such problems (check Figure 2).

To better model this relationship, we investigate a relaxed condition of the form $\|\mathbf{J}(x)\| \leq L_0 + L_1\|F(x)\|^\alpha$ with $\alpha \in (0, 1]$ which generalizes the standard Lipschitz bound (for $L_1 = 0$, this boils down to $\|\mathbf{J}(x)\| \leq L_0$, which is the Lipschitz property). Note that, instead of $\alpha = 1$, our formulation permits $\alpha$ to lie in the broader interval $(0, 1]$. This condition is motivated by the plot in Figure 2, which suggests a sublinear relationship, resembling the form $h(r) = L_0 + L_1 r^\alpha$ for some $\alpha \in (0, 1)$, rather than a linear trend.

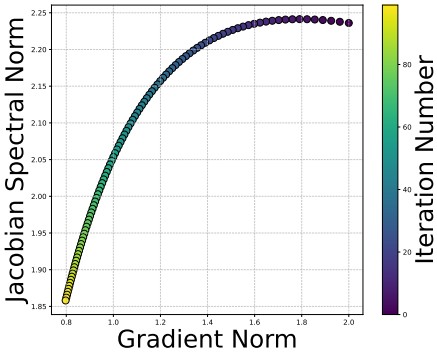

Figure 2: Scatter plot of $\|\mathbf{J}(x_k)\|$ on $y$-axis and $\|F(x_k)\|$ on $x$-axis.

As we will prove later (in Theorem 2.1), for any doubly differentiable min-max optimization problem $\min_{w_1} \max_{w_2} \mathcal{L}(w_1, w_2)$, the condition $\|\mathbf{J}(x)\| \leq L_0 + L_1\|F(x)\|^\alpha$ is equivalent to the operator $F$ satisfying the $\alpha$-symmetric $(L_0, L_1)$-Lipschitz condition (see Assumption 1.1). The equivalent $\alpha$-symmetric $(L_0, L_1)$-Lipschitz condition does not rely on second-order information and applies to a broader class of problems (no need for double differentiability). Therefore, in the remainder of this work, we focus on analyzing the convergence of EG under the $\alpha$-symmetric $(L_0, L_1)$-Lipschitz assumption [Vankov et al., 2024] on the operator $F$, defined below.

**Assumption 1.1.** $F$ is called $\alpha$-symmetric $(L_0, L_1)$-Lipschitz operator if for some $L_0, L_1 \geq 0$ and $\alpha \in (0, 1]$,

$$\|F(x) - F(y)\| \leq \left(L_0 + L_1 \max_{\theta \in [0,1]} \|F(\theta x + (1 - \theta)y)\|^\alpha\right)\|x - y\| \quad \forall x, y \in \mathbb{R}^d. \tag{7}$$

Instead of a fixed Lipschitz constant in (4), Assumption 1.1 allows the Lipschitz-like quantity to depend on the norm of the operator itself along the path from $x$ to $y$. This assumption generalizes the standard $L$-Lipschitz condition (4), corresponding to the special case where $L_0 = L$ and $L_1 = 0$. Moreover, the $\alpha$-symmetric $(L_0, L_1)$-Lipschitz condition provides a more refined characterisation of operators whose Lipschitz constant depends on their norm, offering a tighter bound by balancing $L_0 \ll L$ and $L_1 \ll L$. Additionally, (7) provides a more relaxed bound compared to (5) with $\alpha = 1$.

**Classes of Root-finding Problems.** Apart from the condition (7), we will also assume additional structure on the operator $F$ to prove convergence. We say the operator $F$ is monotone or strongly monotone if it satisfies the following assumption.

**Assumption 1.2.** $F$ is called monotone if

$$\langle F(x) - F(y), x - y \rangle \geq 0 \quad \forall x, y \in \mathbb{R}^d \tag{8}$$

and strongly monotone if there is some $\mu > 0$ such that

$$\langle F(x) - F(y), x - y \rangle \geq \mu\|x - y\|^2 \quad \forall x, y \in \mathbb{R}^d. \tag{9}$$

This captures convex minimization and convex-concave min-max optimization problems as a special case. Apart from the monotone operators, we are also interested in some non-monotone operators, weak Minty operators [Diakonikolas et al., 2021], which satisfy the following assumption.

Table 1: Summary of step size selection for EG under the $L$-Lipschitz and $\alpha$-symmetric $(L_0, L_1)$-Lipschitz assumptions. Our proposed step size strategy is of the general form $\gamma_k = \frac{1}{c_0 + c_1 \|F(x_k)\|^\alpha}$, tailored for solving problems involving $\alpha$-symmetric $(L_0, L_1)$-Lipschitz operators.

| Setup | Assumption | $\alpha$ | $\gamma_k$ | $\omega_k$ |
|---|---|---|---|---|
| Strongly Monotone [1] | $L$-Lipschitz [2] [Mokhtari et al., 2020] | - | $\frac{0.25}{L}$ | $\gamma_k$ |
| | $\alpha$-symmetric $(L_0, L_1)$-Lipschitz [Vankov et al., 2024] | 1 | $\min\left\{\frac{1}{4\mu}, \frac{1}{2\sqrt{2}eL_0}, \frac{1}{2\sqrt{2}eL_1\|F(x_k)\|}\right\}$ | $\gamma_k$ |
| | $\alpha$-symmetric $(L_0, L_1)$-Lipschitz (Theorem 3.2) | 1 | $\frac{0.21}{L_0 + L_1\|F(x_k)\|}$ | $\gamma_k$ |
| | $\alpha$-symmetric $(L_0, L_1)$-Lipschitz (Theorem 3.4) | $(0,1)$ | $\frac{0.61}{2K_0 + (2K_1 + 2^{1-\alpha}K_2^{1-\alpha})\|F(x_k)\|^\alpha}$ [2] | $\gamma_k$ |
| Monotone [1] | $L$-Lipschitz [Gorbunov et al., 2022b] | - | $\frac{1}{L}$ | $\frac{\gamma_k}{2}$ |
| | $\alpha$-symmetric $(L_0, L_1)$-Lipschitz (Theorem 3.5) | 1 | $\frac{0.45}{L_0 + L_1\|F(x_k)\|}$ | $\gamma_k$ |
| | $\alpha$-symmetric $(L_0, L_1)$-Lipschitz (Theorem 3.7) | $(0,1)$ | $\frac{1}{2\sqrt{2}K_0 + (2\sqrt{2}K_1 + 2^{3(1-\alpha)/2}K_2^{1-\alpha})\|F(x_k)\|^\alpha}$ | $\gamma_k$ |
| Weak Minty [1] | $L$-Lipschitz [Diakonikolas et al., 2021] | - | $\frac{1}{L}$ | $\frac{\gamma_k}{2}$ |
| | $L$-Lipschitz [Pethick et al., 2022] | - | $\frac{1}{L}$ | $\rho + \frac{\langle F(\hat{x}_k), x_k - \hat{x}_k\rangle}{\|F(\hat{x}_k)\|^2}$ |
| | $\alpha$-symmetric $(L_0, L_1)$-Lipschitz (Theorem 3.8) | 1 | $\frac{0.56}{L_0 + L_1\|F(x_k)\|}$ | $\frac{\gamma_k}{2}$ |
| | $\alpha$-symmetric $(L_0, L_1)$-Lipschitz (Theorem 3.9) | $(0,1)$ | $\frac{1}{2\sqrt{2}K_0 + (2\sqrt{2}K_1 + 2^{3(1-\alpha)/2}K_2^{1-\alpha})\|F(x_k)\|^\alpha}$ | $\frac{\gamma_k}{2}$ |

[1] Convergence measure ● strongly monotone: $\|x_K - x_*\|^2$, ● monotone: $\min_{0\le k\le K}\|F(x_k)\|^2$, ● weak minty: $\min_{0\le k\le K}\|F(\hat{x}_k)\|^2$.

[2] For $K_0, K_1, K_2$, check Proposition 3.1. Note that, for $L_1 = 0$ we have $K_1 = K_2 = 0$.

**Assumption 1.3.** Operator $F$ is called weak Minty if for some $\rho \geq 0$,

$$\langle F(x), x - x_*\rangle \geq -\rho\|F(x)\|^2 \qquad \forall x \in \mathbb{R}^d. \tag{10}$$

## 1.1 Main Contributions

We summarize the main contributions of our work below.

- **Tighter analysis for strongly monotone:** We establish linear convergence guarantees for strongly monotone (9) $\alpha$-symmetric $(L_0, L_1)$-Lipschitz problems (see Theorem 3.2, 3.4). In contrast to the results in Vankov et al. [2024] for $\alpha = 1$, our analysis shows that linear convergence can be achieved without incurring exponential dependence on the initial distance to the solution $\|x_0 - x_*\|$ (see Corollary 3.3).
- **First analysis for monotone and weak Minty:** We provide the first convergence analysis of EG for solving monotone (8) and weak Minty (10) problems under $\alpha$-symmetric $(L_0, L_1)$-Lipschitz assumption. We establish global sublinear convergence for monotone problems (see Theorem 3.5, 3.7) and local sublinear convergence for weak Minty problems (see Theorem 3.8, 3.9).
- **Novel step size for EG:** We propose a novel adaptive step size strategy for the EG method designed to handle $\alpha$-symmetric $(L_0, L_1)$-Lipschitz operators. Specifically, all our step size schemes adopt the general form $\gamma_k = \frac{1}{c_0 + c_1\|F(x_k)\|^\alpha}$, where $c_0, c_1 > 0$ are constants determined by the problem-dependent parameters $L_0$, $L_1$, and $\alpha$. In Table 1, we included a detailed summary of our proposed step size selection for different classes of operators and compared it with closely related works.
- **Numerical experiments:** Finally, in Section 4, we present experiments validating different aspects of our theoretical results. We compare our proposed step size selections with existing alternatives, demonstrating the effectiveness and robustness of our approach.

## 2 On the $\alpha$-Symmetric $(L_0, L_1)$-Lipschitz Assumption

We divide this section into two parts. In the first subsection, we present an equivalent reformulation of the $\alpha$-symmetric $(L_0, L_1)$-Lipschitz condition (7) in the context of min-max optimization. In

the second subsection, we provide some examples of operators that satisfy (7) and highlight its significance.

## 2.1 Equivalent Formulation of $\alpha$-Symmetric $(L_0, L_1)$-Lipschitz Assumption

In this subsection, we consider the min-max optimization problem $\min_{w_1} \max_{w_2} \mathcal{L}(w_1, w_2)$. The corresponding operator $F$ and Jacobian $\mathbf{J}$ are defined as

$$F(x) = \begin{bmatrix} \nabla_{w_1} \mathcal{L}(w_1, w_2) \\ -\nabla_{w_2} \mathcal{L}(w_1, w_2) \end{bmatrix} \text{ and } \mathbf{J}(x) = \begin{bmatrix} \nabla^2_{w_1 w_1} \mathcal{L}(w_1, w_2) & \nabla^2_{w_2 w_1} \mathcal{L}(w_1, w_2) \\ -\nabla^2_{w_1 w_2} \mathcal{L}(w_1, w_2) & -\nabla^2_{w_2 w_2} \mathcal{L}(w_1, w_2) \end{bmatrix}, \quad (11)$$

where $x = (w_1^\top, w_2^\top)^\top$. Assuming that $F$ is $\alpha$-symmetric $(L_0, L_1)$-Lipschitz, we obtain the following theorem.

> **Theorem 2.1.** Suppose $F$ is the differentiable operator associated with the problem $\min_{w_1} \max_{w_2} \mathcal{L}(w_1, w_2)$. Then $F$ satisfies the $\alpha$-symmetric $(L_0, L_1)$-Lipschitz condition (7) if and only if
> $$\|\mathbf{J}(x)\| \le L_0 + L_1 \|F(x)\|^\alpha. \quad (12)$$
> Here $\mathbf{J}(x)$ is the Jacobian defined in (11) and $\|\mathbf{J}(x)\| = \sigma_{\max}(\mathbf{J}(x))$ i.e. maximum singular value of $\mathbf{J}(x)$. In particular, we have $\|\mathbf{J}(x)\| \le L$ when operator $F$ is $L$-Lipschitz.

This result provides an equivalent characterization of the $\alpha$-symmetric $(L_0, L_1)$-Lipschitz condition (7) for min-max optimization problems. In practice, it is often easier to verify (12) than to directly check (7). In Appendix E, we provide an example where we use Theorem 2.1 to verify if an operator satisfies (7).

## 2.2 Examples of $\alpha$-Symmetric $(L_0, L_1)$-Lipschitz Operators

To motivate the significance of this relaxed assumption (7), we present a few instances of $\alpha$-symmetric $(L_0, L_1)$-Lipschitz operators that highlight its advantages over the conventional $L$-Lipschitz assumption.

**Example 1**[Gorbunov et al., 2025]: We start with an example from the minimization setting. Consider the logistic regression loss function $f(x) = \log\left(1 + \exp\left(-a^\top x\right)\right)$. Then the corresponding gradient operator $F = \nabla f$ satisfies the $L$-Lipschitz assumption with $L = \|a\|^2$ and $\alpha$-symmetric $(L_0, L_1)$-Lipschitz assumption with $L_0 = 0, L_1 = \|a\|, \alpha = 1$. Therefore, when $\|a\| \gg 1$, the bound provided by 1-symmetric $(L_0, L_1)$-Lipschitz can be significantly tighter than the one imposed by the $L$-Lipschitz condition. This example emphasizes the benefit of the $\alpha$-symmetric $(L_0, L_1)$-Lipschitz framework in scenarios where standard Lipschitz constants are overly pessimistic.

**Example 2**: Consider the operator $F(x) = (u_1^2, u_2^2)$ for $x = (u_1, u_2) \in \mathbb{R}^2$ with $x_* = (0, 0)$. Then we can show that

$$\|F(x) - F(y)\| \le 2\left\|F\left(\tfrac{x+y}{2}\right)\right\|^{1/2} \|x - y\| \le 2\left\|\max_{\theta \in [0,1]} F\left(\theta x + (1-\theta)y\right)\right\|^{1/2} \|x - y\|.$$

This establishes that $F$ is $1/2$-symmetric $(0, 2)$-Lipschitz operator. However, this operator $F$ does not satisfy the standard $L$-Lipschitz assumption for any finite choice of $L$. We add the related details to Appendix C. Therefore, this example highlights the need for relaxed assumptions on operators beyond standard $L$-Lipschitz (4).

**Example 3.** Consider the following min-max optimization problem, for any $p > 1$,

$$\min_{w_1} \max_{w_2} \ \mathcal{L}(w_1, w_2) = \tfrac{1}{p+1}\|w_1\|^{p+1} + w_1^\top \mathbf{B} w_2 - \tfrac{1}{p+1}\|w_2\|^{p+1}.$$

The corresponding operator $F(x)$ defined in (2) is 1-symmetric $\left(2\tau_0 + \|\mathbf{M}\|, 2^{\frac{2p^2-1}{p^2}} \tau_1\right)$-Lipschitz, for any choice of $\tau_1 > 0$ and $\tau_0 = \left(\frac{p-1}{\tau_1}\right)^{p-1}, \mathbf{M} = \begin{bmatrix} 0 & \mathbf{B} \\ -\mathbf{B}^\top & 0 \end{bmatrix}$. Moreover, $F$ is not $L$-Lipschitz for any finite $L$. We have added the proof in Appendix C.

In Appendix C, we also provide additional examples that illustrate cases where the operator associated with a general bilinearly coupled min-max optimization problem or an $N$-player game satisfies the $\alpha$-symmetric $(L_0, L_1)$-Lipschitz condition.

# 3 Convergence Analysis

In this section, we present the convergence guarantees of EG for solving monotone, strongly monotone, and weakly Minty operators. For strongly monotone operators, we have linear convergence, while for monotone and weak Minty operators, we provide sublinear convergence guarantees. To prove these results, we rely on the similar expression presented in Chen et al. [2023] for the $(L_0, L_1)$-smooth minimization problem. For completeness, we include the proof for $\alpha$-symmetric $(L_0, L_1)$-Lipschitz operators in the Appendix.

**Proposition 3.1.** Suppose $F$ is $\alpha$-symmetric $(L_0, L_1)$-Lipschitz operator. Then, for $\alpha = 1$

$$\|F(x) - F(y)\| \leq (L_0 + L_1 \|F(x)\|) \exp (L_1 \|x - y\|) \|x - y\|, \tag{13}$$

and for $\alpha \in (0, 1)$ we have

$$\|F(x) - F(y)\| \leq \left(K_0 + K_1 \|F(x)\|^{\alpha} + K_2 \|x - y\|^{\alpha/1-\alpha}\right) \|x - y\| \tag{14}$$

where $K_0 = L_0(2^{\alpha^2/1-\alpha} + 1)$, $K_1 = L_1 \cdot 2^{\alpha^2/1-\alpha}$ and $K_2 = L_1^{1/1-\alpha} \cdot 2^{\alpha^2/1-\alpha} \cdot 3^{\alpha}(1 - \alpha)^{\alpha/1-\alpha}$.

Proposition 3.1 eliminates the maximum over $\theta \in [0, 1]$ in (7) and provides a simpler upper bound on the $\|F(x) - F(y)\|$. We divide the rest of the section into three subsections based on the structure of operators. Moreover, each of these subsections is divided into two parts depending on the value of $\alpha$, i.e. $\alpha = 1$ and $\alpha \in (0, 1)$.

## 3.1 Convergence Guarantees for Strongly Monotone Operators

In case of strongly monotone operators (9), we achieve linear convergence rates, analogous to those obtained under standard $L$-Lipschitz assumptions [Tseng, 1995, Mokhtari et al., 2020]. For $\alpha = 1$, operator $F$ satisfies the condition (13). To guarantee convergence for this class of operators, we use the EG with step size $\gamma_k = \omega_k = \nu/(L_0 + L_1 \|F(x_k)\|)$ and $\nu > 0$. Gorbunov et al. [2025] used similar step sizes for the Gradient Descent algorithm to solve convex minimization problems.

**Theorem 3.2.** Suppose $F$ is $\mu$-strongly monotone and 1-symmetric $(L_0, L_1)$-Lipschitz operator. Then EG with step size $\gamma_k = \omega_k = \frac{\nu}{L_0 + L_1 \|F(x_k)\|}$ satisfy

$$\|x_{k+1} - x_*\|^2 \leq \left(1 - \frac{\nu\mu}{L_0 \left(1 + L_1 \exp (L_1 \|x_0 - x_*\|) \|x_0 - x_*\|\right)}\right)^{k+1} \|x_0 - x_*\|^2$$

where $\nu > 0$ satisfy $1 - 2\nu - \nu^2 \exp 2\nu = 0$.

The equation $1 - 2\nu - \nu^2 \exp (2\nu) = 0$ admits a positive solution, approximately $\nu \approx 0.363$. Specifically, to ensure $\|x_K - x_*\|^2 \leq \varepsilon$, we require $K = \mathcal{O}\left(\left(\frac{L_0}{\mu} + \frac{L_0 L_1 \|x_0 - x_*\| \exp (L_1 \|x_0 - x_*\|)}{\mu}\right) \log \frac{1}{\varepsilon}\right)$ iterations. When $L_1 = 0$, we recover the best-known results for the strongly monotone $L$-Lipschitz setting [Tseng, 1995, Mokhtari et al., 2020]. Vankov et al. [2024] also studied constrained strongly monotone problems and obtained similar guarantees with an alternative step size scheme.

However, using a refined proof technique, we can eliminate the $\exp(L_1 \|x_0 - x_*\|)$ term from the convergence rate and establish a tighter bound. One of the intermediate steps of Theorem 3.2 is proving a lower bound on the step size $\gamma_k$, which can be very loose for large $k$. We now show that after a certain number of iterations $K'$ (51), the operator norm satisfies $\|F(x_k)\| \leq L_0/L_1$ for all $k \geq K'$, which implies $\gamma_k = \omega_k \geq \nu/2L_0$ for all $k \geq K'$.

**Corollary 3.3.** Suppose $F$ is a $\mu$-strongly monotone and 1-symmetric $(L_0, L_1)$-Lipschitz operator. Then, EG with step sizes $\gamma_k = \omega_k = \frac{\nu}{L_0 + L_1 \|F(x_k)\|}$ guarantees $\|x_{K+1} - x_*\|^2 \leq \varepsilon$ after at most

$$K = \underbrace{\frac{2L_0}{\nu\mu} \log \left(\frac{\|x_0 - x_*\|^2}{\varepsilon}\right)}_{\text{Term I}} + \underbrace{\frac{1}{\zeta\mu} \log \left(\frac{2L_1 \|x_0 - x_*\|^2}{\zeta^2 L_0}\right)}_{\text{Term II}} \tag{15}$$

iterations, where we have $\zeta := \nu/L_0(1 + L_1 \exp(L_1 \|x_0 - x_*\|) \|x_0 - x_*\|)$, and $\nu > 0$ satisfies $1 - 4\nu - 2\nu^2 \exp(2\nu) = 0$.

This result shows that to reach an accuracy of $\varepsilon > 0$, we need (15) iterations. Importantly, Term II in (15) is independent of $\varepsilon$, and the Term I of (15) no longer depends on $\exp(L_1\|x_0 - x_*\|)$. Technically, Term II corresponds to the number of iterations required for the step sizes $\gamma_k$ and $\omega_k$ to exceed $\nu/2L_0$, while Term I captures the iteration complexity of EG with a fixed step size $\nu/2L_0$.

Now, we investigate the behavior of $\alpha$-symmetric $(L_0, L_1)$-Lipschitz operators for $0 < \alpha < 1$. In this regime, we adopt a step size of the order $\mathcal{O}\left(\|F(x_k)\|^{-\alpha}\right)$ and prove the following result.

---

**Theorem 3.4.** Suppose $F$ is $\mu$-strongly monotone and $\alpha$-symmetric $(L_0, L_1)$-Lipschitz operator with $\alpha \in (0, 1)$. Then EG with $\gamma_k = \omega_k = \frac{\nu}{2K_0 + \left(2K_1 + 2^{1-\alpha}K_2^{1-\alpha}\right)\|F(x_k)\|^\alpha}$ satisfy

$$\|x_{k+1} - x_*\|^2 \leq \left(1 - \frac{\nu\mu}{2K_0 + (2K_1 + 2^{1-\alpha}K_2^{1-\alpha})(K_0 + K_2\|x_0 - x_*\|^{\alpha/1-\alpha})^\alpha\|x_0 - x_*\|^\alpha}\right)^{k+1}\|x_0 - x_*\|^2$$

where $\nu \in (0, 1)$ is a constant such that $1 - \nu - \nu^2 = 0$.

---

This result establishes linear convergence. In particular, to ensure $\|x_K - x_*\|^2 \leq \varepsilon$, it suffices to run $K = \mathcal{O}\left(\left(\frac{K_0}{\mu} + \frac{(K_1 K_2^\alpha + K_2)\|x_0 - x_*\|^{\alpha/1-\alpha}}{\mu}\right)\log\frac{1}{\varepsilon}\right)$ iterations. Compared to the $L$-Lipschitz setting, the bound here includes an additional dependence on $\|x_0 - x_*\|^{\alpha/1-\alpha}$, which grows larger as $\alpha \to 1$.

## 3.2 Convergence Guarantees for Monotone Operators

In this subsection, we focus on the monotone operators (8). Here we provide the first analysis for the monotone 1-symmetric $(L_0, L_1)$-Lipschitz operators.

---

**Theorem 3.5.** Suppose $F$ is monotone and 1-symmetric $(L_0, L_1)$-Lipschitz operator. Then EG with step size $\gamma_k = \omega_k = \frac{\nu}{L_0 + L_1\|F(x_k)\|}$ satisfy

$$\min_{0 \leq k \leq K}\|F(x_k)\|^2 \leq \frac{2L_0^2(1 + L_1\exp(L_1\|x_0 - x_*\|)\|x_0 - x_*\|)^2\|x_0 - x_*\|^2}{\nu^2(K+1)}. \tag{16}$$

where $\nu\exp\nu = 1/\sqrt{2}$.

---

Note that the solution of $\nu\exp\nu = 1/\sqrt{2}$ is approximately 0.45. Hence, this result proves sublinear convergence of EG when $F$ is monotone. Moreover, (16) implies, EG will need $K = \mathcal{O}\left(\frac{L_0^2\|x_0 - x_*\|^2}{\varepsilon} + \frac{L_0^2 L_1^2\exp(2L_1\|x_0 - x_*\|)\|x_0 - x_*\|^4}{\varepsilon}\right)$ iterations to get $\|F(x_k)\|^2 \leq \varepsilon$ for some $k \leq K$. Therefore the convergence rate exponentially depends on $\|x_0 - x_*\|$ when $L_1 > 0$. This shows that 1-symmetric $(L_0, L_1)$-Lipschitz operators potentially require more iterations of EG compared to $L$-Lipschitz operators when initialization $x_0$ is far from $x_*$. However, (16) recovers the best known dependence on $\|x_0 - x_*\|$ as a special case when $L_1 = 0$, i.e. $F$ is a standard Lipschitz operator [Gorbunov et al., 2022b].

Theorem 3.5 shows that the EG's convergence rate has an extra term $\exp(L_1\|x_0 - x_*\|)$ compared to the results of the Lipschitz setting. One of the intermediate steps in this proof involves an upper bound on $\sum_{k=0}^K \gamma_k^2\|F(x_k)\|^2$ (see (56) in Appendix D). Then the simple approach is to get a lower bound on $\gamma_k^2$ for all $k$ and derive (16). This lower bound on $\gamma_k^2$ involves the $\exp(L_1\|x_0 - x_*\|)$ term (see (55) in Appendix D) and can be potentially very small. However, it is possible to eliminate this exponential dependence using a refined proof technique.

---

**Theorem 3.6.** Suppose $F$ is monotone and 1-symmetric $(L_0, L_1)$-Lipschitz operator. Then EG with step size $\gamma_k = \omega_k = \frac{\nu}{L_0 + L_1\|F(x_k)\|}$ satisfy

$$\min_{0 \leq k \leq K}\|F(x_k)\| \leq \frac{\sqrt{2}L_0\|x_0 - x_*\|}{\nu\sqrt{K+1} - \sqrt{2}L_1\|x_0 - x_*\|}$$

where $\nu\exp\nu = 1/\sqrt{2}$ and $K + 1 \geq \frac{2L_1^2\|x_0 - x_*\|^2}{\nu^2}$.

---

Note that to obtain this convergence guarantee, a sufficiently large number of iterations is required, specifically $K + 1 \geq (2L_1^2\|x_0 - x_*\|^2)/\nu^2$. Gorbunov et al. [2025] employed a similar proof technique

to eliminate the exponential dependence on the initial distance $\exp(L_1\|x_0 - x_*\|)$ in the context of the Adaptive Gradient method.

Next we state our result for $\alpha$-symmetric $(L_0, L_1)$-Lipschitz monotone operator with $\alpha \in (0, 1)$.

**Theorem 3.7.** Suppose $F$ is monotone and $\alpha$-symmetric $(L_0, L_1)$-Lipschitz operator with $\alpha \in (0, 1)$. Then EG with $\gamma_k = \omega_k = \frac{1}{2\sqrt{2}K_0 + (2\sqrt{2}K_1 + 2^{3(1-\alpha)/2}K_2^{1-\alpha})\|F(x_k)\|^\alpha}$ satisfy

$$\min_{0 \le k \le K} \|F(x_k)\|^2 \le \frac{16\left(K_0 + (K_1 + 2^{-3/2}K_2^{1-\alpha})(K_0 + K_2\|x_0 - x_*\|^{\alpha/1-\alpha})^\alpha\|x_0 - x_*\|^\alpha\right)^2\|x_0 - x_*\|^2}{K+1}.$$

This theorem establishes a sublinear convergence rate for $\alpha \in (0, 1)$. In the special case where $L_1 = 0$ (i.e., the standard $L$-Lipschitz setting), we have $K_1 = K_2 = 0$ by Proposition 3.1. Thus, our result recovers the best-known rate $\mathcal{O}\left(L_0^2\|x_0 - x_*\|^2/K+1\right)$ from Gorbunov et al. [2022b]. On the other hand, when $L_1 > 0$, we obtain a convergence rate of $\mathcal{O}\left(\|x_0 - x_*\|^{\frac{2+4\alpha-2\alpha^2}{1-\alpha}}/K+1\right)$. Furthermore, as $\alpha \to 0$—which corresponds again to the $L$-Lipschitz setting—our step sizes $\gamma_k$ and $\omega_k$ become constant, and we recover the standard convergence rate $\mathcal{O}\left(\|x_0 - x_*\|^2/K+1\right)$. This matches the classical result for monotone $L$-Lipschitz operators up to constants, emphasizing the tightness of our analysis.

### 3.3 Local Convergence Guarantees for Weak Minty Operators

Beyond the monotone operators, it is also possible to provide convergence for weak Minty operators (10) under some restrictions on $\rho > 0$. In contrast to the monotone problems where we used the same extrapolation and update step $\gamma_k, \omega_k$, here we use smaller update step size $\omega_k$. Specifically, we employ $\omega_k = \gamma_k/2$, similar to Diakonikolas et al. [2021] for handling weak Minty $L$-Lipschitz operators.

**Theorem 3.8.** Suppose $F$ is weak Minty and 1-symmetric $(L_0, L_1)$-Lipschitz assumption. Moreover we assume

$$\Delta_1 := \frac{\nu}{L_0\left(1 + L_1\|x_0 - x_*\|e^{L_1\|x_0 - x_*\|}\right)} - 4\rho > 0. \tag{17}$$

Then EG with step size $\gamma_k = \frac{\nu}{L_0 + L_1\|F(x_k)\|}$ and $\omega_k = \gamma_k/2$ satisfies

$$\min_{0 \le k \le K} \|F(\hat{x}_k)\|^2 \le \frac{4L_0\left(1 + L_1\exp\left(L_1\|x_0 - x_*\|\right)\|x_0 - x_*\|\right)\|x_0 - x_*\|^2}{\nu\Delta_1(K+1)} \tag{18}$$

where $\nu\exp\nu = 1$.

To the best of our knowledge, this is the first result establishing convergence guarantees for weak Minty, $\alpha$-symmetric $(L_0, L_1)$-Lipschitz operators. Similar to the monotone case, we obtain a sublinear convergence rate for weak Minty operators. However, the condition in (17) indicates that the initialization point $x_0$ must be sufficiently close to the solution $x_*$ in order to ensure convergence. Consequently, Theorem 3.8 only provides a local convergence guarantee.

In the special case where $L_1 = 0$, i.e., the standard $L$-Lipschitz setting, condition (17) reduces to the simpler requirement $\rho < \nu/4L_0$. Similar assumptions on $\rho$ have been made in prior works such as Diakonikolas et al. [2021] and Pethick et al. [2022] for the $L$-Lipschitz weak Minty setting. Finally, we extend our analysis to the case $\alpha \in (0, 1)$, and present a corresponding theorem establishing sublinear convergence under analogous restrictions on $\rho$.

**Theorem 3.9.** Suppose $F$ is weak Minty and $\alpha$-symmetric $(L_0, L_1)$-Lipschitz operator with $\alpha \in (0, 1)$. Moreover we assume

$$\Delta_\alpha := \frac{1}{2\sqrt{2}K_0 + 2\sqrt{2}(K_1 + 2^{-3/2}K_2^{1-\alpha})(K_0 + K_2\|x_0 - x_*\|^{\alpha/1-\alpha})^\alpha\|x_0 - x_*\|^\alpha} - 4\rho > 0. \tag{19}$$

Then EG with step size $\gamma_k = \frac{1}{2\sqrt{2}K_0 + (2\sqrt{2}K_1 + 2^{3(1-\alpha)/2}K_2^{1-\alpha})\|F(x_k)\|^\alpha}$ and $\omega_k = \gamma_k/2$ satisfy

$$\min_{0 \le k \le K} \|F(\hat{x}_k)\|^2 \le \frac{4\left(K_0 + \left(K_1 + 2^{-3/2}K_2^{1-\alpha}\right)(K_0 + K_2\|x_0 - x_*\|^{\alpha/1-\alpha})^\alpha\|x_0 - x_*\|^\alpha\right)\|x_0 - x_*\|^2}{\Delta_\alpha(K+1)}.$$

## 4 Numerical Experiments

In this section, we conduct experiments to validate the efficiency of our proposed step size strategy $\gamma_k = \frac{1}{c_0 + c_1 \|F(x_k)\|^\alpha}$ with $\alpha = 1$. In the first experiment, we compare our step size choice with that of Vankov et al. [2024] on a strongly monotone problem, and in the second experiment, we make a comparison with the constant step size strategy for solving a monotone problem. Finally, we evaluate our scheme for solving the GlobalForsaken problem from Pethick et al. [2022]. All experiments in this work were conducted using a personal MacBook with an Apple M3 chip and 16GB of RAM. We provide the code for all of our experiments at https://github.com/isayantan/L0L1extragradient.

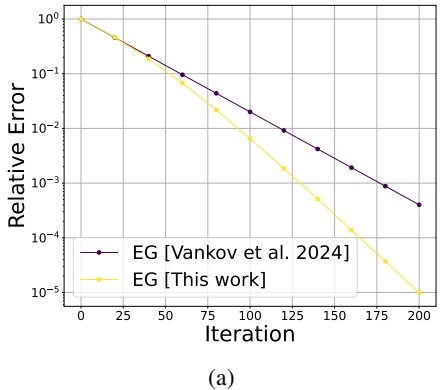
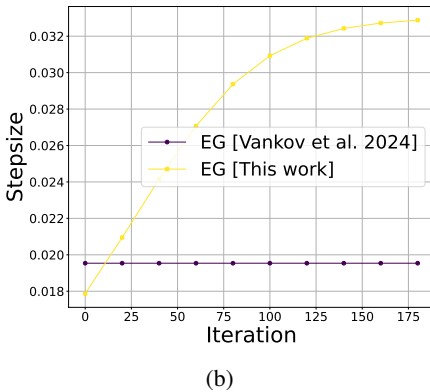

(a)                                         (b)

Figure 3: In Figures 3a and 3b, we compare our proposed adaptive step size strategy with that of Vankov et al. [2024]. We report the relative error and the magnitude of the step size over iterations.

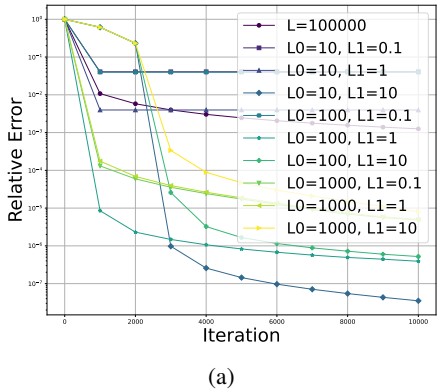
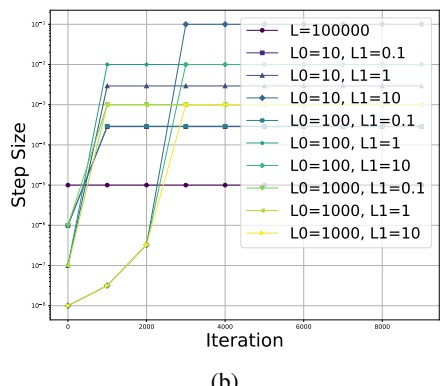

(a)                                         (b)

Figure 4: In Figures 4a and 4b, we evaluate the performance of the EG method on the problem in (20), using both a constant step size and the $(L_0, L_1)$-adaptive step size. We report the relative error and the magnitude of the step size over iterations.

**Performance on a Strongly Monotone Problem.** In this experiment, we compare our theoretical step sizes with those from Vankov et al. [2024]. Here, we implement EG for solving the operator $F(x) = (\text{sign}\,(u_1)\,|u_1| + u_2, \text{sign}\,(u_2)\,|u_2| - u_1)$. This problem has constants $L_0 = 1 + 2\sqrt{2}$ and $L_1 = 2\sqrt{2}$. For our method, we use $\gamma_k = \omega_k = \frac{\nu}{L_0 + L_1 \|F(x_k)\|}$ while for EG [Vankov et al., 2024] we use stepsize $\gamma_k = \omega_k = \min\left\{\frac{1}{4\mu}, \frac{1}{2\sqrt{2}eL_0}, \frac{1}{2\sqrt{2}eL_1\|F(x_k)\|}\right\}$. In Figure 3a, we plot the relative error $\frac{\|x_k - x_*\|^2}{\|x_0 - x_*\|^2}$ on the $y$-axis while number of iterations on the $x$-axis. We find that our proposed step size outperforms that of Vankov et al. [2024]. Moreover, in Figure 3b, we compare the magnitude of the step size and how it evolves over the iterations. We find that the step size of Vankov et al. [2024] remains constant at approximately $0.02$, whereas our proposed step size increases to a value larger than $0.032$. These experiments highlight the efficiency of our proposed step size.

**Performance on a Monotone Problem.** Here we consider the following min-max optimization problem

$$\min_{w_1 \in \mathbb{R}^d} \max_{w_2 \in \mathbb{R}^d} \mathcal{L}(w_1, w_2) = \frac{1}{3} \left( w_1^\top \mathbf{A} w_1 \right)^{3/2} + w_1^\top \mathbf{B} w_2 - \frac{1}{3} \left( w_2^\top \mathbf{C} w_2 \right)^{3/2}. \quad (20)$$

where $\mathbf{A}, \mathbf{B}, \mathbf{C} \in \mathbb{R}^{d \times d}$ are positive definite matrices. Note that, when $d = 1$, and $\mathbf{A}, \mathbf{B}, \mathbf{C}$ are just scalars equal to 1, this problem reduces to (6). The corresponding operator of this problem is given by

$$F(x) = \begin{bmatrix} \nabla_{w_1} \mathcal{L}(w_1, w_2) \\ -\nabla_{w_2} \mathcal{L}(w_1, w_2) \end{bmatrix} = \begin{bmatrix} \left( w_1^\top \mathbf{A} w_1 \right)^{1/2} \mathbf{A} w_1 + \mathbf{B} w_2 \\ \left( w_2^\top \mathbf{C} w_2 \right)^{1/2} \mathbf{C} w_2 - \mathbf{B}^\top w_1 \end{bmatrix}.$$

Furthermore, we show that $\mathcal{L}$ is convex-concave and has an equilibrium only at $w_1, w_2 = 0 \in \mathbb{R}^d$ (check Appendix F). To solve (20), we implement the EG method using two types of step size strategies: (1) a constant step size $\gamma_k = \omega_k = 1/c$, and (2) an adaptive step size $\gamma_k = \omega_k = 1/(c_0 + c_1 \|F(x_k)\|)$. For the constant step size EG, we perform a grid search over $c \in \{10^2, 10^3, 10^4, 10^5, 10^6, 10^7\}$. We find that $c = 10^5$ yields the best performance: larger values lead to slower convergence, while smaller values cause divergence. Figures 4a and 4b present the relative error and step size for the case $c = 10^5$. For our adaptive EG method, we perform a grid search over $c_0 \in \{10, 100, 1000\}$ and $c_1 \in \{0.1, 1, 10\}$, evaluating all 9 possible combinations. The performance of all adaptive variants is plotted in Figures 4a and 4b. We observe that most combinations outperform the constant step size EG, with $(c_0, c_1) = (10, 10)$ achieving the best results (see Figure 4a).

Interestingly, while the adaptive EG starts with smaller step sizes compared to the constant step size EG, its step sizes increase over time and eventually surpass those of the constant step size approach (see Figure 4b). This highlights the practical effectiveness of our proposed method in handling non-Lipschitz operators such as the one in (20).

**Performance on a Weak Minty Problem.** Here we consider the unconstrained GlobalForsaken problem from Pethick et al. [2022] given by

$$\min_{w_1 \in \mathbb{R}} \max_{w_2 \in \mathbb{R}} \mathcal{L}(w_1, w_2) := w_1 w_2 + \psi(w_1) - \psi(w_2), \quad (21)$$

where $\psi(w) = \frac{2w^6}{21} - \frac{w^4}{3} + \frac{w^2}{3}$. As shown in Pethick et al. [2022], the saddle-point problem in (21) admits a global Nash equilibrium at $(w_1, w_2) = (0, 0)$ and satisfies the weak Minty condition (10) with parameter $\rho \approx 0.119732$. We implement Adaptive EG+, EG+ and EG with our step size strategy to solve this problem. For each algorithm, we perform step size tuning on a a grid of $\{10^{-5}, 10^{-4}, \cdots, 10^2\}$. We observe that both AdaptiveEG+ and EG+ perform best with a fixed step size of $\gamma_k = 0.1$. For our method, we set the step size parameters as $(c_0, c_1) = (1, 1)$. In Figure 5, we present the trajectory plots of these algorithms, all initialized at $(w_1, w_2) = (1, 1)$. Our findings indicate that all algorithms eventually converge to the equilibrium $(0, 0)$, but the convergence of our method is significantly faster. This demonstrates the advantage of our step size strategy in solving challenging problems that satisfy only weak Minty conditions.

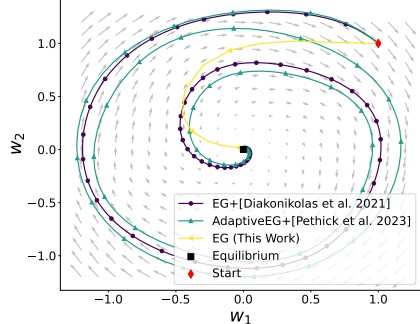

Figure 5: Trajectories of algorithms for solving problem (21).

# 5 Conclusion

This work extends the analysis of the EG method to a broader class of $\alpha$-symmetric $(L_0, L_1)$-Lipschitz operators. We establish new convergence guarantees for strongly monotone, monotone, and weak Minty settings, supported by a novel adaptive step size rule.

A limitation of our current analysis is that it focuses solely on root-finding problems and does not handle the constrained setup. However, the results included in this work advance the theoretical understanding of EG and open several promising directions, including extensions to constrained and stochastic settings, as well as the analysis of optimistic gradient methods under this relaxed assumption. Furthermore, another important research direction is the estimation of the unknown constants $\alpha, L_0, L_1$, which would pave the way for fully adaptive algorithms.

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

# Supplementary Material

The supplementary material is organized as follows. In Section A, we discuss papers that are closely related to our work. Section B presents several technical lemmas used in our theoretical analysis. Section C provides illustrative examples of $\alpha$-symmetric $(L_0, L_1)$-Lipschitz operators, while Section D contains the detailed proofs of the main convergence theorems presented in the paper. Next, in Section E, we discuss an equivalent formulation of the $\alpha$-symmetric $(L_0, L_1)$-Lipschitz condition. Finally, Section F offers additional details on the experimental setup and results.

## Contents

# A  Further Related Work

**Relaxing Lipschitzness.** The $(L_0, L_1)$-smoothness assumption was first introduced by Zhang et al. [2020b] and further developed in Zhang et al. [2020a], Chen et al. [2023]. More recently, improved convergence guarantees for optimization under this relaxed smoothness were obtained by Gorbunov et al. [2025], Vankov et al. [2025]. This assumption has also been used in the analysis of modern large-scale optimization algorithms, for example, Gluon for LLM training [Riabinin et al., 2025].

Beyond $(L_0, L_1)$-smoothness, several alternative notions or relaxations of smoothness have been proposed in the literature, including relative smoothness [Bauschke et al., 2017], glocal smoothness [Fox et al., 2025], and Hölder smoothness [Gorbunov et al., 2024], among others. The $(L_0, L_1)$-smoothness framework has recently been extended to the variational inequality setting by Vankov et al. [2024], while Xian et al. [2024] analyzed min–max problems under a similarly general smoothness perspective. A recent line of work by Loizou et al. [2021], Gorbunov et al. [2022b], Beznosikov et al. [2023] studies star-cocoercive operators, which could hold even when the operator is not Lipschitz.

**On Methods for Solving VIs.** A variety of methods have been developed to solve root-finding problems and, more generally, variational inequalities. The most basic approach is the gradient method and its stochastic counterpart [Loizou et al., 2021, Beznosikov et al., 2023]. More variants of the gradient method have been proposed in different settings, like the dissipative gradient method [Zheng et al., 2024] and distributed methods for VIs [Zhang et al., 2024]. However the gradient method fails to converge in simple monotone problems [Gidel et al., 2019].To obtain convergence guarantees in more relaxed settings (including monotone problems), Korpelevich [1977] introduced the EG method, which has since been extended to adaptive [Antonakopoulos et al., 2021], stochastic [Gorbunov et al., 2022a, Gidel et al., 2019, Mishchenko et al., 2020, Diakonikolas et al., 2021, Li et al., 2022], and decentralized variants [Beznosikov et al., 2022]. A key drawback of EG is that it requires two oracle calls per iteration. To address this, Popov [1980] proposed the Optimistic Gradient (OG) method, which uses only a single oracle call. Similar to EG, OG has been analyzed in stochastic settings [Hsieh et al., 2019, Gidel et al., 2019, Daskalakis et al., 2017, Choudhury et al., 2024, Böhm, 2023]. Beyond these methods, accelerated variants have also been developed, achieving faster convergence rates [Yoon and Ryu, 2021, Diakonikolas, 2020, Lee and Kim, 2021].

# B  Technical Lemmas

In this section, we present some technical lemmas, which will be used to prove the main results of the work in subsequent sections.

**Lemma B.1.** For $a, b \in \mathbb{R}^d$, we have

$$2 \langle a, b \rangle = \|a\|^2 + \|b\|^2 - \|a - b\|^2. \tag{22}$$

**Lemma B.2.** For $a, b \in \mathbb{R}^d$, we have

$$-\|a\|^2 \leq -\frac{1}{2}\|a + b\|^2 + \|b\|^2. \tag{23}$$

**Lemma B.3.** For positive numbers $a, b > 0$ we have

$$a + b \leq \sqrt{2}\sqrt{a^2 + b^2} \tag{24}$$

*Proof.* From AM-GM inequality on $a, b > 0$ we have

$$2ab \leq a^2 + b^2.$$

Adding $a^2 + b^2$ on both sides we have

$$(a + b)^2 \leq 2(a^2 + b^2).$$

Then the result follows by taking square root on both sides. $\square$

**Lemma B.4** (Cauchy-Schwarz Inequality)**.** For vectors $a, b \in \mathbb{R}^d$, we have

$$\langle a, b \rangle \leq \|a\|\|b\|. \tag{25}$$

**Lemma B.5.** Chen et al. [2023] Operator $F$ is $\alpha$-symmetric $(L_0, L_1)$-Lipschitz if and only if

$$\|F(x) - F(y)\| \leq \left( L_0 + L_1 \int_0^1 \|F(\theta x + (1 - \theta)y)\|^\alpha \, d\theta \right) \|x - y\| \qquad \forall x, y \in \mathbb{R}^d. \tag{26}$$

**Lemma B.6.** For a $2 \times 2$ symmetric matrix, the maximum eigenvalue is given by

$$\lambda_{\max} \left( \begin{bmatrix} a & b \\ b & d \end{bmatrix} \right) = \frac{(a + d) + \sqrt{(a - d)^2 + 4b^2}}{2} \tag{27}$$

where $a, b, d \in \mathbb{R}$.

*Proof.* Let $A$ be a symmetric $2 \times 2$ matrix given by

$$A = \begin{bmatrix} a & b \\ b & d \end{bmatrix}$$

where $a, b, d \in \mathbb{R}$. Since $A$ is symmetric, it has real eigenvalues. The eigenvalues of $A$ are the roots of its characteristic polynomial:

$$\det(A - \lambda I) = \det \left( \begin{bmatrix} a - \lambda & b \\ b & d - \lambda \end{bmatrix} \right) = (a - \lambda)(d - \lambda) - b^2.$$

Expanding the determinant, we obtain the characteristic equation:

$$\lambda^2 - (a + d)\lambda + (ad - b^2) = 0.$$

This is a quadratic equation in $\lambda$, and its solutions are:

$$\lambda = \frac{(a + d) \pm \sqrt{(a - d)^2 + 4b^2}}{2}.$$

Thus, the maximum eigenvalue is the larger of the two roots:

$$\lambda_{\max}(A) = \frac{(a + d) + \sqrt{(a - d)^2 + 4b^2}}{2}.$$

This completes the proof. $\square$

**Lemma B.7.** For the quadratic problem $\min_{w_1} \max_{w_2} \mathcal{L}(w_1, w_2) = \frac{1}{2}w_1^2 + w_1 w_2 - \frac{1}{2}w_2^2$, the Jacobian $\mathbf{J}(x)$ is given by

$$\mathbf{J}(x) = \begin{bmatrix} 1 & 1 \\ -1 & 1 \end{bmatrix}.$$

In this case we get $\|\mathbf{J}(x)\| = \sigma_{\max}(\mathbf{J}(x)) = \sqrt{2}$.

*Proof.* Note that

$$\mathbf{J}(x)^\top \mathbf{J}(x) = \begin{bmatrix} 2 & 0 \\ 0 & 2 \end{bmatrix} = 2\mathbf{I}.$$

which has maximum eigenvalue $\sqrt{2}$. Hence, $\|\mathbf{J}(x)\| = \sigma_{\max}(\mathbf{J}(x)) = \sqrt{\lambda_{\max}(\mathbf{J}(x)^\top \mathbf{J}(x))} = \sqrt{2}$. $\square$

**Lemma B.8** (Vankov et al. [2025] Example 2.2)**.** The function $f(x) = \frac{1}{p+1}\|x\|^{p+1}$, where $p > 1$ is $(\tau_0, \tau_1)$-smooth with arbitary $\tau_1 > 0$ and $\tau_0 = \left(\frac{p-1}{\tau_1}\right)^{p-1}$ i.e.

$$\|\nabla^2 f(x)\| \leq \left(\frac{p-1}{\tau_1}\right)^{p-1} + \tau_1 \|\nabla f(x)\| \tag{28}$$

for any $\tau_1 > 0$.

**Lemma B.9** (Equivalence of $L_q$ Norm)**.** For any vector $x \in \mathbb{R}^d$, and $0 < r < q$, we have

$$\|x\|_q \leq \|x\|_r \leq d^{\frac{1}{r} - \frac{1}{q}} \|x\|_q. \tag{29}$$

In particular, for any $a, b \in \mathbb{R}$ and $0 < r < q$ we have

$$(a^q + b^q)^{\frac{1}{q}} \leq (a^r + b^r)^{\frac{1}{r}} \tag{30}$$

and

$$(a^r + b^r)^{\frac{1}{r}} \leq 2^{\frac{1}{r} - \frac{1}{q}} (a^q + b^q)^{\frac{1}{q}} \tag{31}$$

*Proof.* The proof of (29) follows from Holder's Inequality. (30) and (31) follows from (29) with $d = 2$ and $x = (a, b)$. $\square$

# C    Further Examples of $\alpha$-Symmetric $(L_0, L_1)$-Lipschitz Operators

In this section, we provide further examples of operators that satisfy the $\alpha$-symmetric $(L_0, L_1)$-Lipschitz assumption. We first start with the details of Examples 2 and 3 from Section 2.2, then we provide three more examples for the min-max optimization problem and $N$-player game that satisfy the assumption.

**Example 2:** Here we consider the operator $F(x) = (u_1^2, u_2^2)$ for $x = (u_1, u_2)$. Then for $y = (v_1, v_2)$, we have

$$
\begin{aligned}
\|F(x) - F(y)\| &= \left\|\left(u_1^2 - v_1^2, u_2^2 - v_2^2\right)\right\| \\
&= \left(\left(u_1^2 - v_1^2\right)^2 + \left(u_2^2 - v_2^2\right)^2\right)^{1/2} \\
&= \left((u_1 - v_1)^2 (u_1 + v_1)^2 + (u_2 - v_2)^2 (u_2 + v_2)^2\right)^{1/2} \\
&\leq \left((u_1 - v_1)^4 + (u_2 - v_2)^4\right)^{1/4} \left((u_1 + v_1)^4 + (u_2 + v_2)^4\right)^{1/4} \\
&\leq \left((u_1 - v_1)^2 + (u_2 - v_2)^2\right)^{1/2} \left((u_1 + v_1)^4 + (u_2 + v_2)^4\right)^{1/4} \\
&= \left((u_1 + v_1)^4 + (u_2 + v_2)^4\right)^{1/4} \|x - y\| \\
&= 2\left(\left(\frac{u_1 + v_1}{2}\right)^4 + \left(\frac{u_2 + v_2}{2}\right)^4\right)^{1/4} \|x - y\| \\
&= 2\left\|\left(\left(\frac{u_1 + v_1}{2}\right)^2, \left(\frac{u_2 + v_2}{2}\right)^2\right)\right\|^{1/2} \|x - y\| \\
&= 2\left\|F\left(\frac{x + y}{2}\right)\right\|^{1/2} \|x - y\| \\
&\leq 2 \max_{\theta \in [0,1]} \|F(\theta x + (1 - \theta)y)\|^{1/2} \|x - y\|.
\end{aligned}
$$

Here, the first inequality follows from the Cauchy-Schwarz inequality. This completes the proof of $\frac{1}{2}$-symmetric-$(0, 2)$ Lipschitz property of $F$. Now, we consider the vectors $x = \alpha \mathbf{1}_2$ and $y = \mathbf{1}_2$ where $\mathbf{1}_2 = (1, 1)$. Then we have

$$
\begin{aligned}
\frac{\|F(x) - F(y)\|}{\|x - y\|} &= \frac{\sqrt{(\alpha^2 - 1)^2 + (\alpha^2 - 1)^2}}{\sqrt{(\alpha - 1)^2 + (\alpha - 1)^2}} \\
&= \frac{\sqrt{2(\alpha^2 - 1)^2}}{\sqrt{2(\alpha - 1)^2}} \\
&= \sqrt{\frac{(\alpha - 1)^2(\alpha + 1)^2}{(\alpha - 1)^2}} \\
&= |\alpha + 1|.
\end{aligned}
$$

Therefore,

$$
\begin{aligned}
\lim_{\alpha \to \infty} \frac{\|F(x) - F(y)\|}{\|x - y\|} &= \lim_{\alpha \to \infty} |\alpha + 1| \\
&= \infty
\end{aligned}
$$

This proves that $F$ is not Lipschitz.

**Example 3.** Consider the min-max optimization problem

$$\min_{w_1} \max_{w_2} \mathcal{L}(w_1, w_2) = \frac{1}{p+1}\|w_1\|^{p+1} + w_1^\top \mathbf{B} w_2 - \frac{1}{p+1}\|w_2\|^{p+1}. \tag{32}$$

Then the corresponding operator $F$ defined in (2) is 1-symmetric $\left(2\tau_0 + \|\mathbf{M}\|, 2^{\frac{2p^2-1}{p^2}}\tau_1\right)$-Lipschitz with any arbitary $\tau_1 > 0$ and $\tau_0 = \left(\frac{p-1}{\tau_1}\right)^{p-1}$ where $\mathbf{M} = \begin{bmatrix} 0 & \mathbf{B} \\ -\mathbf{B}^\top & 0 \end{bmatrix}$. Moreover, $F$ is not Lipschitz for any finite $L$.

*Proof.* For the simplicity of the proof, we define the function $f(w) := \frac{1}{p+1}\|w\|^{p+1}$. The following properties of this function will be used to prove our result.

**1. Gradient of $f(w)$:** The gradient of $f(w) = \frac{1}{p+1}\|w\|^{p+1} = \frac{1}{p+1}\left(\|w\|^2\right)^{\frac{p+1}{2}}$ is given by

$$\nabla f(w) = \frac{1}{p+1} \cdot \frac{p+1}{2}\left(\|w\|^2\right)^{\frac{p+1}{2}-1} \cdot \nabla\left(\|w\|^2\right) = \frac{1}{2}\left(\|w\|^2\right)^{\frac{p-1}{2}} \cdot 2w = \|w\|^{p-1}w. \tag{33}$$

Here, the first equality follows from chain rule.

**2. $f$ is $(\tau_0, \tau_1)$-smooth:** From Lemma B.8, we know $f$ is $(\tau_0, \tau_1)$-smooth. Choose arbitary $\tau_1 > 0$, then from Lemma B.8, $f$ satisfies

$$\|\nabla^2 f(w)\| \overset{(28)}{\leq} \tau_0 + \tau_1\|\nabla f(w)\| \tag{34}$$

where $\tau_0 = \left(\frac{p-1}{\tau_1}\right)^{p-1}$.

Now, we will get back to the original min-max problem. Note that, using the definition of $f(w)$, we can rewrite the objective function in (32) as

$$\mathcal{L}(w_1, w_2) = f(w_1) + w_1^\top \mathbf{B} w_2 - f(w_2). \tag{35}$$

Then the operator $F$ corresponding to the min-max problem (32) is given by

$$F(x) = \begin{bmatrix} \nabla_{w_1}\mathcal{L}(w_1, w_2) \\ -\nabla_{w_2}\mathcal{L}(w_1, w_2) \end{bmatrix} \overset{(35)}{=} \begin{bmatrix} \nabla f(w_1) + \mathbf{B} w_2 \\ \nabla f(w_2) - \mathbf{B}^\top w_1 \end{bmatrix} \overset{(33)}{=} \begin{bmatrix} \|w_1\|^{p-1}w_1 + \mathbf{B} w_1 \\ \|w_2\|^{p-1}w_2 - \mathbf{B}^\top w_2 \end{bmatrix}. \tag{36}$$

Moreover, the Jacobian matrix corresponding to this min-max problem (32) is

$$\mathbf{J}(x) = \begin{bmatrix} \nabla^2_{w_1 w_1}\mathcal{L}(w_1, w_2) & \nabla^2_{w_2 w_1}\mathcal{L}(w_1, w_2) \\ -\nabla^2_{w_1 w_2}\mathcal{L}(w_1, w_2) & -\nabla^2_{w_2 w_2}\mathcal{L}(w_1, w_2) \end{bmatrix} \overset{(35)}{=} \begin{bmatrix} \nabla^2 f(w_1) & \mathbf{B} \\ -\mathbf{B}^\top & \nabla^2 f(w_2) \end{bmatrix}. \tag{37}$$

Furthermore, note that, we can break the operator $F$ from (36) as $F(x) = H(x) + \mathbf{M}x$ with

$$H(x) = \begin{bmatrix} \nabla f(w_1) \\ \nabla f(w_2) \end{bmatrix} = \begin{bmatrix} \|w_1\|^{p-1}w_1 \\ \|w_2\|^{p-1}w_2 \end{bmatrix} \tag{38}$$

and $\mathbf{M} = \begin{bmatrix} 0 & \mathbf{B} \\ -\mathbf{B}^\top & 0 \end{bmatrix}$. Then the Jacobian $\mathbf{J}(x)$ of this min-max problem satisfies

$$\begin{aligned}
\|\mathbf{J}(x)\| &\overset{(37)}{=} \left\| \begin{bmatrix} \nabla^2 f(w_1) & \mathbf{B} \\ -\mathbf{B}^\top & \nabla^2 f(w_2) \end{bmatrix} \right\| \\
&= \left\| \begin{bmatrix} \nabla^2 f(w_1) & 0 \\ 0 & \nabla^2 f(w_2) \end{bmatrix} + \mathbf{M} \right\| \\
&\leq \left\| \begin{bmatrix} \nabla^2 f(w_1) & 0 \\ 0 & \nabla^2 f(w_2) \end{bmatrix} \right\| + \|\mathbf{M}\| \\
&\leq \max\left\{\|\nabla^2 f(w_1)\|, \|\nabla^2 f(w_2)\|\right\} + \|\mathbf{M}\| \\
&\leq \|\nabla^2 f(w_1)\| + \|\nabla^2 f(w_2)\| + \|\mathbf{M}\| \\
&\overset{(34)}{\leq} \tau_0 + \tau_1\|\nabla f(w_1)\| + \tau_0 + \tau_1\|\nabla f(w_2)\| + \|\mathbf{M}\| \\
&= 2\tau_0 + \|\mathbf{M}\| + \tau_1\left(\|\nabla f(w_1)\| + \|\nabla f(w_2)\|\right) \\
&\overset{(24)}{\leq} (2\tau_0 + \|\mathbf{M}\|) + \sqrt{2}\tau_1\sqrt{\|\nabla f(w_1)\|^2 + \|\nabla f(w_2)\|^2} \\
&\overset{(38)}{=} (2\tau_0 + \|\mathbf{M}\|) + \sqrt{2}\tau_1\|H(x)\|.
\end{aligned} \tag{39}$$

where $\tau_1 > 0$ is arbitary and $\tau_0 = \left(\frac{p-1}{\tau_1}\right)^{p-1}$. Above, the second inequality follows from the fact that the maximum eigenvalue of a block diagonal matrix is less than or equal to the maximum eigenvalues of it's diagonal

matrices. Now, we want to find a bound on $\|H(x)\|$ in terms of $\|F(x)\|$. Towards that end, we will first find the relation between $\|H(x)\|$ and $\langle F(x), x \rangle$. Note that

$$
\begin{aligned}
\langle F(x), x \rangle &= \langle H(x), x \rangle + \langle \mathbf{M}x, x \rangle \\
&= \langle H(x), x \rangle + \left\langle \begin{bmatrix} \mathbf{B}w_2 \\ -\mathbf{B}^\top w_1 \end{bmatrix}, \begin{bmatrix} w_1 \\ w_2 \end{bmatrix} \right\rangle \\
&= \langle H(x), x \rangle \\
&\overset{(38)}{=} \left\langle \begin{bmatrix} \|w_1\|^{p-1} w_1 \\ \|w_2\|^{p-1} w_2 \end{bmatrix}, \begin{bmatrix} w_1 \\ w_2 \end{bmatrix} \right\rangle \\
&= \|w_1\|^{p+1} + \|w_2\|^{p+1}
\end{aligned}
\tag{40}
$$

and

$$
\begin{aligned}
\|H(x)\| &\overset{(38)}{=} \left\| \begin{bmatrix} \|w_1\|^{p-1} w_1 \\ \|w_2\|^{p-1} w_2 \end{bmatrix} \right\| \\
&= \left( \left( \|w_1\|^{p-1} \right)^2 \|w_1\|^2 + \left( \|w_2\|^{p-1} \right)^2 \|w_2\|^2 \right)^{\frac{1}{2}} \\
&= \left( \|w_1\|^{2p} + \|w_2\|^{2p} \right)^{\frac{1}{2}}
\end{aligned}
\tag{41}
$$

Then note that

$$
\begin{aligned}
\|H(x)\| &\overset{(41)}{=} \left( \|w_1\|^{2p} + \|w_2\|^{2p} \right)^{\frac{1}{2}} \\
&= \left( \left( \|w_1\|^{2p} + \|w_2\|^{2p} \right)^{\frac{1}{2p}} \right)^{p} \\
&\overset{(30)}{\leq} \left( \left( \|w_1\|^{p+1} + \|w_2\|^{p+1} \right)^{\frac{1}{p+1}} \right)^{p} \\
&= \left( \|w_1\|^{p+1} + \|w_2\|^{p+1} \right)^{\frac{p}{p+1}} \\
&\overset{(40)}{=} \langle F(x), x \rangle^{\frac{p}{p+1}} \\
&\overset{(25)}{\leq} \|F(x)\|^{\frac{p}{p+1}} \|x\|^{\frac{p}{p+1}} \\
&= \|F(x)\|^{\frac{p}{p+1}} \left( \left( \|w_1\|^2 + \|w_2\|^2 \right)^{\frac{1}{2}} \right)^{\frac{p}{p+1}} \\
&\overset{(31)}{\leq} \|F(x)\|^{\frac{p}{p+1}} \left( 2^{\frac{1}{2}-\frac{1}{2p}} \left( \|w_1\|^{2p} + \|w_2\|^{2p} \right)^{\frac{1}{2p}} \right)^{\frac{p}{p+1}} \\
&= 2^{\frac{p-1}{2p}} \|F(x)\|^{\frac{p}{p+1}} \left( \left( \|w_1\|^{2p} + \|w_2\|^{2p} \right)^{\frac{1}{2p}} \right)^{\frac{p}{p+1}} \\
&= 2^{\frac{p-1}{2p}} \|F(x)\|^{\frac{p}{p+1}} \left( \left( \|w_1\|^{2p} + \|w_2\|^{2p} \right)^{\frac{1}{2}} \right)^{\frac{1}{p+1}} \\
&\overset{(41)}{=} 2^{\frac{p-1}{2p}} \|F(x)\|^{\frac{p}{p+1}} \|H(x)\|^{\frac{1}{p+1}}
\end{aligned}
$$

Therefore, dividing both sides by $\|H(x)\|^{\frac{1}{p+1}}$ we get

$$
\|H(x)\|^{\frac{p}{p+1}} \leq 2^{\frac{p-1}{2p}} \|F(x)\|^{\frac{p}{p+1}}.
$$

Thus raising both sides to the power $\frac{p+1}{p}$ we get

$$
\|H(x)\| \leq 2^{\frac{p^2-1}{2p^2}} \|F(x)\|.
\tag{42}
$$

Combining this bound on $\|H(x)\|$ with (39) we get

$$
\begin{aligned}
\|\mathbf{J}(x)\| &\overset{(39)}{\leq} (2\tau_0 + \|\mathbf{M}\|) + 2^{\frac{1}{2}} \tau_1 \|H(x)\| \\
&\overset{(42)}{\leq} (2\tau_0 + \|\mathbf{M}\|) + 2^{\frac{1}{2} + \frac{p^2-1}{2p^2}} \tau_1 \|F(x)\| \\
&= (2\tau_0 + \|\mathbf{M}\|) + 2^{\frac{2p^2-1}{2p^2}} \tau_1 \|F(x)\|.
\end{aligned}
$$

Thus, the result follows using Theorem 2.1.

Next we show $F$ is not Lipschitz. Consider, $x = (tw_1, 0)$ and $y = (0, 0)$ with $\|w_1\| = 1$. Then we get

$$\frac{\|F(x) - F(y)\|}{\|x - y\|} = \frac{1}{p+1}\frac{t^{p+1}}{t} = \frac{t^p}{p+1}.$$

As we take $t \to \infty$, we get $\frac{\|F(x)-F(y)\|}{\|x-y\|} \to \infty$. Hence, there doesn't exist any finite bound on $\frac{\|F(x)-F(y)\|}{\|x-y\|}$ or $F$ is not $L$-Lipschitz. $\qquad\square$

**Example 4:** Consider the min-max problem

$$\min_{w_1} \max_{w_2} \mathcal{L}(w_1, w_2) = \frac{1}{p+1}\|w_1\|^{p+1} - \frac{1}{p+1}\|w_2\|^{p+1}.$$

Then the corresponding operator $F$ defined in (2) is 1-symmetric $(L_0, L_1)$-Lipschitz where $L_1 = \sqrt{2}\tau_1$ and $L_0 = 2\left(\frac{p-1}{\tau_1}\right)^{p-1}$ for any $\tau_1 > 0$. Moreover, $F$ is not $L$-Lipschitz for any finite $L$.

*Proof.* Define $f(w) = \frac{1}{p+1}\|w\|^{p+1}$. From Lemma B.8, we know $f$ is $(\tau_0, \tau_1)$-smooth. Choose arbitary $\tau_1 > 0$, then from Lemma B.8, $f$ satisfies

$$\|\nabla^2 f(w)\| \overset{(28)}{\leq} \tau_0 + \tau_1\|\nabla f(w)\| \tag{43}$$

where $\tau_0 = \left(\frac{p-1}{\tau_1}\right)^{p-1}$. Then we have $\mathcal{L}(w_1, w_2) = f(w_1) - f(w_2)$ and the corresponding operator and Jacobian are given by

$$F(x) = \begin{bmatrix} \nabla f(w_1) \\ \nabla f(w_2) \end{bmatrix} \quad \text{and} \quad \mathbf{J}(x) = \begin{bmatrix} \nabla^2 f(w_1) & 0 \\ 0 & \nabla^2 f(w_2) \end{bmatrix},$$

respectively. Note that the Jacobian matrix $\mathbf{J}(x)$ is block diagonal. Hence, the maximum eigenvalue of $\mathbf{J}(x)$ is less than the maxiumum eigenvalue of both $\nabla^2 f(w_1)$ and $\nabla^2 f(w_2)$. Therefore, we get

$$\begin{aligned}
\|\mathbf{J}(x)\| \quad &\leq \quad \max\left\{\|\nabla^2 f(w_1)\|, \|\nabla^2 f(w_2)\|\right\} \\
&\leq \quad \|\nabla^2 f(w_1)\| + \|\nabla^2 f(w_2)\| \\
&\overset{(43)}{\leq} \quad \tau_0 + \tau_1\|\nabla f(w_1)\| + \tau_0 + \tau_1\|\nabla f(w_2)\| \\
&= \quad 2\tau_0 + \tau_1\left(\|\nabla f(w_1)\| + \|\nabla f(w_2)\|\right) \\
&\overset{(24)}{\leq} \quad 2\tau_0 + \sqrt{2}\tau_1\sqrt{\|\nabla f(w_1)\|^2 + \|\nabla f(w_2)\|^2} \\
&= \quad 2\tau_0 + \sqrt{2}\tau_1\|F(x)\|.
\end{aligned}$$

Then $F$ is 1-symmetric $(L_0, L_1)$-Lipschitz from Theorem 2.1 with $L_1 = \sqrt{2}\tau_1$ and $L_0 = 2\tau_0 = 2\left(\frac{p-1}{\tau_1}\right)^{p-1}$ for any $\tau_1 > 0$.

Next we show $F$ is not Lipschitz. Consider, $x = (tw_1, 0)$ and $y = (0, 0)$ with $\|w_1\| = 1$. Then we get

$$\frac{\|F(x) - F(y)\|}{\|x - y\|} = \frac{1}{p+1}\frac{t^{p+1}}{t} = \frac{t^p}{p+1}.$$

As we take $t \to \infty$, we get $\frac{\|F(x)-F(y)\|}{\|x-y\|} \to \infty$. Hence, there doesn't exist any finite bound on $\frac{\|F(x)-F(y)\|}{\|x-y\|}$. Therefore, $F$ is not $L$-Lipschitz.

$\qquad\square$

**Example 5:** Min-max optimization problems can be studied using operators. Here we consider one such example, the bilinearly coupled min-max optimization Chambolle and Pock [2016] problem

$$\min_{\|w_1\| \leq R} \max_{\|w_2\| \leq R} \mathcal{L}(w_1, w_2) := f(w_1) + w_1^\top \mathbf{B} w_2 - g(w_2)$$

for matrix $\mathbf{B} \in \mathbb{R}^{d \times d}$ and functions $f, g : \mathbb{R}^d \to \mathbb{R}$. The associated operator for this problem is given by $F(x) = H(x) + \mathbf{M}x$ where

$$H(x) = \begin{bmatrix} \nabla f(w_1) \\ \nabla g(w_2) \end{bmatrix} \quad \text{and} \quad \mathbf{M} = \begin{bmatrix} 0 & \mathbf{B} \\ -\mathbf{B}^\top & 0 \end{bmatrix}.$$

If $f, g$ are individually $(L_0, L_1)$-smooth Zhang et al. [2020b], we show that

$$\|F(x) - F(y)\| \quad \leq \quad \left(2L_0 + (1 + 2L_1 R)\|\mathbf{M}\| + \sqrt{2}L_1\|F(x)\|\right)\|x - y\|.$$

Thus, $F$ is 1-symmetric $\left(2L_0 + (1 + 2L_1R)\|\mathbf{M}\|, \sqrt{2}L_1\right)$-Lipschitz. Consider the min-max problem

$$\min_{\|w_1\| \leq R} \max_{\|w_2\| \leq R} \mathcal{L}(w_1, w_2) := f(w_1) + w_1^\top \mathbf{B} w_2 - g(w_2).$$

where $f, g$ are $(L_0, L_1)$-smooth. Then for $x = (w_1, w_2)$ we have

$$F(x) = \begin{pmatrix} \nabla f(w_1) \\ \nabla g(w_2) \end{pmatrix} + \begin{pmatrix} 0 & \mathbf{B} \\ -\mathbf{B}^\top & 0 \end{pmatrix} \begin{pmatrix} w_1 \\ w_2 \end{pmatrix} = H(x) + \mathbf{M}x$$

where $\mathbf{M}$ is a matrix and $H$ is an operator. Now note that for $x = (w_1, w_2)$ and $y = (v_1, v_2)$ we get

$$
\begin{aligned}
\|H(x) - H(y)\| &= \left\| \begin{bmatrix} \nabla f(w_1) - \nabla f(v_1) \\ \nabla g(w_2) - \nabla g(v_2) \end{bmatrix} \right\| \\
&= \left( \|\nabla f(w_1) - \nabla f(v_1)\|^2 + \|g(w_2) - g(v_2)\|^2 \right)^{1/2} \\
&\leq \left( (L_0 + L_1\|\nabla f(w_1)\|)^2 \|w_1 - v_1\|^2 + (L_0 + L_1\|\nabla g(w_2)\|)^2 \|w_2 - v_2\|^2 \right)^{1/2} \\
&\leq \left( (L_0 + L_1\|\nabla f(w_1)\|)^4 + (L_0 + L_1\|\nabla g(w_2)\|)^4 \right)^{1/4} \left( \|w_1 - v_1\|^4 + \|w_2 - v_2\|^4 \right)^{1/4} \\
&\leq \left( (L_0 + L_1\|\nabla f(w_1)\|)^2 + (L_0 + L_1\|\nabla g(w_2)\|)^2 \right)^{1/2} \left( \|w_1 - v_1\|^2 + \|w_2 - v_2\|^2 \right)^{1/2} \\
&\leq \left( 4L_0^2 + 2L_1^2 \left( \|\nabla f(w_1)\|^2 + \|\nabla g(w_2)\|^2 \right) \right)^{1/2} \left( \|w_1 - v_1\|^2 + \|w_2 - v_2\|^2 \right)^{1/2} \\
&= \left( 4L_0^2 + 2L_1^2\|H(x)\|^2 \right)^{1/2} \|x - y\| \\
&\leq \left( 2L_0 + \sqrt{2}L_1\|H(x)\| \right) \|x - y\|
\end{aligned}
$$

Therefore, using the above inequality, we derive

$$
\begin{aligned}
\|F(x) - F(y)\| &\leq \|H(x) - H(y)\| + \|\mathbf{M}x - \mathbf{M}y\| \\
&\leq \left( 2L_0 + \sqrt{2}L_1\|H(x)\| \right) \|x - y\| + \|\mathbf{M}\|\|x - y\| \\
&\leq \left( 2L_0 + \|\mathbf{M}\| + \sqrt{2}L_1\|H(x)\| \right) \|x - y\|
\end{aligned}
$$

Now using $\|H(x)\| = \|H(x) + \mathbf{M}x - \mathbf{M}x\| \leq \|F(x)\| + \|\mathbf{M}\|\|x\| \leq \|F(x)\| + \sqrt{2}R\|\mathbf{M}\|$ we get

$$\|F(x) - F(y)\| \leq \left( 2L_0 + (1 + 2L_1R)\|\mathbf{M}\| + \sqrt{2}L_1\|F(x)\| \right) \|x - y\|.$$

**Example 6:** In this example, we consider an $N$-player game Balduzzi et al. [2018], Loizou et al. [2021], Yoon et al. [2025], where each player $i \in [N]$ selects an action $w_i \in \mathbb{R}^{d_i}$, and the joint action vector of all players is denoted as $x = (w_1, w_2, \cdots, w_N) \in \mathbb{R}^{d_1 + \cdots d_N}$. Each player $i$ aims to minimize their loss function $f_i$ for their action $w_i$. The objective is to find an equilibrium $x_* = (w_{1*}, w_{2*}, \cdots, w_{N*})$ such that

$$w_{i*} = \operatorname{argmin}_{w_i \in \mathbb{R}^{d_i}} f_i(w_i, w_{-i*}).$$

Here we abuse the notation to denote $w_{-i} = (w_1, \cdots, w_{i-1}, w_{i+1}, \cdots, w_N)$ and $f_i(w_i, w_{-i}) = f_i(w_1, \cdots, w_N)$. When the functions $f_i$ are convex, this equilibrium corresponds to solving $F(x_*) = 0$, where the operator $F$ is defined as

$$F(x) = (\nabla_1 f_1(x), \nabla_2 f_2(x), \ldots, \nabla_N f_N(x)).$$

In case each of these partial gradients $\nabla_i f_i$ are $(L_0, L_1)$-Lipschitz i.e.

$$\|\nabla_i f_i(x) - \nabla_i f_i(y)\| \leq (L_0 + L_1\|\nabla_i f_i(x)\|)\|x - y\|$$

then we obtain

$$
\begin{aligned}
\|F(x) - F(y)\|^2 &= \sum_{i=1}^N \|\nabla_i f_i(x) - \nabla_i f_i(y)\|^2 \\
&\leq \sum_{i=1}^N (L_0 + L_1\|\nabla_i f_i(x)\|)^2 \|x - y\|^2 \\
&\leq \|x - y\|^2 \sum_{i=1}^N (2L_0^2 + 2L_1^2\|\nabla_i f_i(x)\|^2) \\
&= \|x - y\|^2 (2NL_0^2 + 2L_1^2\|F(x)\|^2) \\
&\leq \|x - y\|^2 (\sqrt{2N}L_0 + \sqrt{2}L_1\|F(x)\|)^2.
\end{aligned}
$$

This completes the proof.

# D Convergence Analysis

In this section, we present the missing proofs from Section 3. We start with the proof of Proposition 3.1 and then provide the results related to strongly monotone, monotone and weak Minty problems.

## D.1 Proof of Proposition 3.1

**Proposition D.1.** Suppose $F$ is $\alpha$-symmetric $(L_0, L_1)$-Lipschitz operator. Then, for $\alpha = 1$

$$\|F(x) - F(y)\| \leq (L_0 + L_1\|F(x)\|)\exp\left(L_1\|x - y\|\right)\|x - y\|,$$

and for $\alpha \in (0, 1)$ we have

$$\|F(x) - F(y)\| \leq \left(K_0 + K_1\|F(x)\|^\alpha + K_2\|x - y\|^{\alpha/1-\alpha}\right)\|x - y\|$$

where $K_0 = L_0(2^{\alpha^2/1-\alpha} + 1)$, $K_1 = L_1 \cdot 2^{\alpha^2/1-\alpha}$ and $K_2 = L_1^{1/1-\alpha} \cdot 2^{\alpha^2/1-\alpha} \cdot 3^\alpha(1 - \alpha)^{\alpha/1-\alpha}$.

*Proof.* For proving this theorem, we follow the proof technique similar to Chen et al. [2023]. We start with $\alpha = 1$ case. Let $x, y \in \mathbb{R}^d$ and define $x_\theta := \theta x + (1 - \theta)y$. Since $F$ is 1-symmetric $(L_0, L_1)$-Lipschitz , we have for all $\theta \in [0, 1]$,

$$\|F(x_\theta) - F(y)\| \overset{(26)}{\leq} \left(L_0 + L_1 \int_0^1 \|F(x_{\theta\tau})\|d\tau\right)\|x_\theta - y\|.$$

Note that
$$x_{\theta\tau} = \tau x_\theta + (1 - \tau)y = \tau(\theta x + (1 - \theta)y) + (1 - \tau)y = \theta\tau x + (1 - \theta\tau)y.$$

Let us define a function
$$H(\theta) := L_0\theta + L_1 \int_0^\theta \|F(x_u)\|du.$$

Then, note that $H'(\theta) = L_0 + L_1\|F(x_\theta)\|$. Moreover, we have

$$
\begin{aligned}
\|F(x_\theta) - F(y)\| &\leq \left(L_0 + L_1 \int_0^1 \|F(x_{\theta\tau})\|d\tau\right)\|x_\theta - y\| \\
&= \left(L_0 + L_1 \int_0^1 \|F(x_{\theta\tau})\|d\tau\right)\|\theta x + (1 - \theta)y - y\| \\
&= \left(L_0 + L_1 \int_0^1 \|F(x_{\theta\tau})\|d\tau\right)\|\theta x - \theta y\| \\
&= \left(L_0\theta + L_1 \int_0^1 \|F(x_{\theta\tau})\|\theta d\tau\right)\|x - y\| \\
&= \left(L_0\theta + L_1 \int_0^1 \|F(\theta\tau x + (1 - \theta\tau)y)\|\theta d\tau\right)\|x - y\| \\
&= \left(L_0\theta + L_1 \int_0^\theta \|F(ux + (1 - u)y)\|du\right)\|x - y\| \\
&= \left(L_0\theta + L_1 \int_0^\theta \|F(x_u)\|du\right)\|x - y\| \\
&= H(\theta)\|x - y\|.
\end{aligned}
$$

Therefore we obtain

$$
\begin{aligned}
H'(\theta) &= L_0 + L_1\|F(x_\theta)\| \\
&\leq L_0 + L_1\left(\|F(x_\theta) - F(y)\| + \|F(y)\|\right) \\
&\leq L_0 + L_1\left(H(\theta)\|x - y\| + \|F(y)\|\right) \\
&= aH(\theta) + b,
\end{aligned}
$$

where $a = L_1\|x - y\|, b = L_0 + L_1\|F(y)\|$. Then we integrate both sides for $\theta \in [0, \theta']$ to get
$$H(\theta') \leq \frac{b}{a}(e^{a\theta'} - 1).$$

Here, we set $\theta' = 1$ to obtain

$$
\begin{aligned}
H(1) &\leq \frac{b}{a}(e^a - 1) \\
&= \frac{L_0 + L_1\|F(y)\|}{L_1\|x - y\|}(e^{L_1\|x-y\|} - 1).
\end{aligned}
$$

Now, put this back into the original inequality

$$
\begin{aligned}
\|F(x) - F(y)\| &\leq H(1)\|x - y\| \\
&\leq (L_0 + L_1\|F(y)\|) \cdot \frac{e^{L_1\|x-y\|} - 1}{L_1} \\
&= \left(\frac{L_0}{L_1} + \|F(y)\|\right)(e^{L_1\|x-y\|} - 1).
\end{aligned}
$$

Finally, using the inequality $e^z - 1 \leq ze^z$ for $z \geq 0$, we get

$$
\begin{aligned}
\|F(x) - F(y)\| &\leq \left(\frac{L_0}{L_1} + \|F(y)\|\right)L_1\|x - y\|e^{L_1\|x-y\|} \\
&\leq (L_0 + L_1\|F(y)\|)e^{L_1\|x-y\|}\|x - y\|
\end{aligned}
$$

This completes the proof for $\alpha = 1$. The proof for $\alpha \in (0, 1)$ follows similarly from Chen et al. [2023]. $\qquad\square$

## D.2 Convergence Guarantees for Strongly Monotone Operators

**Lemma D.2.** Suppose $F$ is $\mu$-strongly monotone operator. Then Extragradient method with step size $\gamma_k = \omega_k$ satisfy

$$
\|x_{k+1} - x_*\|^2 \leq (1 - \gamma_k\mu)\|x_k - x_*\|^2 - \gamma_k^2(1 - 2\gamma_k\mu)\|F(x_k)\|^2 + \gamma_k^2\|F(x_k) - F(\hat{x}_k)\|^2. \quad (44)
$$

*Proof.* From the update step of the Extragradient method, we obtain

$$
\begin{aligned}
\|x_{k+1} - x_*\|^2 &= \|x_k - \gamma_k F(\hat{x}_k) - x_*\|^2 \\
&= \|x_k - x_*\|^2 - 2\gamma_k \langle F(\hat{x}_k), x_k - x_*\rangle + \gamma_k^2\|F(\hat{x}_k)\|^2 \\
&= \|x_k - x_*\|^2 - 2\gamma_k \langle F(\hat{x}_k), \hat{x}_k - x_*\rangle - 2\gamma_k \langle F(\hat{x}_k), x_k - \hat{x}_k\rangle + \gamma_k^2\|F(\hat{x}_k)\|^2 \\
&\overset{(9)}{\leq} \|x_k - x_*\|^2 - 2\gamma_k\mu\|\hat{x}_k - x_*\|^2 - 2\gamma_k \langle F(\hat{x}_k), x_k - \hat{x}_k\rangle + \gamma_k^2\|F(\hat{x}_k)\|^2 \\
&\overset{(23)}{\leq} \|x_k - x_*\|^2 - \gamma_k\mu\|x_k - x_*\|^2 + 2\gamma_k\mu\|x_k - \hat{x}_k\|^2 - 2\gamma_k \langle F(\hat{x}_k), x_k - \hat{x}_k\rangle \\
&\quad + \gamma_k^2\|F(\hat{x}_k)\|^2 \\
&= (1 - \gamma_k\mu)\|x_k - x_*\|^2 + 2\gamma_k\mu\|x_k - \hat{x}_k\|^2 - 2\gamma_k \langle F(\hat{x}_k), x_k - \hat{x}_k\rangle + \gamma_k^2\|F(\hat{x}_k)\|^2 \\
&= (1 - \gamma_k\mu)\|x_k - x_*\|^2 + 2\gamma_k^3\mu\|F(x_k)\|^2 - 2\gamma_k^2 \langle F(\hat{x}_k), F(x_k)\rangle + \gamma_k^2\|F(\hat{x}_k)\|^2 \\
&\overset{(22)}{=} (1 - \gamma_k\mu)\|x_k - x_*\|^2 + 2\gamma_k^3\mu\|F(x_k)\|^2 \\
&\quad -\gamma_k^2\left(\|F(\hat{x}_k)\|^2 + \|F(x_k)\|^2 - \|F(x_k) - F(\hat{x}_k)\|^2\right) + \gamma_k^2\|F(\hat{x}_k)\|^2 \\
&= (1 - \gamma_k\mu)\|x_k - x_*\|^2 - \gamma_k^2(1 - 2\gamma_k\mu)\|F(x_k)\|^2 + \gamma_k^2\|F(x_k) - F(\hat{x}_k)\|^2.
\end{aligned}
$$

$\qquad\square$

### D.2.1 Proof of Theorem 3.2

**Theorem D.3.** Suppose $F$ is $\mu$-strongly monotone and 1-symmetric $(L_0, L_1)$-Lipschitz operator. Then Extragradient method with step size $\gamma_k = \omega_k = \frac{\nu}{L_0 + L_1\|F(x_k)\|}$ satisfy

$$
\|x_{k+1} - x_*\|^2 \leq \left(1 - \frac{\nu\mu}{L_0\left(1 + L_1\exp\left(L_1\|x_0 - x_*\|\right)\|x_0 - x_*\|\right)}\right)^{k+1}\|x_0 - x_*\|^2
$$

where $\nu > 0$ root of $1 - 2\nu - \nu^2\exp 2\nu = 0$.

*Proof.* From the update rule of Extragradient and using $\mu$-strong monotonicty, we obtain

$$
\begin{aligned}
\|x_{k+1} - x_*\|^2 &\overset{(44)}{\leq} (1 - \gamma_k\mu)\|x_k - x_*\|^2 - \gamma_k^2(1 - 2\gamma_k\mu)\|F(x_k)\|^2 + \gamma_k^2\|F(x_k) - F(\hat{x}_k)\|^2 \\
&\overset{(13)}{\leq} (1 - \gamma_k\mu)\|x_k - x_*\|^2 - \gamma_k^2(1 - 2\gamma_k\mu)\|F(x_k)\|^2 \\
&\quad + \gamma_k^2(L_0 + L_1\|F(x_k)\|)^2\exp\left(2L_1\|x_k - \hat{x}_k\|\right)\|x_k - \hat{x}_k\|^2 \\
&= (1 - \gamma_k\mu)\|x_k - x_*\|^2 \\
&\quad -\gamma_k^2\left(1 - 2\gamma_k\mu - \gamma_k^2(L_0 + L_1\|F(x_k)\|)^2\exp\left(2\gamma_k L_1\|F(x_k)\|\right)\right)\|F(x_k)\|^2. \quad (45)
\end{aligned}
$$

Now we set $\gamma_k = \frac{\nu}{L_0 + L_1 \|F(x_k)\|}$. Then we want to choose $\nu$ such that

$$1 - \frac{2\nu\mu}{L_0 + L_1 \|F(x_k)\|} - \nu^2 \exp 2\nu \geq 0.$$

Note that $\mu \leq L_0 + L_1 \|F(x_k)\|$ for any $x_k$. Therefore, we get $1 - \frac{2\nu\mu}{L_0 + L_1 \|F(x_k)\|} - \nu^2 \exp 2\nu \geq 1 - 2\nu - \nu^2 \exp 2\nu$ and it is enough to choose $\nu$ such that

$$1 - 2\nu - \nu^2 \exp 2\nu \geq 0.$$

This inequality holds for any $\nu \leq 0.22$. Hence, for this choice of $\nu$ we get the following inequality from (45).

$$\|x_{k+1} - x_*\|^2 \quad \leq \quad (1 - \gamma_k \mu) \|x_k - x_*\|^2. \tag{46}$$

This proves that the distance of the iterates $x_k$ from $x_*$ are bounded by $\|x_0 - x_*\|$. Now note that using (13) with $x = x_k, y = x_*$ with $\|x_k - x_*\| \leq \|x_0 - x_*\|$ we get

$$
\begin{aligned}
\|F(x_k)\| &\overset{(13)}{\leq} L_0 \exp\left(L_1 \|x_k - x_*\|\right) \|x_k - x_*\| \\
&\overset{(46)}{\leq} L_0 \exp\left(L_1 \|x_0 - x_*\|\right) \|x_0 - x_*\|.
\end{aligned}
\tag{47}
$$

Therefore, we have the following lower bound on the step size

$$
\begin{aligned}
\gamma_k &= \frac{\nu}{L_0 + L_1 \|F(x_k)\|} \\
&\overset{(47)}{\geq} \frac{\nu}{L_0 \left(1 + L_1 \exp\left(L_1 \|x_0 - x_*\|\right) \|x_0 - x_*\|\right)}.
\end{aligned}
\tag{48}
$$

Hence, we get

$$
\begin{aligned}
\|x_{k+1} - x_*\|^2 &\overset{(46),(48)}{\leq} \left(1 - \frac{\nu\mu}{L_0 \left[1 + L_1 \exp\left(L_1 \|x_0 - x_*\|\right) \|x_0 - x_*\|\right]}\right) \|x_k - x_*\|^2 \\
&\leq \left(1 - \frac{\nu\mu}{L_0 \left[1 + L_1 \exp\left(L_1 \|x_0 - x_*\|\right) \|x_0 - x_*\|\right]}\right)^{k+1} \|x_0 - x_*\|^2.
\end{aligned}
$$

$\square$

### D.2.2 Proof of Corollary 3.3

**Corollary D.4.** Suppose $F$ is $\mu$-strongly monotone and 1-symmetric $(L_0, L_1)$-Lipschitz operator. Then Extragradient with step size $\gamma_k = \omega_k = \frac{\nu}{L_0 + L_1 \|F(x_k)\|}$ satisfy $\|x_{K+1} - x_*\|^2 \leq \varepsilon$ after

$$K = \frac{2L_0}{\nu\mu} \log\left(\frac{\|x_0 - x_*\|^2}{\varepsilon}\right) + \frac{1}{\gamma\mu} \log\left(\frac{2L_1 \|x_0 - x_*\|^2}{\gamma^2 L_0}\right)$$

many iterations, where $\nu > 0$ satisfy $1 - 4\nu - 2\nu^2 \exp 2\nu = 0$ and

$$\gamma := \frac{\nu}{L_0 \left(1 + L_1 \exp\left(L_1 \|x_0 - x_*\|\right) \|x_0 - x_*\|\right)}.$$

*Proof.* We set $\gamma_k = \frac{\nu}{L_0 + L_1 \|F(x_k)\|}$ and we choose $\nu \in (0, 1)$ such that $1 - 4\nu - 2\nu^2 \exp 2\nu = 0$. Then we have

$$
\begin{aligned}
1 - 2\gamma_k \mu - \gamma_k^2 (L_0 + L_1 \|F(x_k)\|)^2 \exp\left(2\gamma_k L_1 \|F(x_k)\|\right) &\geq 1 - \frac{2\nu\mu}{L_0 + L_1 \|F(x_k)\|} - \nu^2 \exp 2\nu \\
&\geq 1 - 2\nu - \nu^2 \exp 2\nu \\
&= \frac{1}{2}.
\end{aligned}
\tag{49}
$$

Therefore, from (45) and (49) we get

$$\|x_{k+1} - x_*\|^2 \quad \leq \quad (1 - \gamma_k \mu) \|x_k - x_*\|^2 - \frac{\gamma_k^2}{2} \|F(x_k)\|^2. \tag{50}$$

In (48), we found that a lower bound of $\gamma_k$ is

$$\gamma_k \geq \gamma := \frac{\nu}{L_0 \left(1 + L_1 \exp\left(L_1 \|x_0 - x_*\|\right) \|x_0 - x_*\|\right)}.$$

Then from (50) we get

$$\frac{\gamma^2}{2} \|F(x_k)\|^2 \leq (1 - \gamma\mu) \|x_k - x_*\|^2 \leq (1 - \gamma\mu)^{k+1} \|x_0 - x_*\|^2 \tag{51}$$

This can be rearranged to write

$$\|F(x_k)\|^2 \le \frac{2(1-\gamma\mu)^{k+1}}{\gamma^2}\|x_0 - x_*\|^2.$$

This implies $\|F(x_k)\|^2 \le \frac{L_0}{L_1}$ after

$$K' = \frac{1}{\gamma\mu}\log\left(\frac{2L_1\|x_0 - x_*\|^2}{\gamma^2 L_0}\right) \tag{52}$$

many iterations. Hence for $k \ge K'$ we have

$$\begin{aligned}
\gamma_k &= \frac{\nu}{L_0 + L_1\|F(x_k)\|} \\
&\ge \frac{\nu}{2L_0}.
\end{aligned}$$

In the last inequality we used $\|F(x_k)\|^2 \le \frac{L_0}{L_1}$ for $k \ge K'$. Therefore for $k \ge K'$ we obtain

$$\begin{aligned}
\|x_{k+1} - x_*\|^2 &\le \left(1 - \frac{\nu\mu}{2L_0}\right)\|x_k - x_*\|^2 \\
&\le \left(1 - \frac{\nu\mu}{2L_0}\right)^{k+1-K'}\|x_{K'} - x_*\|^2 \\
&\le \left(1 - \frac{\nu\mu}{2L_0}\right)^{k+1-K'}\|x_0 - x_*\|^2
\end{aligned}$$

Thus we conclude that, $\|x_{K+1} - x_*\|^2 \le \varepsilon$ after atmost

$$K = \frac{2L_0}{\nu\mu}\log\left(\frac{\|x_0 - x_*\|^2}{\varepsilon}\right) + K'$$

many iterations. □

### D.2.3 Proof of Theorem 3.4

**Theorem D.5.** Suppose $F$ is $\mu$-strongly monotone and $\alpha$-symmetric $(L_0, L_1)$-Lipschitz operator with $\alpha \in (0, 1)$. Then Extragradient method with $\gamma_k = \omega_k = \frac{\nu}{2K_0 + \left(2K_1 + 2^{1-\alpha}K_2^{1-\alpha}\right)\|F(x_k)\|^\alpha}$ satisfy

$$\|x_{k+1} - x_*\|^2 \le \left(1 - \frac{\nu\mu}{2K_0 + (2K_1 + 2^{1-\alpha}K_2^{1-\alpha})(K_0 + K_2\|x_0 - x_*\|^{\alpha/1-\alpha})^\alpha\|x_0 - x_*\|^\alpha}\right)^{k+1}\|x_0 - x_*\|^2.$$

where $\nu \in (0, 1)$ is a constant such that $1 - \nu - \nu^2 \ge 0$.

*Proof.* Using the update steps of the Extragradient method and using $\mu$-strong monotonicity, we have

$$\begin{aligned}
\|x_{k+1} - x_*\|^2 &\overset{(44)}{\le} (1 - \gamma_k\mu)\|x_k - x_*\|^2 - \gamma_k^2(1 - 2\gamma_k\mu)\|F(x_k)\|^2 + \gamma_k^2\|F(x_k) - F(\hat{x}_k)\|^2 \\
&\overset{(14)}{\le} (1 - \gamma_k\mu)\|x_k - x_*\|^2 - \gamma_k^2(1 - 2\gamma_k\mu)\|F(x_k)\|^2 \\
&\quad + \gamma_k^2\left(K_0 + K_1\|F(x_k)\|^\alpha + K_2\|x_k - \hat{x}_k\|^{\alpha/1-\alpha}\right)^2\|x_k - \hat{x}_k\|^2 \\
&= (1 - \gamma_k\mu)\|x_k - x_*\|^2 \\
&\quad + \gamma_k^2\left(1 - 2\gamma_k\mu - \gamma_k^2\left(K_0 + K_1\|F(x_k)\|^\alpha + \gamma_k^{\alpha/1-\alpha}K_2\|F(x_k)\|^{\alpha/1-\alpha}\right)^2\right)\|F(x_k)\|^2.
\end{aligned}$$

Here we will choose $\gamma_k > 0$ such that

$$1 - 2\gamma_k\mu - \gamma_k^2\left(K_0 + K_1\|F(x_k)\|^\alpha + \gamma_k^{\alpha/1-\alpha}K_2\|F(x_k)\|^{\alpha/1-\alpha}\right)^2 \ge 0$$

Let us choose $\gamma_k = \frac{\nu}{2K_0 + \left(2K_1 + 2^{1-\alpha}K_2^{1-\alpha}\right)\|F(x_k)\|^\alpha}$ for some $\nu \in (0, 1)$. Then we observe that

$$\begin{aligned}
\gamma_k\left(K_0 + K_1\|F(x_k)\|^\alpha\right) + \gamma_k^{1/1-\alpha}K_2\|F(x_k)\|^{\alpha/1-\alpha} &\le \frac{\nu\left(K_0 + K_1\|F(x_k)\|^\alpha\right)}{2K_0 + \left(2K_1 + 2^{1-\alpha}K_2^{1-\alpha}\right)\|F(x_k)\|^\alpha} \\
&\quad + \frac{\nu^{1/1-\alpha}K_2\|F(x_k)\|^{\alpha/1-\alpha}}{\left(2K_0 + \left(2K_1 + 2^{1-\alpha}K_2^{1-\alpha}\right)\|F(x_k)\|^\alpha\right)^{1/1-\alpha}} \\
&\le \frac{\nu}{2} + \frac{\nu^{1/1-\alpha}}{2} \\
&\le \nu.
\end{aligned}$$

The last inequality follows from $\nu \in (0, 1)$. Therefore it is enough to choose $\nu \in (0, 1)$ such that

$$1 - \frac{2\nu\mu}{2K_0 + \left(2K_1 + 2^{1-\alpha}K_2^{1-\alpha}\right)\|F(x_k)\|^\alpha} - \nu^2 \geq 0$$

However, note that, we always have $\mu \leq K_0$, thus it is enough to choose $\nu \in (0, 1)$ such that

$$1 - \nu - \nu^2 \geq 0.$$

Hence, for this choice of $\nu$ we get

$$\|x_{k+1} - x_*\|^2 \quad \leq \quad (1 - \gamma_k\mu)\|x_0 - x_*\|^2.$$

Here we lower bound the step size $\gamma_k$ with

$$\gamma_k \geq \frac{\nu}{2K_0 + (2K_1 + 2^{1-\alpha}K_2^{1-\alpha})(K_0 + K_2\|x_0 - x_*\|^{\alpha/1-\alpha})^\alpha\|x_0 - x_*\|^\alpha}.$$

Hence we obtain

$$\|x_{k+1} - x_*\|^2 \leq \quad \left(1 - \frac{\nu\mu}{2K_0 + (2K_1 + 2^{1-\alpha}K_2^{1-\alpha})(K_0 + K_2\|x_0 - x_*\|^{\alpha/1-\alpha})^\alpha\|x_0 - x_*\|^\alpha}\right)\|x_k - x_*\|^2$$

$$\leq \quad \left(1 - \frac{\nu\mu}{2K_0 + (2K_1 + 2^{1-\alpha}K_2^{1-\alpha})(K_0 + K_2\|x_0 - x_*\|^{\alpha/1-\alpha})^\alpha\|x_0 - x_*\|^\alpha}\right)^{k+1}\|x_0 - x_*\|^2.$$

This completes the proof of the theorem. $\qquad\square$

### D.3 Convergence Guarantees for Monotone Operators

**Lemma D.6.** Suppose $F$ is a monotone operator. Then EG with step size $\gamma_k = \omega_k$ satisfy

$$\|x_{k+1} - x_*\|^2 \leq \|x_k - x_*\|^2 - \gamma_k^2 \|F(x_k)\|^2 + \gamma_k^2 \|F(x_k) - F(\hat{x}_k)\|^2. \tag{53}$$

*Proof.* From the update rule of the Extragradient method, we have

$$
\begin{aligned}
\|x_{k+1} - x_*\|^2 &= \|x_k - \gamma_k F(\hat{x}_k) - x_*\|^2 \\
&= \|x_k - x_*\|^2 - 2\gamma_k \langle F(\hat{x}_k), x_k - x_* \rangle + \gamma_k^2 \|F(\hat{x}_k)\|^2 \\
&= \|x_k - x_*\|^2 - 2\gamma_k \langle F(\hat{x}_k), \hat{x}_k - x_* \rangle - 2\gamma_k \langle F(\hat{x}_k), x_k - \hat{x}_k \rangle + \gamma_k^2 \|F(\hat{x}_k)\|^2 \\
&\overset{(8)}{\leq} \|x_k - x_*\|^2 - 2\gamma_k \langle F(\hat{x}_k), x_k - \hat{x}_k \rangle + \gamma_k^2 \|F(\hat{x}_k)\|^2 \\
&= \|x_k - x_*\|^2 - 2\gamma_k^2 \langle F(\hat{x}_k), F(x_k) \rangle + \gamma_k^2 \|F(\hat{x}_k)\|^2 \\
&\overset{(22)}{=} \|x_k - x_*\|^2 - \gamma_k^2 \|F(\hat{x}_k)\|^2 - \gamma_k^2 \|F(x_k)\|^2 + \gamma_k^2 \|F(x_k) - F(\hat{x}_k)\|^2 + \gamma_k^2 \|F(\hat{x}_k)\|^2 \\
&= \|x_k - x_*\|^2 - \gamma_k^2 \|F(x_k)\|^2 + \gamma_k^2 \|F(x_k) - F(\hat{x}_k)\|^2.
\end{aligned}
$$

$\square$

### D.3.1 Proof of Theorem 3.5

**Theorem D.7.** Suppose $F$ is monotone and 1-symmetric $(L_0, L_1)$-Lipschitz operator. Then EG with step size $\gamma_k = \omega_k = \frac{\nu}{L_0 + L_1 \|F(x_k)\|}$ satisfy

$$\min_{0 \leq k \leq K} \|F(x_k)\|^2 \leq \frac{2L_0^2 \left(1 + L_1 \exp\left(L_1 \|x_0 - x_*\|\right) \|x_0 - x_*\|\right)^2 \|x_0 - x_*\|^2}{\nu^2 (K+1)}.$$

where $\nu \exp \nu = 1/\sqrt{2}$.

*Proof.* From the update rule of the Extragradient method and using monotonicity, we have

$$
\begin{aligned}
\|x_{k+1} - x_*\|^2 &\overset{(53)}{\leq} \|x_k - x_*\|^2 - \gamma_k^2 \|F(x_k)\|^2 + \gamma_k^2 \|F(x_k) - F(\hat{x}_k)\|^2 \\
&\overset{(13)}{\leq} \|x_k - x_*\|^2 - \gamma_k^2 \|F(x_k)\|^2 \\
&\quad + \gamma_k^2 (L_0 + L_1 \|F(x_k)\|)^2 \exp\left(2L_1 \|x_k - \hat{x}_k\|\right) \|x_k - \hat{x}_k\|^2 \\
&= \|x_k - x_*\|^2 - \gamma_k^2 \left(1 - \gamma_k^2 (L_0 + L_1 \|F(x_k)\|)^2 \exp\left(2\gamma_k L_1 \|F(x_k)\|\right)\right) \|F(x_k)\|^2 \\
&\leq \|x_k - x_*\|^2 \\
&\quad - \gamma_k^2 \left(1 - \gamma_k^2 (L_0 + L_1 \|F(x_k)\|)^2 \exp\left(2\gamma_k (L_0 + L_1 \|F(x_k)\|)\right)\right) \|F(x_k)\|^2.
\end{aligned}
$$

Here we use $\gamma_k = \frac{\nu}{L_0 + L_1 \|F(x_k)\|}$ for some $\nu \in (0,1)$ to get

$$\|x_{k+1} - x_*\|^2 \leq \|x_k - x_*\|^2 - \gamma_k^2 \left(1 - \nu^2 \exp(2\nu)\right) \|F(x_k)\|^2$$

Then we choose $\nu$ such that $\nu \exp \nu = 1/\sqrt{2}$ to obtain

$$\|x_{k+1} - x_*\|^2 \leq \|x_k - x_*\|^2 - \frac{\gamma_k^2}{2} \|F(x_k)\|^2. \tag{54}$$

In particular the distance of the iterates $x_k$ from $x_*$ are bounded i.e. $\|x_{k+1} - x_*\| \leq \|x_k - x_*\| \leq \|x_0 - x_*\|$. Therefore, using (13) with $y = x_*$ and $x = x_k$, we get

$$
\begin{aligned}
\|F(x_k)\| &\leq L_0 \exp\left(L_1 \|x_k - x_*\|\right) \|x_k - x_*\| \\
&\leq L_0 \exp\left(L_1 \|x_0 - x_*\|\right) \|x_0 - x_*\|.
\end{aligned}
$$

Then we have the lower bound on step size given as follows

$$\gamma_k = \frac{\nu}{L_0 + L_1 \|F(x_k)\|} \geq \frac{\nu}{L_0 \left(1 + L_1 \exp\left(L_1 \|x_0 - x_*\|\right) \|x_0 - x_*\|\right)}. \tag{55}$$

Rearranging (54) we have

$$\frac{\gamma_k^2}{2} \|F(x_k)\|^2 \leq \|x_k - x_*\|^2 - \|x_{k+1} - x_*\|^2.$$

Summing up the above inequality for $k = 0, 1, \cdots, K$ and dividing by $K+1$ we get

$$\frac{1}{K+1} \sum_{k=0}^{K} \frac{\gamma_k^2}{2} \|F(x_k)\|^2 \leq \frac{\|x_0 - x_*\|^2 - \|x_{K+1} - x_*\|^2}{K+1} \leq \frac{\|x_0 - x_*\|^2}{K+1}. \tag{56}$$

Here we will use the lower bound on step size $\gamma_k$ given in (55) to get

$$\frac{1}{K+1}\sum_{k=0}^{K}\|F(x_k)\|^2 \quad\leq\quad \frac{2L_0^2\left(1+L_1\exp\left(L_1\|x_0-x_*\|\right)\|x_0-x_*\|\right)^2\|x_0-x_*\|^2}{\nu^2(K+1)}.$$

Finally, note that $\min_{0\leq k\leq K}\|F(x_k)\|^2 \leq \frac{1}{K+1}\sum_{k=0}^{K}\|F(x_k)\|^2$. Therefore we have

$$\min_{0\leq k\leq K}\|F(x_k)\|^2 \quad\leq\quad \frac{2L_0^2\left(1+L_1\exp\left(L_1\|x_0-x_*\|\right)\|x_0-x_*\|\right)^2\|x_0-x_*\|^2}{\nu^2(K+1)}.$$

This completes the proof of the Theorem. $\qquad\square$

### D.3.2 Proof of Theorem 3.6

**Theorem D.8.** Suppose $F$ is monotone and 1-symmetric $(L_0, L_1)$-Lipschitz operator. Then Extragradient method with step size $\gamma_k = \frac{\nu}{L_0+L_1\|F(x_k)\|}$ satisfy

$$\min_{0\leq k\leq K}\|F(x_k)\| \quad\leq\quad \frac{\sqrt{2}L_0\|x_0-x_*\|}{\nu\sqrt{K+1}-\sqrt{2}L_1\|x_0-x_*\|}$$

where $\nu\exp\nu = 1/\sqrt{2}$ and $K+1 \geq \frac{2L_1^2\|x_0-x_*\|^2}{\nu^2}$.

*Proof.* From (56), we know steps of Extragradient method satisfy

$$\frac{1}{K+1}\sum_{k=0}^{K}\frac{\gamma_k^2}{2}\|F(x_k)\|^2 \quad\leq\quad \frac{\|x_0-x_*\|^2}{K+1}.$$

Taking the minimum on the left-hand side we have

$$\min_{0\leq k\leq K}\gamma_k^2\|F(x_k)\|^2 \quad\leq\quad \frac{2\|x_0-x_*\|^2}{K+1},$$

or equivalently,

$$\min_{0\leq k\leq K}\frac{\nu^2\|F(x_k)\|^2}{(L_0+L_1\|F(x_k)\|)^2} \quad\leq\quad \frac{2\|x_0-x_*\|^2}{K+1}.$$

Taking the square root on both sides we have

$$\min_{0\leq k\leq K}\frac{\nu\|F(x_k)\|}{L_0+L_1\|F(x_k)\|} \quad\leq\quad \frac{\sqrt{2}\|x_0-x_*\|}{\sqrt{K+1}}.$$

Therefore, for some $0\leq k_0\leq K$ we have

$$\frac{\|F(x_{k_0})\|}{L_0+L_1\|F(x_{k_0})\|} \quad\leq\quad \frac{\sqrt{2}\|x_0-x_*\|}{\nu\sqrt{K+1}}.$$

Therefore, rearranging these terms, we get

$$\left(\nu\sqrt{K+1}-\sqrt{2}L_1\|x_0-x_*\|\right)\|F(x_{k_0})\| \quad\leq\quad \sqrt{2}L_0\|x_0-x_*\|.$$

When we have $K+1 \geq \frac{2L_1^2\|x_0-x_*\|^2}{\nu^2}$ then we can rearrange the terms to obtain

$$\|F(x_{k_0})\| \quad\leq\quad \frac{\sqrt{2}L_0\|x_0-x_*\|}{\left(\nu\sqrt{K+1}-\sqrt{2}L_1\|x_0-x_*\|\right)}.$$

for some $0\leq k_0\leq K$. Hence, we complete the proof of the theorem

$$\min_{0\leq k\leq K}\|F(x_k)\| \quad\leq\quad \frac{\sqrt{2}L_0\|x_0-x_*\|}{\nu\sqrt{K+1}-\sqrt{2}L_1\|x_0-x_*\|}.$$

$\qquad\square$

### D.3.3 Proof of Theorem 3.7

**Theorem D.9.** Suppose $F$ is monotone and $\alpha$-symmetric $(L_0, L_1)$-Lipschitz operator with $\alpha \in (0, 1)$. Then Extragradient method with step size $\gamma_k = \frac{1}{2\sqrt{2}K_0+\left(2\sqrt{2}K_1+2^{3(1-\alpha)/2}K_2^{1-\alpha}\right)\|F(x_k)\|^\alpha}$ satisfy

$$\min_{0\leq k\leq K}\|F(x_k)\|^2 \leq \frac{16\left(K_0+\left(K_1+2^{-3/2}K_2^{1-\alpha}\right)(K_0+K_2\|x_0-x_*\|^{\alpha/1-\alpha})^\alpha\|x_0-x_*\|^\alpha\right)^2\|x_0-x_*\|^2}{K+1}.$$

*Proof.* Here operator $F$ is a $\alpha$-symmetric $(L_0, L_1)$-Lipschitz i.e. it satisfies (14). For the update steps of the Extragradient method, we have

$$
\begin{aligned}
\|x_{k+1} - x_*\|^2 &\overset{(53)}{\leq} \|x_k - x_*\|^2 - \gamma_k^2 \|F(x_k)\|^2 + \gamma_k^2 \|F(x_k) - F(\hat{x}_k)\|^2 \\
&\overset{(14)}{\leq} \|x_k - x_*\|^2 - \gamma_k^2 \|F(x_k)\|^2 \\
&\quad + \gamma_k^2 \left( K_0 + K_1 \|F(x_k)\|^\alpha + K_2 \|x_k - \hat{x}_k\|^{\alpha/1-\alpha} \right)^2 \|x_k - \hat{x}_k\|^2 \\
&= \|x_k - x_*\|^2 \\
&\quad - \gamma_k^2 \left( 1 - \gamma_k^2 \left( K_0 + K_1 \|F(x_k)\|^\alpha + \gamma_k^{\alpha/1-\alpha} K_2 \|F(x_k)\|^{\alpha/1-\alpha} \right)^2 \right) \|F(x_k)\|^2 . \quad (57)
\end{aligned}
$$

Here we want to choose $\gamma_k$ such that

$$
\gamma_k \left( K_0 + K_1 \|F(x_k)\|^\alpha \right) + \gamma_k^{1/1-\alpha} K_2 \|F(x_k)\|^{\alpha/1-\alpha} \leq \frac{1}{\sqrt{2}}.
$$

For this, it is enough to make sure

$$
\gamma_k \left( K_0 + K_1 \|F(x_k)\|^\alpha \right) \leq \frac{1}{2\sqrt{2}} \quad \text{and} \quad \gamma_k^{1/1-\alpha} K_2 \|F(x_k)\|^{\alpha/1-\alpha} \leq \frac{1}{2\sqrt{2}}.
$$

Therefore, we choose $\gamma_k = \frac{1}{2\sqrt{2}(K_0 + K_1 \|F(x_k)\|^\alpha) + 2^{3(1-\alpha)/2} K_2^{1-\alpha} \|F(x_k)\|^\alpha}$ and we get the following from (57)

$$
\|x_{k+1} - x_*\|^2 \leq \|x_k - x_*\|^2 - \frac{\gamma_k^2}{2} \|F(x_k)\|^2. \tag{58}
$$

Rearranging this inequality, we have

$$
\frac{\gamma_k^2}{2} \|F(x_k)\|^2 \leq \|x_k - x_*\|^2 - \|x_{k+1} - x_*\|^2.
$$

Then we sum up this inequality for $k = 0, 1, \cdots K$ to get

$$
\frac{1}{K+1} \sum_{k=0}^{K} \gamma_k^2 \|F(x_k)\|^2 \leq \frac{2\|x_0 - x_*\|^2}{K+1}. \tag{59}
$$

For this step size, we also have $\|x_k - x_0\|^2 \leq \|x_0 - x_*\|^2$ from (62). Now note that from (14) we obtain the following bound with $x = x_k$ and $y = x_*$

$$
\begin{aligned}
\|F(x_k)\|^\alpha &\leq (K_0 + K_2 \|x_k - x_*\|^{\alpha/1-\alpha})^\alpha \|x_k - x_*\|^\alpha \\
&\overset{(62)}{\leq} (K_0 + K_2 \|x_0 - x_*\|^{\alpha/1-\alpha})^\alpha \|x_0 - x_*\|^\alpha.
\end{aligned}
$$

We use this to lower bound the step size $\gamma_k$ as follows

$$
\begin{aligned}
\gamma_k &= \frac{1}{2\sqrt{2}(K_0 + K_1 \|F(x_k)\|^\alpha) + 2^{3(1-\alpha)/2} K_2^{1-\alpha} \|F(x_k)\|^\alpha} \\
&\geq \frac{1}{2\sqrt{2} K_0 + 2\sqrt{2}(K_1 + 2^{-3/2} K_2^{1-\alpha})(K_0 + K_2 \|x_0 - x_*\|^{\alpha/1-\alpha})^\alpha \|x_0 - x_*\|^\alpha}.
\end{aligned}
$$

Therefore from (63) we obtain

$$
\min_{0 \leq k \leq K} \|F(x_k)\|^2 \leq \frac{16 \left( K_0 + (K_1 + 2^{-3/2} K_2^{1-\alpha})(K_0 + K_2 \|x_0 - x_*\|^{\alpha/1-\alpha})^\alpha \|x_0 - x_*\|^\alpha \right)^2 \|x_0 - x_*\|^2}{K+1}.
$$

$\square$

## D.4 Local Convergence Guarantees for Weak Minty Operator

### D.4.1 Proof of Theorem 3.8

**Theorem D.10.** Suppose $F$ is weak Minty and 1-symmetric $(L_0, L_1)$-Lipschitz assumption. Moreover we assume

$$\Delta_1 := \frac{\nu}{L_0 \left(1 + L_1 \|x_0 - x_*\| e^{L_1 \|x_0 - x_*\|}\right)} - 4\rho > 0. \tag{60}$$

Then EG with step size $\gamma_k = \frac{\nu}{L_0 + L_1 \|F(x_k)\|}$ and $\omega_k = \gamma_k/2$ satisfies

$$\min_{0 \le k \le K} \|F(\hat{x}_k)\|^2 \le \frac{4L_0 \left(1 + L_1 \exp\left(L_1 \|x_0 - x_*\|\right) \|x_0 - x_*\|\right) \|x_0 - x_*\|^2}{\nu \Delta_1 (K+1)} \tag{61}$$

where $\nu \exp \nu = 1$.

*Proof.* From the update rule of the Extragradient method, we have

$$\|x_{k+1} - x_*\|^2 = \left\|x_k - \frac{\gamma_k}{2} F(\hat{x}_k) - x_*\right\|^2$$

$$= \|x_k - x_*\|^2 - \gamma_k \langle F(\hat{x}_k), x_k - x_* \rangle + \frac{\gamma_k^2}{4} \|F(\hat{x}_k)\|^2$$

$$= \|x_k - x_*\|^2 - \gamma_k \langle F(\hat{x}_k), \hat{x}_k - x_* \rangle - \gamma_k \langle F(\hat{x}_k), x_k - \hat{x}_k \rangle + \frac{\gamma_k^2}{4} \|F(\hat{x}_k)\|^2$$

$$\overset{(10)}{\le} \|x_k - x_*\|^2 + \gamma_k \rho \|F(\hat{x}_k)\|^2 - \gamma_k \langle F(\hat{x}_k), x_k - \hat{x}_k \rangle + \frac{\gamma_k^2}{4} \|F(\hat{x}_k)\|^2$$

$$= \|x_k - x_*\|^2 + \gamma_k \rho \|F(\hat{x}_k)\|^2 - \gamma_k^2 \langle F(\hat{x}_k), F(x_k) \rangle + \frac{\gamma_k^2}{4} \|F(\hat{x}_k)\|^2$$

$$\overset{(22)}{=} \|x_k - x_*\|^2 + \gamma_k \rho \|F(\hat{x}_k)\|^2 - \frac{\gamma_k^2}{2} \|F(\hat{x}_k)\|^2 - \frac{\gamma_k^2}{2} \|F(x_k)\|^2$$
$$+ \frac{\gamma_k^2}{2} \|F(x_k) - F(\hat{x}_k)\|^2 + \frac{\gamma_k^2}{4} \|F(\hat{x}_k)\|^2$$

$$= \|x_k - x_*\|^2 + \gamma_k \rho \|F(\hat{x}_k)\|^2 - \frac{\gamma_k^2}{4} \|F(\hat{x}_k)\|^2 - \frac{\gamma_k^2}{2} \|F(x_k)\|^2$$
$$+ \frac{\gamma_k^2}{2} \|F(x_k) - F(\hat{x}_k)\|^2$$

$$\overset{(13)}{\le} \|x_k - x_*\|^2 + \gamma_k \rho \|F(\hat{x}_k)\|^2 - \frac{\gamma_k^2}{4} \|F(\hat{x}_k)\|^2 - \frac{\gamma_k^2}{2} \|F(x_k)\|^2$$
$$+ \frac{\gamma_k^2}{2} \left(L_0 + L_1 \|F(x_k)\|\right)^2 \exp\left(2L_1 \|x_k - \hat{x}_k\|\right) \|x_k - \hat{x}_k\|^2$$

$$= \|x_k - x_*\|^2 - \frac{\gamma_k}{4} \left(\gamma_k - 4\rho\right) \|F(\hat{x}_k)\|^2$$
$$- \frac{\gamma_k^2}{2} \left(1 - \gamma_k^2 (L_0 + L_1 \|F(x_k)\|)^2 \exp\left(2\gamma_k L_1 \|F(x_k)\|\right)\right) \|F(x_k)\|^2$$

$$\le \|x_k - x_*\|^2 - \frac{\gamma_k}{4} \left(\gamma_k - 4\rho\right) \|F(\hat{x}_k)\|^2$$
$$- \frac{\gamma_k^2}{2} \left(1 - \gamma_k^2 (L_0 + L_1 \|F(x_k)\|)^2 \exp\left(2\gamma_k (L_0 + L_1 \|F(x_k)\|)\right)\right) \|F(x_k)\|^2.$$

Similar to the proof of Theorem 3.5, we have

$$\|x_{k+1} - x_*\|^2 \le \|x_k - x_*\|^2 - \frac{\gamma_k}{4} (\gamma_k - 4\rho) \|F(\hat{x}_k)\|^2$$

for $\gamma_k = \frac{\nu}{L_0 + L_1 \|F(x_k)\|}$ and $\nu \exp \nu = 1$. Again similar to Theorem 3.5, step size $\gamma_k$ is lower bounded with

$$\gamma_k = \frac{\nu}{L_0 + L_1 \|F(x_k)\|} \ge \frac{\nu}{L_0 \left(1 + L_1 \exp\left(L_1 \|x_0 - x_*\|\right) \|x_0 - x_*\|\right)}.$$

Hence from (17) we get

$$\frac{\gamma_k \Delta_1}{4} \|F(\hat{x}_k)\|^2 \le \|x_k - x_*\|^2 - \|x_{k+1} - x_*\|^2$$

Then we sum up this inequality for $k = 0, 1, \cdots, K$ to get

$$\frac{1}{K+1} \sum_{k=0}^{K} \frac{\gamma_k \Delta_1}{4} \|F(\hat{x}_k)\|^2 \le \frac{\|x_0 - x_*\|^2}{K+1}.$$

Therefore, we get

$$\min_{0\le k\le K} \|F(\hat{x}_k)\|^2 \le \frac{4\|x_0 - x_*\|^2}{\gamma\Delta_1(K+1)}.$$

$\square$

### D.4.2 Proof of Theorem 3.9

**Theorem D.11.** Suppose $F$ is weak Minty and $\alpha$-symmetric $(L_0, L_1)$-Lipschitz with $\alpha \in (0, 1)$. Moreover we assume

$$\Delta_\alpha := \frac{1}{2\sqrt{2}K_0 + 2\sqrt{2}(K_1 + 2^{-3/2}K_2^{1-\alpha})(K_0 + K_2\|x_0 - x_*\|^{\alpha/1-\alpha})^\alpha\|x_0 - x_*\|^\alpha} - 4\rho > 0.$$

Then EG with step size $\gamma_k = \frac{1}{2\sqrt{2}K_0 + (2\sqrt{2}K_1 + 2^{3(1-\alpha)/2}K_2^{1-\alpha})\|F(x_k)\|^\alpha}$ and $\omega_k = \frac{\gamma_k}{2}$ satisfy

$$\min_{0\le k\le K} \|F(\hat{x}_k)\|^2 \le \frac{4\left(K_0 + \left(K_1 + 2^{-3/2}K_2^{1-\alpha}\right)(K_0 + K_2\|x_0-x_*\|^{\alpha/1-\alpha})^\alpha\|x_0-x_*\|^\alpha\right)\|x_0-x_*\|^2}{\Delta_\alpha(K+1)}.$$

*Proof.* From the update rule of the Extragradient method, we have

$$
\begin{aligned}
\|x_{k+1} - x_*\|^2 &= \left\|x_k - \frac{\gamma_k}{2}F(\hat{x}_k) - x_*\right\|^2 \\
&= \|x_k - x_*\|^2 - \gamma_k \langle F(\hat{x}_k), x_k - x_* \rangle + \frac{\gamma_k^2}{4}\|F(\hat{x}_k)\|^2 \\
&= \|x_k - x_*\|^2 - \gamma_k \langle F(\hat{x}_k), \hat{x}_k - x_* \rangle - \gamma_k \langle F(\hat{x}_k), x_k - \hat{x}_k \rangle + \frac{\gamma_k^2}{4}\|F(\hat{x}_k)\|^2 \\
&\overset{(10)}{\le} \|x_k - x_*\|^2 + \gamma_k\rho\|F(\hat{x}_k)\|^2 - \gamma_k \langle F(\hat{x}_k), x_k - \hat{x}_k \rangle + \frac{\gamma_k^2}{4}\|F(\hat{x}_k)\|^2 \\
&= \|x_k - x_*\|^2 + \gamma_k\rho\|F(\hat{x}_k)\|^2 - \gamma_k^2 \langle F(\hat{x}_k), F(x_k) \rangle + \frac{\gamma_k^2}{4}\|F(\hat{x}_k)\|^2 \\
&\overset{(22)}{=} \|x_k - x_*\|^2 + \gamma_k\rho\|F(\hat{x}_k)\|^2 - \frac{\gamma_k^2}{2}\|F(\hat{x}_k)\|^2 - \frac{\gamma_k^2}{2}\|F(x_k)\|^2 \\
&\quad + \frac{\gamma_k^2}{2}\|F(x_k) - F(\hat{x}_k)\|^2 + \frac{\gamma_k^2}{4}\|F(\hat{x}_k)\|^2 \\
&= \|x_k - x_*\|^2 + \gamma_k\rho\|F(\hat{x}_k)\|^2 - \frac{\gamma_k^2}{4}\|F(\hat{x}_k)\|^2 - \frac{\gamma_k^2}{2}\|F(x_k)\|^2 \\
&\quad + \frac{\gamma_k^2}{2}\|F(x_k) - F(\hat{x}_k)\|^2 \\
&\overset{(13)}{\le} \|x_k - x_*\|^2 + \gamma_k\rho\|F(\hat{x}_k)\|^2 - \frac{\gamma_k^2}{4}\|F(\hat{x}_k)\|^2 - \frac{\gamma_k^2}{2}\|F(x_k)\|^2 \\
&\quad + \frac{\gamma_k^2}{2}\left(K_0 + K_1\|F(x_k)\|^\alpha + K_2\|x_k - \hat{x}_k\|^{\alpha/1-\alpha}\right)^2 \|x_k - \hat{x}_k\|^2 \\
&= \|x_k - x_*\|^2 - \frac{\gamma_k}{4}\left(\gamma_k - 4\rho\right)\|F(\hat{x}_k)\|^2 \\
&\quad - \frac{\gamma_k^2}{2}\left(1 - \gamma_k^2\left(K_0 + K_1\|F(x_k)\|^\alpha + \gamma_k^{\alpha/1-\alpha}K_2\|F(x_k)\|^{\alpha/1-\alpha}\right)^2\right)\|F(x_k)\|^2.
\end{aligned}
$$

Here we choose $\gamma_k = \frac{1}{2(K_0 + K_1\|F(x_k)\|^\alpha) + 2^{3(1-\alpha)/2}K_2^{1-\alpha}\|F(x_k)\|^\alpha}$ and we get the following from (57)

$$\|x_{k+1} - x_*\|^2 \le \|x_k - x_*\|^2 - \frac{\gamma_k}{4}\left(\gamma_k - 4\rho\right)\|F(\hat{x}_k)\|^2. \tag{62}$$

Rearranging this inequality, we have

$$\frac{\gamma_k}{4}\left(\gamma_k - 4\rho\right)\|F(\hat{x}_k)\|^2 \le \|x_k - x_*\|^2 - \|x_{k+1} - x_*\|^2.$$

Then we sum up this inequality for $k = 0, 1, \cdots K$ to get

$$\frac{1}{K+1}\sum_{k=0}^{K}\frac{\gamma_k}{4}\left(\gamma_k - 4\rho\right)\|F(\hat{x}_k)\|^2 \le \frac{2\|x_0 - x_*\|^2}{K+1}. \tag{63}$$

For this step size, we also have $\|x_k - x_*\|^2 \leq \|x_0 - x_*\|^2$ from (62). Now note that from (14) we obtain the following bound with $x = x_k$ and $y = x_*$

$$
\begin{aligned}
\|F(x_k)\|^\alpha \quad &\leq \quad (K_0 + K_2\|x_k - x_*\|^{\alpha/1-\alpha})^\alpha \|x_k - x_*\|^\alpha \\
&\overset{(62)}{\leq} \quad (K_0 + K_2\|x_0 - x_*\|^{\alpha/1-\alpha})^\alpha \|x_0 - x_*\|^\alpha.
\end{aligned}
$$

We use this to lower bound the step size $\gamma_k$ as follows

$$
\begin{aligned}
\gamma_k \quad &= \quad \frac{1}{2\sqrt{2}(K_0 + K_1\|F(x_k)\|^\alpha) + 2^{3(1-\alpha)/2}K_2^{1-\alpha}\|F(x_k)\|^\alpha} \\
&\geq \quad \frac{1}{2\sqrt{2}K_0 + 2\sqrt{2}(K_1 + 2^{-3/2}K_2^{1-\alpha})(K_0 + K_2\|x_0 - x_*\|^{\alpha/1-\alpha})^\alpha\|x_0 - x_*\|^\alpha}.
\end{aligned}
$$

Therefore from (63) we obtain

$$
\min_{0 \leq k \leq K} \|F(x_k)\|^2 \leq \frac{4\|x_0 - x_*\|^2}{\gamma\Delta(K+1)}.
$$

$\square$

# E  Equivalent Formulation of $\alpha$-Symmetric $(L_0, L_1)$-Lipschitz Assumption

In this section, we consider the min-max optimization problem given by $\min_{w_1} \max_{w_2} \mathcal{L}(w_1, w_2)$ and provide an equivalent formulation of $\alpha$-symmetric $(L_0, L_1)$-Lipschitz operator. Next, we provide an example where we use this formulation to compute the constants $\alpha, L_0, L_1$.

## E.1  Proof of Theorem 2.1

**Theorem E.1.** Suppose $F$ is the operator for the problem

$$\min_{w_1} \max_{w_2} \mathcal{L}(w_1, w_2).$$

Then $F$ satisfies $\alpha$-symmetric $(L_0, L_1)$-Lipschitz assumption if and only if

$$\|\mathbf{J}(x)\| = \sup_{\|u\|=1} \|\mathbf{J}(x)u\| \leq L_0 + L_1 \|F(x)\|^\alpha$$

where

$$\mathbf{J}(x) = \begin{bmatrix} \nabla^2_{w_1 w_1} \mathcal{L}(w_1, w_2) & \nabla^2_{w_2 w_1} \mathcal{L}(w_1, w_2) \\ -\nabla^2_{w_1 w_2} \mathcal{L}(w_1, w_2) & -\nabla^2_{w_2 w_2} \mathcal{L}(w_1, w_2) \end{bmatrix}.$$

Here $\|\mathbf{J}(x)\| = \sigma_{\max}(\mathbf{J}(x))$ i.e. maximum singular value of $\mathbf{J}(x)$.

*Proof.* Following (26), we have the equivalent characterization of $F$ given by

$$\|F(y) - F(x)\| \leq \left( L_0 + L_1 \int_0^1 \|F(\theta y + (1-\theta)x)\|^\alpha \, d\theta \right) \|y - x\| \qquad \forall x, y \in \mathbb{R}^d.$$

As this inequality holds for any $x, y \in \mathbb{R}^d$, we choose $y = x + \theta' u$ where $\|u\| = 1$ and $\theta' \in (0, 1)$. Then we get

$$\|F(x + \theta' u) - F(x)\| \leq \left( L_0 + L_1 \int_0^1 \|F(x + \theta' \theta u)\|^\alpha \, d\theta \right) \|\theta' u\| \qquad \forall x \in \mathbb{R}^d.$$

The right-hand side of this inequality can be rewritten as

$$\left( L_0 + L_1 \int_0^1 \|F(x + \theta' \theta u)\|^\alpha \, d\theta \right) \|\theta' u\| \;=\; \theta' \left( L_0 + L_1 \int_0^1 \|F(x + \theta' \theta u)\|^\alpha \, d\theta \right)$$

$$= L_0 \theta' + L_1 \int_0^1 \|F(x + \theta' \theta u)\|^\alpha \, \theta' d\theta$$

$$= L_0 \theta' + L_1 \int_0^{\theta'} \|F(x + \varphi u)\|^\alpha \, d\varphi.$$

In the last line, we used the change of variable with $\varphi = \theta' \theta$. Therefore, we get

$$\frac{\|F(x + \theta' u) - F(x)\|}{\theta'} \leq L_0 + \frac{L_1}{\theta'} \int_0^{\theta'} \|F(x + \varphi u)\|^\alpha \, d\varphi.$$

Then we take $\theta' \to 0$ and use L'Hôpital's rule and Leibniz Integral rule to obtain

$$\lim_{\theta' \to 0} \frac{\|F(x + \theta' u) - F(x)\|}{\theta'} \leq L_0 + L_1 \|F(x)\|^\alpha.$$

Moreover, note that the left-hand side is given by $\|\mathbf{J}(x)u\|$ where

$$\mathbf{J}(x) = \begin{bmatrix} \nabla^2_{w_1 w_1} \mathcal{L}(w_1, w_2) & \nabla^2_{w_2 w_1} \mathcal{L}(w_1, w_2) \\ -\nabla^2_{w_1 w_2} \mathcal{L}(w_1, w_2) & -\nabla^2_{w_2 w_2} \mathcal{L}(w_1, w_2) \end{bmatrix}.$$

Therefore, for any $\|u\| = 1$ we have

$$\|\mathbf{J}(x)u\| \leq L_0 + L_1 \|F(x)\|^\alpha.$$

Hence we get

$$\|\mathbf{J}(x)\| = \sup_{\|u\|=1} \|\mathbf{J}(x)u\| \leq L_0 + L_1 \|F(x)\|^\alpha.$$

Now we want to show the other way, i.e. suppose we have $\|\mathbf{J}(x)\| \leq L_0 + L_1 \|F(x)\|^\alpha$. For this we define,

$$q(\theta) := F(\theta x + (1 - \theta)y).$$

Then $q(1) = F(x)$ and $q(0) = F(y)$ and we have

$$
\begin{aligned}
\|F(x) - F(y)\| &= \|q(1) - q(0)\| \\
&= \left\| \int_0^1 \frac{dq(\theta)}{d\theta} d\theta \right\| \\
&= \left\| \int_0^1 \frac{dF(\theta x + (1-\theta)y)}{d\theta} d\theta \right\| \\
&= \left\| \int_0^1 \mathbf{J}(\theta x + (1-\theta)y)(x-y)d\theta \right\| \\
&\leq \int_0^1 \|\mathbf{J}(\theta x + (1-\theta)y)\| \, \|x - y\| \, d\theta \\
&= \left( \int_0^1 \|\mathbf{J}(\theta x + (1-\theta)y)\| \, d\theta \right) \|x - y\| \\
&\leq \left( \int_0^1 L_0 + L_1 \|F(\theta x + (1-\theta)y)\|^\alpha d\theta \right) \|x - y\| \\
&= \left( L_0 + L_1 \int_0^1 \|F(\theta x + (1-\theta)y)\|^\alpha d\theta \right) \|x - y\| .
\end{aligned}
$$

Then, using Lemma B.5, we have the result. $\qquad\square$

## E.2  Computation of $\alpha, L_0, L_1$ for $\mathcal{L}(w_1, w_2)$.

We now revisit the min-max problem defined in (6). Note that, the operator corresponding to this problem is given by

$$
F(x) = \begin{bmatrix} w_1^2 + w_2 \\ w_2^2 - w_1 \end{bmatrix}
$$

Then the norm of operator is $\|F(x)\| = \sqrt{(w_1^2 + w_2)^2 + (w_2^2 - w_1)^2}$. Moreover, the Jacobian matrix is given by

$$
\mathbf{J}(x) = \begin{bmatrix} 2w_1 & 1 \\ -1 & 2w_2 \end{bmatrix}.
$$

Then the maximum singular value at any point $x$ is given by

$$
\begin{aligned}
\|\mathbf{J}(x)\| &= \lambda_{\max}\left( \mathbf{J}(x)^\top \mathbf{J}(x) \right) \\
&= \lambda_{\max}\left( \begin{bmatrix} 4w_1^2 + 1 & 2(w_1 - w_2) \\ 2(w_1 - w_2) & 4w_2^2 + 1 \end{bmatrix} \right) \\
&\overset{(27)}{=} \sqrt{2(w_1^2 + w_2^2) + 1 + 2\sqrt{(w_1 - w_2)^2 + (w_1^2 - w_2^2)^2}}
\end{aligned} \tag{64}
$$

To validate whether the operator $F$ satisfies the condition (7), we examine whether the following function is non-negative:

$$
g(w_1, w_2) = L_0 + L_1 \|F(x)\| - \|\mathbf{J}(x)\|. \tag{65}
$$

In Figure 6, we plot $g(w_1, w_2)$ using $L_0 = 10$ and $L_1 = 10$. We observe that $g(w_1, w_2)$ has no real solution and remains positive for all $w_1, w_2 \in \mathbb{R}$, confirming that the function (6) satisfies (12) with $(\alpha, L_0, L_1) = (1, 10, 10)$. Thus, the corresponding operator $F$ is 1-symmetric $(10, 10)$-Lipschitz.

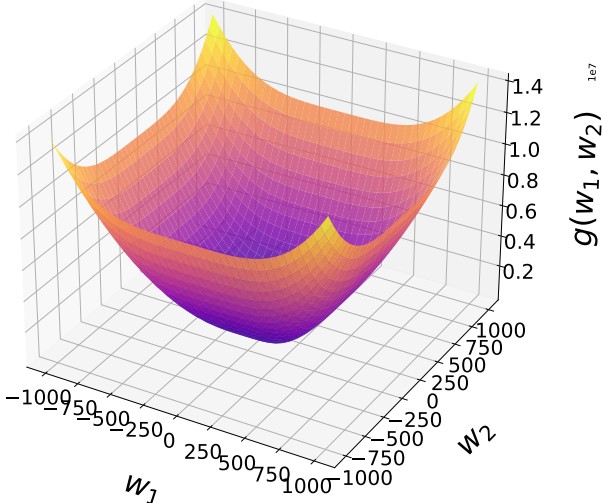

Figure 6: Plot of $g(w_1, w_2)$ (65). Here, the $z$-axis is in $10^7$ scale.

# F  Additional Details on Numerical Experiments

In this section, we provide additional details on the second experiment related to the monotone problem. First, we show that

$$\mathcal{L}(w_1, w_2) = \frac{1}{3} \left( w_1^\top \mathbf{A} w_1 \right)^{3/2} + w_1^\top \mathbf{B} w_2 - \frac{1}{3} \left( w_2^\top \mathbf{C} w_2 \right)^{3/2}$$

is convex-concave, and then we find the equilibrium point of $\mathcal{L}$.

**Convex-Concave $\mathcal{L}(w_1, w_2)$.** Note that for $\mathcal{L}(w_1, w_2)$ in (20) we have $\nabla_{w_1} \mathcal{L}(w_1, w_2) = \left( w_1^\top \mathbf{A} w_1 \right)^{1/2} \mathbf{A} w_1 + \mathbf{B} w_2$ and $\nabla_{w_2} \mathcal{L}(w_1, w_2) = \mathbf{B}^\top w_1 - \left( w_2^\top \mathbf{C} w_2 \right)^{1/2} \mathbf{C} w_2$. Then the second-order derivatives are given by

$$\nabla_{w_1 w_1}^2 \mathcal{L}(w_1, w_2) = \|\mathbf{A}^{1/2} w_1\| \mathbf{A} + \frac{\mathbf{A} w_1 w_1^\top \mathbf{A}^\top}{\|\mathbf{A}^{1/2} w_1\|}$$

Here, $\mathbf{A}$ is positive definite and $\mathbf{A} w_1 w_1^\top \mathbf{A}^\top$ is a positive semidefinite matrix. Hence, $\nabla_{w_1 w_1}^2 \mathcal{L}(w_1, w_2)$ is a positive definite matrix as well and $\mathcal{L}(\cdot, w_2)$ is convex for any $w_2$. Similarly, we show that

$$-\nabla_{w_2 w_2}^2 \mathcal{L}(w_1, w_2) = \|\mathbf{C}^{1/2} w_2\| \mathbf{C} + \frac{\mathbf{C} w_2 w_2^\top \mathbf{C}^\top}{\|\mathbf{C}^{1/2} w_2\|}$$

and $-\nabla_{w_2 w_2}^2 \mathcal{L}(w_1, w_2)$ is positive definite. Therefore, $\mathcal{L}(w_1, \cdot)$ is concave for any $w_1$. This proves that $\mathcal{L}(w_1, w_2)$ is convex with respect to $w_1$ and concave with respect to $w_2$. Thus, we conclude that the corresponding operator $F$ is monotone.

**Equilibrium of $\mathcal{L}(w_1, w_2)$.** To find the equilibrium points, we solve the set of equations given by $\nabla_{w_1} \mathcal{L}(w_1, w_2) = 0$ and $\nabla_{w_2} \mathcal{L}(w_1, w_2) = 0$, i.e., solve for

$$\left( w_1^\top \mathbf{A} w_1 \right)^{1/2} \mathbf{A} w_1 + \mathbf{B} w_2 = 0,$$

$$\mathbf{B}^\top w_1 - \left( w_2^\top \mathbf{C} w_2 \right)^{1/2} \mathbf{C} w_2 = 0.$$

Now multiplying the first equation with $w_1^\top$ and second one with $w_2^\top$, we have

$$\left( w_1^\top \mathbf{A} w_1 \right)^{1/2} w_1^\top \mathbf{A} w_1 + w_1^\top \mathbf{B} w_2 = 0$$

$$w_2^\top \mathbf{B}^\top w_1 - \left( w_2^\top \mathbf{C} w_2 \right)^{1/2} w_2^\top \mathbf{C} w_2 = 0$$

Combining these two equations, we get

$$\left( w_1^\top \mathbf{A} w_1 \right)^{1/2} w_1^\top \mathbf{A} w_1 + \left( w_2^\top \mathbf{C} w_2 \right)^{1/2} w_2^\top \mathbf{C} w_2 = 0$$

which can be equivalently written as

$$\left\| \mathbf{A}^{1/2} w_1 \right\|^3 + \left\| \mathbf{C}^{1/2} w_2 \right\|^3 = 0$$

which implies $w_1 = w_2 = 0$ as both $\mathbf{A}, \mathbf{C}$ are positive definite matrices (hence $\mathbf{A}^{1/2}, \mathbf{C}^{1/2}$ are invertible).

