# OpenReview forum: "Extragradient Method for $(L_0, L_1)$-Lipschitz Root-finding Problems"
_NeurIPS.cc/2025/Conference — NeurIPS 2025 poster_

### Official Review · Reviewer_tsMk · 2025-06-25

**Clarity:** 2
**Significance:** 3
**Originality:** 3
**Rating:** 4
**Confidence:** 3

**Summary:**

The paper extends extragradient methods to variational inequality problems with relaxed assumptions. Most existing work assumes an L-Lipschitz condition on the operator. This paper relaxes this to the more general $\alpha$-symmetric $(L_0,L_1)$-Lipschitz condition. This condition has been studied in the past under a wide range of contexts and captures a number of basic loss functions that the standard L-Lipschitzness cannot capture.

The paper has a few main contributions. The first is that they tighten existing analysis based on extragradient methods. Concretely, consider the function $f(x) = \log (1+\exp(-a^{\top}x))$. This function is $\|a\|^2$ Lipschitz under the standard definition but it is also $\alpha$-symmetric $(L_0,L_1)$-Lipschitz with $\alpha = 1$, $L_0 = 1$, $L_1 = \|a\|$. When $\|a\|>>1$ then these new methods significantly improve upon existing ones. The second contribution is that they give the first convergence analysis of EG methods for weak minty variational inequality problems under this generalized smoothness. The third contribution is that the step size considered for the EG algorithm is new and is adaptive. The rates of convergence in all cases improve upon previous works.

**Questions:**

1. Can the authors give some more information on the methods used to solve these generalized smooth problems? It seems like they are focussed entirely on EG methods, which I am not sure give the best rates of convergence compared to other prox point methods.
2. A complete algorithm box would certainly help.
3. It is not clear what the main challenge is with generalizing the EG methods to this setting. Maybe a bit more description on this would help.

**Ethical Concerns:**

["NO or VERY MINOR ethics concerns only"]

**Final Justification:**

Based on the justifications and rebuttal, I would like to retain my score.

**Limitations:**

I am not sure I see where these are addressed explicitly.

**Quality:**

3

**Strengths And Weaknesses:**

Strengths:

The paper provides a new EG algorithm and the only one which works for VIPs under the generalized smoothness condition. Their algorithm improves upon previous works. The key improvement is via their new adaptive step size.

Weaknesses:

I am not entirely sure of the exact algorithm used in implementations and also the one analyzed. Maybe an algorithm box would help. Also the paper only talks about EG methods. It is not clear if there are other existing works based on other methods that show convergences for these kinds of problems.

---

> ### Author Rebuttal · Authors · 2025-07-31
>
> We thank the reviewer for a positive evaluation of our work and for providing us with valuable questions and comments. We respond to the questions and concerns below.
>
> ## Weakness
> > I am not entirely sure of the exact algorithm used in implementations and also the one analyzed. Maybe an algorithm box would help.] + [...A complete algorithm box would certainly help...
>
> That is an excellent suggestion. We will incorporate that in the updated version of the paper.
>
> ## Questions
>
> > Also the paper only talks about EG methods. It is not clear if there are other existing works based on other methods that show...
>
> In this work, we analyze the extragradient (EG) algorithm under the $\alpha$-symmetric $(L_0, L_1)$-Lipschitz assumption. To the best of our knowledge, [1] is the only work that analyzes extragradient, past extragradient, and the projection method under this assumption. We have included a comparison with this work for EG, and we will include the comparison with the other two algorithms in the updated version.
>
> > It is not clear what the main challenge is with generalizing the EG methods to this setting. Maybe a bit more description on this would help.
>
> In this work we analyze the extragradient method under the $\alpha$-symmetric $(L_0, L_1)$-Lipschitz assumption, which involves a more general bound of the form $L_0 + L_1 \max_{\theta \in [0,1]} \| F(\theta x + (1 - \theta) y) \|^{\alpha}$ instead of the standard Lipschitz constant L. This generalization makes the analysis significantly more challenging. For example, in the standard L-Lipschitz case, one can easily bound $\| F(x_k) - F(\hat{x}_k) \|^2$ by $L^2 \| x_k - \hat{x}_k \|^2$, which appears in line 9 of the proof of Theorem 3.2 (Appendix page 24). However, under the $\alpha$-symmetric $(L_0, L_1)$-Lipschitz assumption, it introduces a much involved term, which complicates the analysis.
>
> This difficulty motivates the use of an adaptive step size of the form $\eta / (L_0 + L_1 \|F(x_k)\|^{\alpha})$, which is better suited for this setting. Moreover, throughout the analysis, several steps require us to carefully use the structure of the $\alpha$-symmetric Lipschitz condition and develop proof techniques that differ from those used under the standard Lipschitz assumption.
>
> We hope this clarifies the technical challenges involved in working under this more general setup. We can easily include this discussion in the updated version of our submission.
>
> **We hope we have addressed all the questions and concerns of the reviewer. In our opinion, all concerns raised are clarification points and not really weaknesses of our work.  If you agree, please consider increasing your score. If not, please let us know so we can respond accordingly.**
>
> ## References
>
> [1] Vankov, Daniil, Angelia Nedich, and Lalitha Sankar. "Generalized smooth variational inequalities: Methods with adaptive stepsizes." Forty-first International Conference on Machine Learning. 2024.

---

> > ### Comment · Reviewer_tsMk · 2025-08-04
> >
> > Dear Authors,
> >
> > Thanks for your response. While you do clarify my questions, my main concern with a higher score is that I think the methods used are standard, and you can get a much simpler algorithm usign mirror prox or something like that.
> >
> > I owuld therefore like to keep my score.

---

> > > ### Author Response · Authors · 2025-08-04
> > >
> > > Indeed, in our paper, we used the extragradient method, which is the most widely used and standard approach for solving variational inequality problems (VIPs). However, to the best of our knowledge, our work presents the first result of its kind for addressing problems beyond the L-Lipschitz setting. While it may be possible to develop a simpler method in the future (perhaps using Mirror Prox or a related approach, as the reviewer suggested), we believe this does not undermine the contribution of our work.

---

### Official Review · Reviewer_bk47 · 2025-06-25

**Clarity:** 3
**Significance:** 2
**Originality:** 2
**Rating:** 4
**Confidence:** 3

**Summary:**

This paper studies the extragradient method (EG) for variational inequality problems under the assumption that the operator in question satisfies the recently introduced $\alpha$-symmetric $(L_0,L_1)$-Lipschitz condition. Under this assumption, a new step-size rule for EG is proposed. Convergence of EG under various additional assumptions on F (strongly monotone, monotone, and weak Minty condition) are shown. The paper is rounded out by a short experimental section.

**Questions:**

- How easily do your results extend to the true VI problem, where there is a non-trivial constraint set?
 - Could you apply your analysis to, say, the optimistic gradient method?
 - The definition of $\alpha$-symmetric is new to me and very interesting. It feels somehow 'dual' to Holder continuity. Is there a reason for choosing to focus on $\alpha$-symmetry as opposed to Holder continuity in this setting?
 - I'm not familiar with the weak Minty condition. Could you provide some intuition for why this is interesting? Is there an analogue of this condition in the convex optimization world? (like the analogue between (strong) monotonicity and (strong) convexity)?
 - In the proof of Theorem 3.4, in the ninth line, I believe there is a sign error. I think $-\gamma^2_k\|F(x_k) - F(\hat{x}_k)\|^2$ should be $+\gamma^2_k\|F(x_k) - F(\hat{x}_k)\|^2$. It seems this will affect some of the constants in the statement of the theorem. Relatedly, the first 9 lines of the proofs of Theorem 3.4 and Theorem 3.2 are the same, so I suggest you package them into a lemma.
 - Similarly, there is duplication in the first several lines of the proofs of Theorems 3.5 and 3.7 which might benefit from being packed into a lemma.

**Ethical Concerns:**

["NO or VERY MINOR ethics concerns only"]

**Final Justification:**

See discussion below.

**Limitations:**

Yes.

**Paper Formatting Concerns:**

None.

**Quality:**

3

**Strengths And Weaknesses:**

### Strengths
 - Writing is clear and easy to follow
 - Theorems are presented in an easy to digest manner. The proofs are well-structured.
 - Variational inequalities are a topic of perennial importance to the optimization/ML community, so this work is likely to be of interest.
 - I like the extension to "weak Minty" operators, which is not something I've seen before. It's good to see some results under assumptions weaker than monotonicity.

### Weaknesses
 - In my opinion, the biggest weakness of this paper pertains to its scope/significance. Specifically:
	 - The authors focus on unconstrained variational inequalities, which might more appropriately be called root-finding problems. This makes comparisons with prior work, particularly [Vankov 2024] somewhat misleading.
	 - The authors focus only on extragradient. It would be nice to see if this analysis applies to similar methods, e.g. KM iteration, optimistic gradient method.
 - It would also be good to see some discussion of, and comparison to Halpern-style methods [Park & Ryu].
 - Although the new step-size rule is nice, it requires knowing three parameters ($L_0,L_1,\alpha$) of $F$. Step-sizes based on the Lipschitz assumption just require one parameter $L$. This should probably be discussed somewhere.

### Typos etc
 - line 674: "maximum eigenvalue $\sqrt{2}$ " should be $2$.
 - line 690--691: "We add the details of this example to 691 Appendix ??." is probably a typo.
 - lines 707--708 duplicate said in lines 705--706.
 - line 714: Can you change the label "Proposition C.1." --> "Proposition 3.1" ? Same for all Theorem/Proposition statements in the appendix.
 - In the proof of Theorem 3.2: the (7) over the first inequality should be (8). Similarly for the proof of Theorem 3.4
 - In final line of proof of Theorem 3.2 suggest using square brackets $[\ldots ]$ in addition to parentheses $(\ldots)$ as the final factorization is a bit challenging to parse.
 - line 729: I think you want $x=x_{*}$ so that $F(x_{*}) = 0$.  Same for line 761

### References
[Vankov 2024]: _Generalized Smooth Variational Inequalities: Methods with Adaptive Stepsizes_ Vankov _et al_ (ICML2024)

[Park & Ryu]: _Exact Optimal Accelerated Complexity for Fixed-Point Iterations_ Park & Ryu (ICML2022)


-----

After author-reviewer discussion, I raised my score 3 --> 4.

---

> ### Author Rebuttal · Authors · 2025-07-31
>
> We thank the Reviewer bk47, for reading our work carefully and providing us with a valuable review. We respond to the questions and concerns raised by the reviewer below.
>
> ## Weaknesses
> > ....This makes comparisons with prior work, particularly [Vankov 2024] somewhat misleading....
>
> Thanks for the suggestion. We use unconstrained variational inequalities because they are extensively used as terminology in several papers over the last few years. This paper provides an analysis of the extragradient method under $\alpha$-symmetric $(L_0, L_1)$ Lipschitz assumption, and to the best of our knowledge, [1] is the only prior work that addresses this, even if in a different setting (constrained VIPs). We intended to ensure that we provide proper credit and offer a fair comparison with existing literature.
>
> > The authors focus only on extragradient. It would be nice to see if this analysis applies to similar methods, e.g. KM iteration, optimistic gradient method
>
> **We do not view this as a weakness of our work.** Prior to this, even the convergence behavior of the extragradient method under generalized Lipschitz conditions was not fully understood. In this paper, we improve the convergence rate for strongly monotone problems and provide the first analysis for monotone and weak Minty problems under $\alpha$-smmteric $(L_0, L_1)$ Lipschitz assumptions. These results lay the groundwork for extending the analysis to more advanced algorithms such as the optimistic gradient method and KM iteration. We appreciate the reviewer’s suggestion, and exploring such extensions is indeed on our research agenda.
>
> Typically, in the limited conference space, a paper can focus on one algorithm and understand the behavior in depth in different important settings, or focus on a setting and explore the behavior of different algorithms. Having several methods in all different settings can be quite challenging in a limited number of pages with a proper exposition.
>
> > It would also be good to see some discussion of, and comparison to Halpern-style methods [Park & Ryu].
>
> Thank you for the suggestion. We will include a discussion in the updated version comparing our results with the Halpern iteration method [2]. **However, this is not a weakness of our work.** We would like to emphasize that [2] analyzes under the standard L-Lipschitz assumption, whereas our analysis is conducted under the more general $\alpha$-symmetric $(L_0, L_1)$-Lipschitz assumption. Therefore, the settings are not directly comparable.
>
> > Although the new step-size rule is nice, it requires knowing three parameters (L_0, L_1, \alpha) of F....
>
> Yes, Reviewer bk47 is correct in observing that solving this more general class of problems requires knowledge of three parameters, in contrast to standard L-Lipschitz settings. **However, this is not a limitation of our work.** The problem class we consider is significantly more general and complex, and thus naturally demands more information for effective algorithm design and analysis. That said, developing methods that do not require prior knowledge of these parameters, i.e., tuning-free algorithms for this broader class of problems, is an exciting direction for future research.
>
> > Typos
>
> We appreciate the reviewer for carefully reviewing the appendix and identifying the typos. We agree with all the typos mentioned and will make the necessary corrections in the updated version.
>
> ## Questions
>
> > How easily do your results extend to the true VI problem, where there is a non-trivial constraint set?
>
> Extending the algorithm to handle non-trivial constraint sets would require significant modifications. In particular, solving the constrained version would involve incorporating projections in both the extrapolation and update steps. Additionally, the presence of projections would also affect the step size choices in that scenario.
>
> For these reasons, we focus on the unconstrained setting in this work, as it lays the theoretical groundwork and provides key insights that can facilitate the analysis of more complex algorithms for constrained problems in the future. It is definitely on our to-do list to extend the analysis to a constrained setting in the future.
>
> > Could you apply your analysis to, say, the optimistic gradient method?
>
> As mentioned earlier, this is one of the directions we intend to explore in future work. We firmly believe that our proof technique can be extended to analyze the optimistic gradient method and KM iterations under the $\alpha$-symmteric $(L_0, L_1)$ Lipschitz assumption.
>
> > Is there a reason for choosing to focus on ...
>
> Thank you for the question. The $\alpha$-symmetric $(L_0, L_1)$-Lipschitz assumption was introduced because many deep learning problems appear to satisfy this condition empirically in the minimization setting with the operator replaced by the gradient of the objective function. We provide a detailed discussion motivating this assumption between lines 32–65 in the introduction, along with illustrative examples in Section 2.2. Our analysis under this assumption is purely motivated by its relevance and empirical support in practical scenarios.
>
> > I'm not familiar with the weak Minty condition. Could you provide some intuition for why this is interesting?..
>
> The weak Minty condition was first introduced in [4]. General unconstrained variational inequality problems are known to be computationally intractable, which has motivated the search for structural assumptions that allow for meaningful convergence guarantees. Among the known conditions, the weak Minty condition is currently the most relaxed assumption under which convergence of algorithms can be established.
>
> > In the proof of Theorem 3.4, in the ninth line, I believe there is a sign error.
>
> Thank you for the pointer. The expressions in lines 9 and 10 contain a typo, which is corrected in line 11 of the same proof. As a result, this does not affect any of the constants appearing in the final result of Theorem 3.4. We will correct the typo in the updated version of the paper.
>
> > ..so I suggest you package them into a lemma...
>
> Thanks for the suggestion. We will incorporate this in the revised version of the paper.
>
> **We hope we have addressed all the questions and concerns of the reviewer. If you agree, please consider increasing your score. If not, please let us know so we can respond accordingly.**
>
> ## References
>
> [1] Vankov, Daniil, Angelia Nedich, and Lalitha Sankar. "Generalized smooth variational inequalities: Methods with adaptive stepsizes." Forty-first International Conference on Machine Learning. 2024.
>
> [2] Park, Jisun, and Ernest K. Ryu. "Exact optimal accelerated complexity for fixed-point iterations." International Conference on Machine Learning. PMLR, 2022.
>
> [3] Zhang, Jingzhao, et al. "Why gradient clipping accelerates training: A theoretical justification for adaptivity." arXiv preprint arXiv:1905.11881 (2019).

---

> > ### Comment · Reviewer_bk47 · 2025-08-04
> > **Response to Author Response**
> >
> > Thanks to the authors for their comprehensive response.
> >
> > 1. *This makes comparisons with prior work, particularly [Vankov 2024] somewhat misleading*
> >
> > **Response:** Glancing at other reviewers comments, it seems I am not the only one who finds the term "unconstrained variational inequality" strange. For the benefit of the AC and other reviewers, let me mention that after a little searching I found a few other papers that use the same teminology (e.g. [1,2]). Nonetheless, I still find the comparison with [Vankov 2024] somewhat misleading. Yes, your analysis is tighter in the unconstrained case, but their theorem is more general.
> >
> > 2. *Could you apply your analysis to, say, the optimistic gradient method?*
> >
> > **Response:** I agree. This would be out of scope for this work.
> >
> > 3. *It would also be good to see some discussion of, and comparison to Halpern-style methods [Park & Ryu]*
> >
> > **Response:** Thanks. Yup, I know that they work in the L-Lipschitz case. My point is that because unconstrained VIs are fixed point problems, the literature on fixed point problems should be referenced.
> >
> > 4. *Although the new step-size rule is nice, it requires knowing three parameters*
> >
> > **Response:** Agreed. I interpreted  "adaptive" as being synonymous with "parameter-free", but I agree that it can also mean "adaptive to the prior iterates seen by the algorithm".
> >
> > 5. *so I suggest you package them into a lemma.*
> >
> > **Response:**  Thanks for being receptive to this.
> >
> > In light of the above, I will raise my score to 4. The main reason I'm not willing to go higher is the lack of analysis of constrained VIs. Thanks again to the authors for engaging in the review process!
> >
> > ### References:
> > [1] Gorbunov, Eduard, Nicolas Loizou, and Gauthier Gidel. "Extragradient method: O (1/k) last-iterate convergence for monotone variational inequalities and connections with cocoercivity." In International Conference on Artificial Intelligence and Statistics, pp. 366-402. PMLR, 2022.
> >
> > [2] Diakonikolas, Jelena, Constantinos Daskalakis, and Michael I. Jordan. "Efficient methods for structured nonconvex-nonconcave min-max optimization." In International Conference on Artificial Intelligence and Statistics, pp. 2746-2754. PMLR, 2021.

---

### Official Review · Reviewer_wYYi · 2025-07-01

**Clarity:** 4
**Significance:** 3
**Originality:** 2
**Rating:** 5
**Confidence:** 3

**Summary:**

This paper studies the Extragradient (EG) method under the $(L_0, L_1)$-Lipschitz condition. This is a generalized and perhaps more practical condition compared to the traditional $L$-Lipschitz assumption commonly used when studying the EG method. The paper provides sublinear convergence under the monotone assumption, linear convergence under the strongly monotone assumption, and local convergence under the weak Minty (non-monotone) assumption. Together with the theoretical analysis, the paper presents numerical experiments that demonstrate the effectiveness of the proposed step sizes.

**Questions:**

- **Q1.** In lines 185–187, the authors state that the results in their paper recover the result in [Gorbunov et al., 2022b]. However, as the title of the cited paper suggests, it provides 'last-iterate' convergence, rather than an ergodic convergence rate. The reviewer is confused, as the results provided in the submitted paper are ergodic convergence rates (Theorem 3.5). Can the authors clarify this point?

- **Q2.**  Is there a significant technical difficulty compared to the function minimization problem or the standard $L$-Lipschitz setup that the authors would like to highlight, which the reviewer might be overlooking?

**Ethical Concerns:**

["NO or VERY MINOR ethics concerns only"]

**Final Justification:**

I believe that minimizing the norm of a monotone and $L$-Lipschitz continuous operator is one of the standard setups in optimization theory. I view this as a classic setup, similar to the minimization of an $L$-smooth convex function. Moreover, I think the Extragradient (EG) method in this setup is as classic as Gradient Descent (GD) is in $L$-smooth convex minimization. And I view this paper as a theoretical work.

I believe it is meaningful to question what the possible applications of a setup might be. However, I am not sure that this point can be the main criterion for judging this paper. The monotone and $L$-Lipschitz operator setup is already a classic framework, and this paper considers a more advanced setup. Additionally, I think this paper provides good examples highlighting improvements over the existing $L$-Lipschitz monotone setup. Criticizing this paper based on the usefulness of the setup sounds to me like rejecting a large portion of the optimization theory field. That said, I understand that some may have a lower preference for theory papers. Nonetheless, I value theoretical contributions and therefore vote to accept.

And, as mentioned in my review, to the best of my understanding, some reviewers seem to have concerns regarding the comparison of this work with PPM or Halpern. Carefully speaking, I think that criticism is not appropriate. To the best of my knowledge, relating methods in fixed-point problems to the monotone operator setup requires considering a proper equivalence that involves the proximal operation. Therefore, I believe that if we want to relate fixed-point problems to monotone operator norm minimization, it should be viewed as an implicit setup. PPM (Proximal Point Method) indeed requires the proximal operation, and Halpern is a technique considered to improve the convergence of methods for fixed-point problems (at least in the referenced work). Just as we can't say “GD is slower than PPM” because they belong to different setups (when considering function value minimization), I think it is difficult to make a direct comparison between EG and PPM or Halpern.

**Limitations:**

Yes.

**Paper Formatting Concerns:**

No issues.

**Quality:**

4

**Strengths And Weaknesses:**

**Strengths**

- This paper starts by introducing the motivation for the $(L_0, L_1)$-Lipschitz condition they are considering, and they did a good job on this point. To the best of this reviewer's understanding, they needed to come up with new observations related to EG for this purpose, and the reviewer believes these are meaningful observations worth sharing. They also provided persuasive examples.

- They provide theoretical convergence analyses under popular assumptions—monotone, strongly monotone, and weak Minty—commonly considered for the EG method. In the context that these results are generalizations of results already established in function minimization problems or setups with only an $L$-Lipschitz condition, they could be seen as somewhat expected. Nonetheless, the reviewer believes they still contribute meaningfully and represent work that needed to be done for the field.

- The paper provides meaningful equivalent formulations of the core assumptions. They seem useful in carrying out the analyses and are also meaningful on their own.


**Weakness**

- **W1.** To the best of this reviewer's understanding, most of the techniques used in the theoretical analyses are extended versions of the techniques used for function minimization or merely the $L$-Lipschitz setup. The extension may not be trivial, but it seems like something that could be expected. However, this reviewer does not consider this a weakness, though it may have contributed to a slightly more moderate overall assessment.

---

> ### Author Rebuttal · Authors · 2025-07-31
>
> We thank the reviewer for the positive evaluation of our work. We address the questions and concerns raised by the reviewer below.
>
> ## Weaknesses
>
> > ...The extension may not be trivial, but it seems like something that could be expected.... + ....  Is there a significant technical difficulty compared to the function minimization problem or the standard Lipschitz ..]
>
> Thank you for the question. Compared to the literature in the minimization setting, our submission solves unconstrained variational inequality problems with the extragradient method. Our submission has no notion of functional values in contrast to the minimization analysis, and moreover, the extragradient method has an extrapolation step, which introduces further challenges. The proof technique of analyzing the extragradient methods is very different from the methods used in minimization problems.
>
> Compared to the analysis of extragradient method under Lipschitz assumption, this new assumption introduces further challenges. In the submission we have, the $\alpha$-symmetric $(L_0, L_1)$-Lipschitz assumption, which involves a more general bound of the form $L_0 + L_1 \max_{\theta \in [0,1]} \| F(\theta x + (1 - \theta) y) \|^{\alpha}$ instead of the standard Lipschitz constant L. This generalization makes the analysis significantly more challenging. For example, in the standard L-Lipschitz case, one can easily bound $\| F(x_k) - F(\hat{x}_k) \|^2$ by $L^2 \| x_k - \hat{x}_k \|^2$, which appears in line 9 of the proof of Theorem 3.2 (Appendix page 24). However, under the $\alpha$-symmetric $(L_0, L_1)$-Lipschitz assumption, it introduces a much involved term, which complicates the analysis.
>
> This difficulty motivates the use of an adaptive step size of the form $\eta / (L_0 + L_1 \|F(x_k)\|^{\alpha})$,
> which is better suited for this setting. Moreover, throughout the analysis, several steps require us to carefully use the structure of the $\alpha$-symmetric Lipschitz condition and develop proof techniques that differ from those used under the standard Lipschitz assumption.
>
> ## Questions
>
> > ...the authors state that the results in their paper recover the result in [Gorbunov et al., 2022b]. However, as the title of the cited paper suggests...]** Thank you for the question. Although [1] only emphasizes last-iterate convergence in its title, which is indeed their main contribution, the paper also provides an ergodic convergence rate, as stated in Theorem 3.1. We hope this clarifies the confusion, and we will include this clarification in the updated version of the submission.
>
> **We hope we have addressed all the questions and concerns of the reviewer. If not, please let us know so we can respond accordingly.**
>
> ## References
>
> [1] Gorbunov, Eduard, Nicolas Loizou, and Gauthier Gidel. "Extragradient method: O (1/k) last-iterate convergence for monotone variational inequalities and connections with cocoercivity." International Conference on Artificial Intelligence and Statistics. PMLR, 2022.

---

> > ### Comment · Reviewer_wYYi · 2025-08-05
> >
> > This reviewer appreciate the authors' respond.
> >
> > It seems that there are some concerns about comparing the content of this paper to PPM [1,2] or its accelerated variants [3, 4, 5]. This reviewer disagrees with such concerns, since, to the best of the reviewer's knowledge, these methods are based on proximal operations (and so implicit methods), while this paper considers EG, which is an explicit method. Putting aside the fact that this paper considers a new setup (thus, comparing with the results in the setups in prior works is somewhat unfair), this reviewer believes that the setups for PPM and its accelerated variants are different enough that they should be considered as a different track. It is true that PPM and EG have a relation, so it would be interesting to have some discussion if possible and perhaps worth mentioning in the prior work, but this reviewer believes that any kind of comparison between PPM (or its accelerated variants) and this paper (or the absence of such a comparison) should not be considered a weakness of this paper. And if someone wants to find the relation (or equivalence) between the PPM setup and fixed point problems, this reviewer believes that they can look at the explanation in the prior works, for example, [§2, 5] or [§3.0, 6].
> >
> > This reviewer believes that EG is a representative baseline method for the $L$-Lipschitz monotone operator setup, and thus exploring the new setup considered in the paper using EG is a meaningful starting point. As a side note, this reviewer wonders whether the authors believe that accelerated variants [7, 8] of EG could also be extended to the $(L_0, L_1)$-Lipschitz setup.
> >
> > To reorganize this reviewer's opinion, this reviewer believes that the paper gave a good motivation for the $(L_0, L_1)$-Lipschitz setup. As it analyzes a famous and important method that has been studied from various perspectives, the analysis may resemble some analysis in prior work, but the reviewer believes that the extension to the $(L_0, L_1)$-Lipschitz setup is nontrivial, as it requires using different inequalities due to the novel setup. Therefore, I will maintain my score.
> >
> >
> > - [1] B. Martinet. Régularisation d’inéquations variationnelles par approximations successives. Revue Française d’Informatique et de Recherche Opérationnelle, Série Rouge, 4(3):154–158, 1970.
> >
> > - [2] Rockafellar, R. T. Monotone operators and the proximal point algorithm. SIAM Journal on Control and Optimization, 14(5):877–898, 1976.
> >
> > - [3] D. Kim. Accelerated proximal point method for maximally monotone operators. Mathematical Programming, 190(1–2):57–87, 2021.
> >
> > - [4] F. Lieder. On the convergence rate of the Halpern-iteration. Optimization Letters, 15(2): 405–418, 2021.
> >
> > - [5] J. Park and E. K. Ryu. Exact optimal accelerated complexity for fixed-point iterations. International Conference on Machine Learning, 2022.
> >
> > - [6] T. Yoon, J. Kim, J. J. Suh, and E. K. Ryu. Optimal acceleration for minimax and fixed-point
> > problems is not unique. International Conference on Machine Learning, 2024.
> >
> > - [7] T. Yoon and E. K. Ryu. Accelerated algorithms for smooth convex-concave minimax problems with $O(1/k^2)$ rate on squared gradient norm. International Conference on Machine Learning, 2021.
> >
> > - [8] S. Lee and D. Kim. Fast extra gradient methods for smooth structured nonconvex-nonconcave minimax problems. Neural Information Processing Systems, 2021.

---

### Official Review · Reviewer_Fpia · 2025-07-02

**Clarity:** 4
**Significance:** 4
**Originality:** 4
**Rating:** 5
**Confidence:** 4

**Summary:**

This work analyzes the convergence rate of extragradient method under the α-symmetric (L0,L1)-Lipschitz continuity, a smoothness condition that generalizes both the classical Lipschitz assumption and the (L0,L1)- Lipschitz framework (where α = 1) introduced by [1]. The authors claim that the classical global L-smooth condition is often too restrictive for certain optimization problems, and instead propose an adaptive method that performs effectively under the α-symmetric assumption. In Section 3, they establish convergence guarantees under three different assumptions on the monotone operator F, which matches known upper bounds in certain extreme cases. Specifically, their Lyapunov-style analysis results in an adaptive stepsize that ensures the distance to the solution decreases across iterations. Finally, the authors present experimental results that support and validate their theoretical findings.

**Questions:**

In addition to the mentioned weaknesses, I have some questions:
• What are the problem parameters (L0,L1, α) of equation (19)? Does it matches the parameteres that lead to good performance in Figure 3c?
• The choice of stepsize differs significantly depending on whether α = 1 or not. However, in practice, α is typically unknown. Given only a problem instance, how should one select an appropriate stepsize? Would it be necessary to sweep over possible α values or estimate α in advance? Is it possible to design an algorithm that works reliably for all values of α without prior knowledge?

**Ethical Concerns:**

["NO or VERY MINOR ethics concerns only"]

**Final Justification:**

I appreciate the contribution of this paper, and the weaknesses that are brought up are minor. I vote to accept this paper.

**Limitations:**

yes

**Quality:**

4

**Strengths And Weaknesses:**

Strengths
 - The results are both strong and straightforward. The authors establish convergence guarantees across all six scenarios by varying the parameter α and the monotonicity assumptions on the operator. They successfully extend the analysis of prior works (e.g., [2]) to the extragradient method. Most of the convergence results are novel and contribute meaningfully to the theory of the extragradient method.
 - In my opinion, this paper is written in a very clear and reader-friendly manner. Six different cases in Section 3 as well as their proof in Appendix are presented in a nice parallel structure, making both the theory and its proof easy to follow. This also makes the paper a valuable reference for future research upon these findings.
- The class of operators considered in this work is broad enough to encompass many practical problems. Combined with the significant role of the extragradient method in optimization, these results broaden its applicability to a wide range of optimization problems. The broad nature of this work opens up future research directions, such as extensions to other momentum-based methods or the development of alternative stepsize schemes.


Weaknesses
 -  The proposed method is not fully “adaptive,” as the step sizes still depend on global problem parameters (L0,L1, α). I understand that the primary goal of this work is not to develop an adaptive method, but to establish convergence guarantees under a new problem assumption. However, the contribution could have been stronger if the method had incorporated local estimates of global parameters. For example, [3] proposes an approach that estimates L0 locally. In fact, the importance of choosing proper (L0,L1) is also evident in the experiments, as the performance appears to vary depending on their choice.
 - As previously mentioned, the core content is well-structured. However, the paper lacks a dedicated prior work section, either in the main text or the appendix. As a result, it is difficult to clearly understand how this work relates to existing literature or how it compares to previous methods developed under similar assumptions. It seems that most of related works cited in the main text appear to focus on the (L0,L1)-smooth and α-symmetric assumption, rather than on method-level analyses.
 - I found the experimental section to be somewhat limited. While the paper is primarily theoretical—so this may not be a major concern—it seems that most experiments were conducted on synthetic data. Including experiments on more practical problems or scaling up the current ones could help further demonstrate the method’s effectiveness in realistic settings.


Typos
• Line 3: Variational inequalities problems → Variational inequality problems
• Line 45 : Memorys → Memories
• For theorems 3.2,3.4,3.5,3.6,3.7,and 3.9: satisfy → satisfies
• x∗ not defined in Assumption 1.3.

---

> ### Author Rebuttal · Authors · 2025-07-31
>
> We thank the reviewer for a positive evaluation and for providing us with valuable feedback. We address the questions and concerns raised by the reviewer below.
>
> ## Weaknesses
>
> > .... For example, [3] proposes an approach that estimates L0 locally....
>
> We are not sure which paper the reviewer is referring to with [3] in this context. We agree that it would be highly valuable to develop a fully adaptive algorithm for this general class of problems. However, we would like to emphasize that even the analysis of such problems, assuming knowledge of $(L_0, L_1, \alpha)$, has been missing in several important settings. Our current work addresses this gap, and in future work, we aim to extend the analysis further and explore techniques for estimating these constants in practice.
>
> > .... It seems that most of related works cited in the main text appear to focus on the (L0,L1)-smooth and α-symmetric assumption, rather than on method-level analyses...
>
> Thank you for the suggestion. Indeed, our related work section is primarily organized around the assumptions rather than the methods. This is because, to the best of our knowledge, there are no prior works analyzing algorithms under the $\alpha$-symmetric $(L_0, L_1)$-Lipschitz assumption, except for [1], which we have referenced multiple times.
> That said, we agree it would be helpful to include a broader discussion. In the revised version, we can easily add a section in the appendix reviewing methods developed in the minimization setting that are analyzed under the $\alpha$-symmetric $(L_0, L_1)$-smoothness assumption.
>
> > .... While the paper is primarily theoretical—so this may not be a major concern—it seems that most experiments were conducted on synthetic data. ...
>
> Yes, the primary focus of this work is to strengthen the theoretical foundations for this general setting. Our current experiments are designed to demonstrate the advantage of using step sizes of the form $ 1 / (c_0 + c_1 \| F(x_k) \|^{\alpha}) $ over constant step sizes. We will include additional experiments in the updated version to validate the effectiveness of our algorithm further.
>
> > Typos
>
> We thank the reviewer for carefully reading our work and pointing out the typos. We will fix them in the updated version of the paper.
>
> ## Questions
>
> > What are the problem parameters (L0,L1, α) of equation (19)? ...
>
> In Appendix D2, we explained a way to compute these values in a simplified setting.
>
> > The choice of stepsize differs significantly depending on whether α = 1 or not. However, in practice, α is typically unknown. Given only a problem instance, how should one select an appropriate stepsize?
>
> At present, for general problems, one can only consider sweeping over different values of $ \alpha$ and check which one performs best. However, our future goal is to develop methods for estimating the parameter $\alpha$, which we believe will be a significant contribution to the literature.
>
> **We hope we have addressed all the questions and concerns of the reviewer. If not, please let us know so we can respond accordingly.**
>
> ## References
>
> [1] Vankov, Daniil, Angelia Nedich, and Lalitha Sankar. "Generalized smooth variational inequalities: Methods with adaptive stepsizes." Forty-first International Conference on Machine Learning. 2024.

---

> > ### Comment · Reviewer_Fpia · 2025-08-05
> >
> > I appreciate the contribution of this paper, and the weaknesses that are brought up are minor. I vote to accept this paper.

---

### Official Review · Reviewer_4hmy · 2025-07-02

**Clarity:** 3
**Significance:** 2
**Originality:** 2
**Rating:** 4
**Confidence:** 4

**Summary:**

This paper analyzes the extragradient method for min-max and variational inequality problems under relaxed assumptions. In particular, the common (and often strong) L-Lipschitz continuous assumption is relaxed to an $\alpha$-symmetric $(L_0, L_1)$-Lipschitz continuity assumption, which allows the Lipschitz constant to scale with the operator norm. Under this more general condition, convergence analysis is given for strongly monotone and monotone variational inequalities. In addition, a local convergence rate is given for variational inequalities which satisfy the weak Minty variational inequality condition. Experimental results are provided that validate the effectiveness of a newly proposed adaptive stepsize based on $(L_0, L_1)$-Lipschitz continuity.

**Questions:**

Is the exponential dependence on $\|x-y\|$ tight in Proposition 3.1? Are there any conditions under which this exponential dependence may vanish (other than $L_1 = 0$)? My curiosity stems from the fact that this appears to cause major issues throughout the paper, such as for the strongly monotone case (where this only shows up in a constant now, but still persists), but especially in the weak Minty variational inequality case, where the neighborhood of convergence is directly dependent on this exponential dependence, which seriously restricts the applicability of the result. If it was possible to find reasonable conditions under which this exponential dependence vanished or was more controlled, it could significantly improve my evaluation score, as it would significantly improve the result in the weak Minty case.

For the weak Minty experiment, do the methods struggle when started far from the solution? I am curious if the authors observe the issue of only obtaining local convergence using their adaptive stepsize or if this is just an artifact of the analysis.

In the theoretical results, the adaptive stepsize is dependent on the constant $\alpha$ in the $\alpha$-symmetric assumption, however in the numerical experiments, $\alpha = 1$ is used throughout. Do the experimental results change much when $\alpha$ varies away from 1? In particular, for the monotone case, the introduction suggests that this problem should use an $\alpha < 1$, which partially motivates the assumption, but this is not used in the experiments, which raises questions about the value behind this relaxation of $\alpha$.

**Ethical Concerns:**

["NO or VERY MINOR ethics concerns only"]

**Final Justification:**

I have decided to improve my score to a borderline accept due to the author's responses regarding some concerns I had related to the exponential dependence on the initialization which is present throughout the paper. I stopped short of a full accept as I find the motivation for root finding problems under these more general assumptions somewhat lacking and am concerned about the limitations of the local convergence result for weak Minty problems.

**Limitations:**

Yes.

**Paper Formatting Concerns:**

No.

**Quality:**

3

**Strengths And Weaknesses:**

Strengths: This paper analyzes the highly popular extragradient method for variational inequalities under a relaxed set of assumptions. The convergence results improve upon known results in the strongly monotone case and are the first convergence results under the $(L_0, L_1)$-Lipschitz continuity condition for monotone operators. In addition, the work provides the first convergence results for weak Minty variational inequalities, though the convergence is only local. These results are proven when a novel adaptive stepsize is used. This stepsize strategy is validated numerically on a number of simulated experiments.

Weaknesses: In the strongly monotone case, the improved convergence rate comes from an argument that shows that the exponential dependence on the distance of the initial point to the solution can be moved into a constant factor, as opposed to multiplying a $\mathcal{O}(\log(\epsilon^{-1}))$ factor (where $\epsilon$ is the optimization tolerance), as it is in prior work. While this is an improvement, it is a relatively minor change that is only relevant for extremely small values of $\epsilon$. In addition, the convergence rate for weak Minty variational inequalities requires that $\rho$ is sufficiently small relative to $L_1 e^{\|x_0-x_*\|}$, which essentially requires that either $\|x_0-x_*\|$ is extremely small for any non-monotone problem or $L_1 = 0$, in which case the generalization to $(L_0, L_1)$ smoothness is irrelevant.

Quality: The focus on extending convergence results for extragradient methods beyond is a reasonable focus, as current results rely on restrictive assumptions that may not hold in practical applications. The relaxed assumptions provide a nice generalization and the results appear to be technically sound. However, a number of the results are only minor improvements over previous works or require extremely strong assumptions that limit their applicability.

Clarity: The paper is well written and well organized. There are a number of examples given to motivate the modified assumptions and the relationship with prior work is clearly stated. The limitations of the work could be stated more clearly, as they are spread throughout the main convergence results. It would also be nice if there was a brief discussion or conclusion section added to this work discussing potential future directions.

Significance: This paper provides some of the first results for extragradient methods under the $(L_0, L_1)$ smoothness assumption. However, the strongly monotone result is only a mild improvement on prior work and the weak Minty variational case only applies to an extremely tight neighborhood of the solution. Therefore, the main contributions are mostly limited to the monotone case.

Novelty: The main change from prior work is the use of the $(L_0, L_1)$ smoothness assumption and applying this to the monotone and weak Minty variational inequality cases. While this is novel, it feels more incremental than a major new direction of research.

---

> ### Author Rebuttal · Authors · 2025-07-31
>
> We thank the reviewer for their insightful comments and suggestions. Our detailed responses are provided below.
>
> ## Weaknesses
> **[In the strongly monotone .... it is a relatively minor change.....]** We respectfully disagree with the reviewer’s comment. In optimization, practical interest typically lies in obtaining high-accuracy solutions, i.e., small values of $\epsilon$. In such cases, any significant constant multiplying the $ \log(1/\epsilon)$ term can have a substantial impact on the convergence rate.
>
> To illustrate this, consider the well-known comparison between Gradient Descent and Accelerated Gradient Descent for smooth, strongly convex functions. Gradient Descent achieves a rate of $O\left(\frac{L}{\mu} \log(1/\epsilon)\right)$, while Accelerated Gradient Descent achieves $O\left(\sqrt{\frac{L}{\mu}} \log(1/\epsilon)\right)$. The difference between $\frac{L}{\mu}$ and $\sqrt{\frac{L}{\mu}}$ leads to huge improvements in performance, especially as higher accuracy is desired.
>
> In our setting, removing the exponential dependence on the initial distance to the solution from the convergence rate and moving it into a constant factor is a meaningful theoretical improvement. The exponential factor can be arbitrarily large depending on the initialization, and its removal explains the performance of the algorithm in practice. Moreover, in experiments, we show improvement with our proposed step size compared to the prior work (check figure 3a, 3b). We hope this clarifies the significance of our result.
>
> > .... for weak Minty variational inequalities, though the convergence is only local....
>
> We agree that the assumption on $\rho$ for weak Minty problems is currently restrictive. However,  this is the first analysis conducted under the $\alpha$-symmetric $(L_0, L_1)$ Lipschitz assumption for this class of problems, and we believe it lays the groundwork for further theoretical developments in this direction.
>
> > However, a number of the results are only minor improvements....] + [it feels more incremental than a major new direction of research.
>
> We disagree with this statement. As mentioned in the previous response, the improvement in the strongly monotone setting is not minor. Moreover, we provide the first analysis in a monotone and weak Minty setting.
>
> > It would also be nice if there was a brief discussion or conclusion section added to this work discussing potential future directions.
>
> We were unable to do that due to space constraints. Thanks for the suggestion, and we will include it in the updated version.
>
> ## Questions
>
> > .... Are there any conditions under which this exponential dependence may vanish ...
>
> Thank you for the insightful question. We would like to clarify that the convergence analysis of our proposed algorithms fundamentally relies on the equivalent formulation in Proposition 3.1. This formulation introduces the exponential dependence you noted, and we currently do not believe it is possible to eliminate this term entirely from the convergence rate. We can only improve how this term appears in the rate and minimize its impact on the convergence rate.
>
> As mentioned in our earlier response, in the strongly monotone case, we show that the exponential dependence no longer appears as a multiplicative factor to the $O(\log(1/\epsilon))$ term, but rather as an additive constant. As explained earlier, this is not a minor improvement. We agree with the reviewer that, in the weak Minty setting, the condition on $\rho$ is currently restrictive. However, we believe that the dependence of $\rho$ on the exponential term can be improved even in the weak minty setting, and we view this as an important direction for future research.
>
> > For the weak Minty experiment, do the methods struggle when started far from the solution?...
>
> That is an excellent suggestion. We can easily include such experimental evaluation in the camera-ready version of our work to visually verify our theoretical findings.
>
> > In the theoretical results, the adaptive stepsize is dependent on the constant $\alpha$ in the $\alpha$-symmetric assumption, however in the numerical experiments, $\alpha = 1$ is used throughout....
>
> Indeed, for our proposed experiments, we used $\alpha = 1$ as all problems satisfy the $1$-symmetric $(L_0, L_1)$ assumption. Let us highlight that the primary goal of our experiments is to demonstrate the benefit of using the adaptive step size $1/(c_0 + c_1 \|F(x_k)\|^{\alpha})$ over the constant step size typically used in the extragradient method. In our experiments, we show that our proposed step-size selection with $\alpha = 1$ yields better performance than the constant step size. That said, we agree that exploring different values of $ \alpha$ may lead to further improvements. We can easily include additional experiments to investigate the impact of varying $\alpha$ in the updated version of our paper.
>
> **We hope we have addressed all the questions and concerns raised by Reviewer 4hmy. We believe that most concerns raised can be easily addressed in the updated version of our work (not major issues). If you find our clarifications satisfactory, please consider increasing your score. If not, we would appreciate the opportunity to respond further.**

---

> > ### Comment · Reviewer_4hmy · 2025-08-05
> >
> > [In the strongly monotone .... it is a relatively minor change.....] In retrospect, I believe that I misread this section a bit and do agree that this change is more significant than I made it out to be. Apologies for this comment in the original review.
> >
> > [.... for weak Minty variational inequalities, though the convergence is only local....] I still find this to be a bit unsatisfying, as I am worried about the exponential dependence on the radius of convergence, though I understand that the weak Minty setting is quite a bit more difficult than the monotone case.
> >
> > [.... Are there any conditions under which this exponential dependence may vanish ...] With respect to this, considering how crucial this exponential dependence is in the results throughout this work, are there alternative conditions to $\alpha$-symmetric $(L_0, L_1)$ (e.g. Holder smooth) that are relaxations of the standard L-Lipschitz continuity used in most works that would not inevitably lead to such an exponential dependence? I understand this is likely outside the scope of this work, but some discussion of other relaxations of L-Lipschitz continuity in root finding problems is likely warranted [1,2].
> >
> > [For the weak Minty experiment, do the methods struggle when started far from the solution?...] Great, thank you. I'd be very curious to see the results.
> >
> > [In the theoretical results, the adaptive stepsize is dependent on the constant $\alpha$ in the $\alpha$-symmetric assumption, however in the numerical experiments, $\alpha = 1$ is used throughout....] Sure, but I would be interested in experiments where this does not hold. As it stands, the $\alpha < 1$ case is mostly motivated via the plot in Section 1 but it's unclear how often this would play a role in practical settings. This is related to concerns mentioned by other reviewers and the AC about the applications of root finding problems. While I agree that root finding problems are common in the literature, are there many known examples where L-Lipschitz continuity (or Holder continuity) does not hold?
> >
> > Overall, I do believe that the response has satisfied some of my concerns and will be increasing my score to reflect this. Thank you for the time and effort spend on the response.
> >
> > [1] Dang, Cong D., and Guanghui Lan. "On the convergence properties of non-euclidean extragradient methods for variational inequalities with generalized monotone operators." Computational Optimization and applications 60.2 (2015): 277-310.
> > [2] Stonyakin, Fedor, et al. "Generalized mirror prox algorithm for monotone variational inequalities: Universality and inexact oracle." Journal of Optimization Theory and Applications 194.3 (2022): 988-1013.

---

> ### Author Response · Authors · 2025-08-09
>
> Dear Reviewer,
>
>
> Thanks for the follow-up question and for the positive evaluation of our work.
>
> > While I agree that root finding problems are common in the literature, are there many known examples where L-Lipschitz continuity (or Holder continuity) does not hold?
>
>
> The AC has already reached out to us, mentioning your question and some follow-up clarification.
>
>
> Please have a look at the long response to the AC. There, we explain that several minimization and min-max problems are not L-Lipschitz to motivate our new condition, and we point out some basic constructed examples of root finding problems, already included in our paper, that are not L-Lipschitz but that satisfy our $(L_0, L_1)$-Lipschitz condition (to justify further the need for our new condition).
>
>
> > are there alternative conditions to $\alpha$ -symmetric $(L_0, L_1)$  (e.g. Holder smooth) that are relaxations of the standard L-Lipschitz continuity used in most works that would not inevitably lead to such an exponential dependence? I understand this is likely outside the scope of this work, but some discussion of other relaxations of L-Lipschitz continuity in root finding problems is likely warranted
>
>
> Thank you for the suggestion.  We will make sure to include such a comparison in the camera-ready version of our work.
>
>
> Thanks again for the support of our work.
>
>
> Best,
> Authors

---

### Comment · Area_Chair_TcHu · 2025-08-04
**Question to the authors**

Dear Authors,

Thanks for your rebuttal! Following-up on a question by bk47, can you please provide some motivating examples for root-finding problems, coming from applications? I mean: what would be applications of this unconstrained template, apart from the toy examples considered in the submission?

I also agree with Reviewer bk47 that for unconstrained problems, the terminology VI is misleading -- even if other papers in the literature may have been using this convention. I recommend avoiding this too.

Thanks,
AC

---

> ### Author Response · Authors · 2025-08-05
>
> Dear AC,
>
> Thank you for your engagement and for asking a valuable question.
>
> >  some motivating examples for root-finding problems, coming from applications
>
> Finding the saddle point of an unconstrained minimization problem $\min_{x \in \mathbb{R}^d} f(x)$ is a simple example of a root-finding problem, where the operator is given by $F = \nabla f$.
>
> Beyond minimization, any unconstrained smooth min-max optimization problem can be formulated as a root-finding problem. Consider the unconstrained problem $\min_u \max_v L(u, v)$, where $L$ is differentiable in both variables but may be nonconvex in $u$ (for fixed $v$) and nonconcave in $v$ (for fixed $u$). Defining $x = (u, v)$ and $F(x) = (\nabla_u L(u, v), − \nabla_v L(u, v))$, the first-order optimality condition becomes
> $$
> (\nabla_u L(u_*, v_*), − \nabla_v L(u_*, v_*)) =  (0, 0) \iff F(x_*) = 0.
> $$
> which is a root-finding problem. Such formulations arise in many practical settings, including:
> - Reinforcement Learning (equation 10 in [1]),
> - Generative Adversarial Network training ([2], equation 1 in [3]),
> - Robust Least Squares (equation 8 in [4]), and
> - Constrained minimization via Lagrangian formulation [5].
>
> Apart from unconstrained smooth min-max problems, root-finding problems are also important for $N$-player games (section 3.1 in [6]), in particular for multi-agent reinforcement learning (equation 4.5 in [7]). Additionally, fixed-point problems of the form “find $x_*$ such that $x_* = T(x_*)$,” which are common across various fields, can be reformulated as root-finding problems by defining $F(x) = x - T(x)$.
>
> > I also agree with Reviewer bk47 that for unconstrained problems, the terminology VI is misleading
>
> We adopted the terminology of unconstrained variational inequality problems (VIPs) following prior works in the literature [8, 9] (also noted by Reviewer bk47 in their last response). However, we are happy to revise the introduction to emphasize the root-finding perspective instead, and we are willing to update the title accordingly.
>
>
> ## References
>
> [1] Du, Simon S., et al. "Stochastic variance reduction methods for policy evaluation." International conference on machine learning. PMLR, 2017.
>
> [2] Goodfellow, Ian J., et al. "Generative adversarial nets." Advances in neural information processing systems 27 (2014).
>
> [3] Daskalakis, Constantinos, et al. "Training gans with optimism." arXiv preprint arXiv:1711.00141 (2017).
>
> [4] Yang, Junchi, Negar Kiyavash, and Niao He. "Global convergence and variance reduction for a class of nonconvex-nonconcave minimax problems." Advances in neural information processing systems 33 (2020): 1153-1165.
>
> [5] Boyd, S., Boyd, S. P. & Vandenberghe, L. Convex optimization (Cambridge
> university press, 2004).
>
> [6] Azizian, Waïss, et al. "A tight and unified analysis of gradient-based methods for a whole spectrum of differentiable games." International conference on artificial intelligence and statistics. PMLR, 2020.
>
> [7] Zhang, Kaiqing, Zhuoran Yang, and Tamer Başar. "Multi-agent reinforcement learning: A selective overview of theories and algorithms." Handbook of reinforcement learning and control (2021): 321-384.
>
> [8] Gorbunov, Eduard, Nicolas Loizou, and Gauthier Gidel. "Extragradient method: O (1/k) last-iterate convergence for monotone variational inequalities and connections with cocoercivity." In International Conference on Artificial Intelligence and Statistics, pp. 366-402. PMLR, 2022.
>
> [9] Diakonikolas, Jelena, Constantinos Daskalakis, and Michael I. Jordan. "Efficient methods for structured nonconvex-nonconcave min-max optimization." In International Conference on Artificial Intelligence and Statistics, pp. 2746-2754. PMLR, 2021.

---

> ### Comment · Area_Chair_TcHu · 2025-08-06
>
> Dear Authors,
>
> Thanks for your reply! Overall, I wanted to hear about specific problems rather than other generic problem formulations that you mostly pointed out. For example by specific problem, I mean a problem such as $\min_x \| x\|_1: \|Ax-b\|\leq \delta$. Can you please provide some specific problems with specific functions to motivate your setting? These applications shouldn't be simple minimization because for minimization problems, we already have many other methods to use, so we wouldn't formulate them as a root-finding problem.
>
> 1. Can you please explain how Constrained minimization via Lagrangian formulation would fit in your template? Unless the problem is minimizing a smooth $f$ over $Ax=b$, you will have either nonsmoothness in $f$ (say, l1 norm) or constraints appearing in the dual (with inequality constraints). For the former template, the only example that comes to mind is $\min_x \|x\|^2: Ax=b$ which is not very interesting since one can just solve $\min_x \|Ax-b\|^2$ instead. If you can find a smooth loss where one needs L0-L1 and then show that with linear constraints $Ax=b$, the problem is still interesting. Then if you can show your L0-L1 assumption holds, this can be a specific example problem.
>
> 2. The example in [1] comes from a least squares problem, I am not sure: why would one not solve the minimization problem which is very simple in that case?
>
> 3. For [6], the only example they have for the template is the standard bilinar problem from Example 1 in [6] which is mostly a toy problem rather than problems coming from applications. Indeed many game theory problems have simplex as constraints. Can you tell us some specific unconstrained games where we need L0-L1 assumption?
>
> 4. GANs are a bit far-fetched since they will not satisfy any of the other assumptions in this paper. Still they can be acceptable if you can show that L0-L1 assumption is necessary to handle these problems. Is this possible?
>
> 5. Your Example 2 seems like a toy problem rather than a problem that can be more useful broadly. Can you comment if GlobalForsaken problem satisfy your assumption?
>
> 6. Example 3 in your appendix has constraints, how does it fit to your template?
>
> 7. Example 4 in your appendix: Can you tell us some specific games?
>
> Moreover, following-up on the question from Reviewer 4hmy, which of these applications, or which applications in general, does one need to handle the L0-L1 Lipschitzness rather than the standard Lipschitzness? Can you please provide motivating specific problems?
>
> One point your Example 1 makes is that it may be good to use L0-L1 Lipschitzness even when regular Lipschitzness hold but then you would have to justify what we lose in this transition. For example, the exponential blow-up in Theorem 3.5 or asymptotic-type rate in Theorem 3.6 are things we don't have with regular Lipschitzness. Needless to say we lose the ability to use constraints.
>
> Overall, can you please make your motivating problems a little more specific?
>
> Regarding terminology, even though other papers may have used it, I don't think it is a strong enough reason to use a terminology that can be misleading to others.
>
> Thanks!

---

> > ### Comment · Area_Chair_TcHu · 2025-08-09
> >
> > Dear Authors,
> >
> > This is a kind reminder that your response to the questions in my last message and also in Reviewer 4hmy’s last message would be very useful for the private discussion process that we will enter soon. This will be for discussion between reviewers and ACs.
> >
> > We look forward to hearing from you.
> >
> > Best,
> > AC

---

> > > ### Author Response · Authors · 2025-08-09
> > >
> > > Dear AC,
> > >
> > >
> > > Thank you again for the follow-up questions and for your interest in our work.
> > >
> > > In your original question, you mentioned,
> > >
> > > > “Can you please provide some motivating examples for root-finding problems, coming from applications?”
> > >
> > > We interpreted this as asking for general problem formulations that can be captured by the root-finding problems.
> > > Let us provide more details on what we consider the main contributions of our work and why we believe it is an excellent fit for a conference like NeurIPS (an assessment that all reviewers agreed with). The analysis of most classical methods for solving root-finding problems, referred to in our work as unconstrained variational inequality problems (which include unconstrained min-max optimization and unconstrained multi-player games), heavily relies on the L-Lipschitz assumption. This implies that there is a uniform bound on the Jacobian $\| J(x)\|\leq L$ for all x. Such analysis is unrealistic in several practical scenarios, expressed mainly as min-max problems, including GANs and Reinforcement Learning. In other words, an analysis of a method that uses the L-Lipschitz assumption does not really describe the behaviour of popular methods in these practical scenarios, as most of the time $\| J(x)\| \leq L$  is not true.
> > >
> > >
> > > With our work, we aim to relax this condition and, inspired by recent progress in the classical minimization setting, we propose the condition $\| J(x)\| \leq L_0 + L_1 \| F(x)\|^{\alpha}$, which is provably more relaxed than the $\| J(x)\|\leq L$  and we believe has the potential to explain the behaviour of popular algorithms in practical settings, similar to the behaviour practitioners notice in minimization settings.
> > >
> > >
> > > We introduce this condition and analyze the behaviour of the Extragradient method (arguably one of the most popular methods for solving VIPs and min-max problems) under this condition for different classes of problems (strongly monotone, monotone, and weak Minty). To achieve convergence, we propose novel adaptive step-size selection for the EG, and to back up our theory, we provide several examples (not real applications) where the problems are actually $\alpha$-symmetric $(L_0 ,L_1)$--Lipschitz and provably not L-Lipschitz (showing that our theory is valid at least for simple synthetic problems). At the end of the paper, we supplement our analysis with experiments.
> > >
> > >
> > > In our opinion, the above contributions are exactly what a theoretically oriented paper should aim for. Having said that, we appreciate the feedback and interest of the AC and the follow-up questions on real practical applications satisfying our setting, and below we do our best to provide more details given the limited time (less than a day) before the end of the discussion period.
> > >
> > > >  Can you please explain how constrained minimization via Lagrangian formulation would fit in your template?
> > >
> > > In our previous response, we provided this example as a general setting that can be formulated as a root finding problem. Unfortunately, given the limited time, we cannot provide a concrete example in this direction.  We agree with the AC that, if you can find a loss where one needs $(L_0,L_1)$ Lipschitz, and then show that with linear constraints Ax=b, the problem is still interesting. We try to have a version of our example 1 (from our paper) in the constrained setting, but the calculations do not trivially work out.
> > >
> > > > The example in [1] comes from a least squares problem, I am not sure: why would one not solve the minimization problem which is very simple in that case?
> > >
> > > Yes, you are correct in identifying that the problem is originally a minimization problem, and [1] transforms it into min-max. The primary reason is that [1] wants to employ variance-reduced methods on their objective function, which doesn’t have a finite-sum structure. They achieve this finite-sum structure by formulating it as a min-max problem.
> > >
> > > > For [6], the only example they have for the template is the standard bilinear problem from Example 1 in [6] which is mostly a toy problem rather than problems coming from applications. Indeed, many game theory problems have simplex as constraints. Can you tell us some specific unconstrained games where we need L0-L1 assumption?
> > >
> > >
> > > Indeed [6] has a bilinear problem, which we agree is a toy example rather than a problem coming from applications. In example 4 of our paper, we provide a specific unconstrained N-player game that satisfies our proposed $(L_0,L_1)$-Lipschitz assumption.

---

> > > > ### Author Response · Authors · 2025-08-09
> > > >
> > > > > GANs are a bit far-fetched ...Is this possible?
> > > >
> > > >
> > > > We agree with the far-fetched assessment for GANs. However, most existing work on theory for this setting assumes L-Lipschitz, which is a much more restrictive condition. At this stage, we are not sure if our  $(L_0, L_1)$ assumption is necessary to handle these problems, but we can aim to show an experimental connection. While it may be difficult to theoretically verify whether the $(L_0, L_1)$-Lipschitz assumption holds in this setting, we can attempt to investigate it empirically. We can aim to provide plots similar to our Figures 1 and 2 for GANs and potentially investigate the performance of our adaptive (novel step-size selection) EG in the training of the new models.
> > > >
> > > >
> > > > > Your Example 2 seems like a toy problem .... your assumption?
> > > >
> > > > Indeed, example 2 is a constructed example to satisfy our conditions. The primary goal of our experiment on the unconstrained GlobalForsaken was to check if our proposed step size can perform better than the constant step size rule suggested by the L-Lipschitz setting. In the given time of this discussion, we were not able to prove if the GlobalForsaken is $(L_0, L_1)$-Lipschitz, or not. We can do that in the camera-ready version of our work, and besides proofs, include also plots similar to figures 1 and 2 that experimentally satisfy our condition.
> > > >
> > > > >  Example 3 in your appendix has constraints, how does it fit to your template?
> > > >
> > > > Yes, example 3 has constraints. In this part, we provided examples showing that the corresponding operator satisfies the $(L_0, L_1)$ condition. We believe this is valuable, even though it is not directly related to our convergence guarantees, which are focused to the unconstrained setting.
> > > >
> > > > > Example 4 in your appendix: Can you tell us some specific games?
> > > >
> > > > Yes, for example 4, we can, for instance, have each of the players’ objective function $f_i(x_i, x_{-i}) = x_i^3$, which will satisfy the $(L_0, L_1)$ assumption for the concatenated operator F.
> > > >
> > > > > Moreover, following-up on the question from Reviewer 4hmy.... motivating specific problems?
> > > >
> > > > At this stage, as we already mentioned above, we have synthetic examples showing the importance of relaxing the classical L-Lipschitz assumption.
> > > >
> > > >
> > > > The standard Lipschitz assumption imposes a global bound, which can be conservative. In contrast, the $(L_0, L_1)$-Lipschitz condition provides an adaptive bound, allowing it to more accurately capture the local curvature of the problem. This makes our assumption more flexible compared to the traditional Lipschitz assumption.
> > > >
> > > >
> > > > Moreover, simple examples like $F(x) = x^2$ does not satisfy the Lipschitz condition, since
> > > > $| F(x) - F(y) | = | x^2 - y^2| = |x+y| |x - y|$ and the term $|x + y|$ is unbounded, making it impossible to find a finite constant $L$ that satisfies the Lipschitz assumption.
> > > >
> > > >
> > > > However, this F does satisfy the $\alpha$-symmetric $(L_0, L_1)$-Lipschitz condition. Specifically, $| F(x) - F(y) | = | x^2 - y^2| = |x+y| |x - y| = 2 (|(x + y)/2|^2)^{1/2} |x - y|$, which implies it satisfies the condition with $\alpha = \frac{1}{2}, L_1 = 2$, and any $L_0 > 0$. This example highlights how the $(L_0, L_1)$-based assumption can model problems that are not globally Lipschitz, yet still admit meaningful bounds that facilitate algorithmic analysis.
> > > >
> > > >
> > > >
> > > > > One point your Example 1 makes .... the ability to use constraints.
> > > >
> > > > Yes, we agree with the AC. The setting is much more general than regular Lipschitzness, and as such, there is the exponential blow-up in Theorem 3.5 and the asymptotic-type rate in Theorem 3.6. This, as we explained in our paper, aligns well with the $(L_0, L_1)$-smoothness results in the minimization setting. Our results are true generalizations of the regular Lipschitzness. To see this, consider the setting when $L_1 = 0$, which recovers the standard Lipschitz case. As shown in our theorem, when $L_1 = 0$, our analysis reduces to that of the classical Lipschitz setting (there is no exponential blow-up in Theorem 3.5 and no asymptotic-type rate in Theorem 3.6). This confirms that our framework generalizes the existing results while extending them to a richer class of problems.
> > > >
> > > >
> > > > >  Regarding terminology, even though other papers may have used it, I don't think it is a strong enough reason to use a terminology that can be misleading to others.
> > > >
> > > > We agree with the AC that simply using terminology that has been used in previous works is not a strong enough reason when it might be misleading to some readers. In our work, we clearly mention in all parts that we focus on “unconstrained variational inequalities”. From our viewpoint, this is clear terminology, but as we mentioned in our previous response, we understand that this might be confusing for some readers, and using “root-finding problem” might be a better name for the problem. We can easily and happily update this in the camera-ready version.

---

### Decision · Program_Chairs · 2025-09-17

**Decision:**

Accept (poster)

**Comment:**

This work focuses on solving root-finding problems, by relaxing the standard Lipschitzness assumption to an L0,L1 Lipschitzness, motivated by the recent advances in the minimization case.

The reviewing team found the developments interesting since they provided a new setting under which the classical extragradient method is analyzed and the reviewers thought that the techniques in this paper could be of value in the future to solve more complicated variational inequalities.

One main concern about this paper is regarding its motivation. Indeed, the L0-L1 smoothness in minimization arose from some interesting examples, whereas in this work, it seems to me that the extension is lacking motivation. When it comes to VIs, constraints are very important for handling even the most basic of interesting problem classes that require VI solvers (such as constrained optimization or matrix games).

Another remark from the discussion phase was: "The proposed step size requires knowing three parameters. Without a clear motivating example, its not clear if the juice is worth the squeeze."

The work needs to be polished further, especially in the appendix: It is not only that long equality chains such as those in pages 24, 26, 28, 31 and 32 are not the easiest to read, but also that many estimates are out of bounds, such as page 30 (here, even the equation numbering is covered with symbols), page 27, 26 and so on.

In summary, the authors are strongly recommended to provide **precise** motivating examples for their new assumption, in line with the discussion process between the authors and the AC and polish their proofs. Moreover, they should make sure to use the precise terminology to differentiate between VIs and root-finding problems and acknowledge clearly in their paper the shortcoming of their results (not able to handle constraints) and acknowledge clearly that extension to general VI is an open question.